# Generalization Measures under Controlled Covariate Shift: A Regime-Aware Benchmark

## Abstract

Predicting generalization from quantities available before target-test evaluation remains a central challenge in deep learning. The systematic benchmark of Jiang et al. (2020) evaluated many generalization measures, but it focused on independent and identically distributed (IID) settings. We revisit this problem for image classifiers evaluated under controlled corruptions and perturbations. Our study uses CIFAR-10-C/P, where the label space and task remain fixed while the input images are degraded or perturbed. This setting also allows us to revisit the robustness concerns raised by Dziugaite et al. (2020), who showed that the apparent reliability of generalization measures can depend strongly on experimental conditions. Our experiments show that the usefulness of generalization measures is strongly regime-dependent. In our exploratory decision analysis across three CNN-style architectures, sharpness- and input-gradient-based measures are among the leading individual signals, whereas family results are close and architecture dependent. Optimization-based measures, Information Criteria, and Sharpness-based measures provide additional regime-dependent signals in correlation or local-reliability analyses. Together, these findings suggest that model selection should not rely only on measures favored by IID evaluation. Instead, within the evaluated CIFAR-10-C/P setting and architectures, generalization measures should be treated as regime-dependent ranking signals whose utility must be evaluated for the intended corruption or perturbation setting.

## 1 Introduction

Deep learning models can fit large training sets and still perform well on held-out data, but predicting this behavior remains difficult (Zhang et al., 2017; Nagarajan & Kolter, 2019; Kawaguchi et al., 2022). For image classification, this question becomes sharper when clean IID evaluation is replaced by controlled image corruptions or perturbations. In this paper, "IID" refers specifically to evaluation on the clean CIFAR-10 test set. A useful generalization measure should indicate, before evaluating on the shifted test images, which trained models are likely to retain smaller train-to-test or clean-to-shifted performance gaps (Jiang et al., 2020; Dziugaite et al., 2020).

Many generalization measures have been proposed using model complexity, optimization dynamics, information theory, margins, and loss-landscape geometry. The number of candidates makes empirical evaluation important. Jiang et al. (2020) benchmarked 40 measures on CIFAR-10 and SVHN, and Dziugaite et al. (2020) showed that measure reliability can change under different experimental conditions. These studies provide the starting point for our work, but they mainly assess clean IID behavior. Two image-classification questions remain open:

    i. Image corruption and perturbation robustness: Existing IID benchmarks do not show whether a measure ranks models correctly when the same classifier is evaluated on corrupted or perturbed images such as CIFAR-10-C/P (Hendrycks & Dietterich, 2019).

    ii. Confidence calibration: A classifier's confidence can carry information about its errors (Guo et al., 2017; Krishnan & Tickoo, 2020). Recent studies connect calibration and generalization behavior (Wald

et al., 2021; Tada & Naganuma, 2023; Yoshida & Naganuma, 2024), but Calibration & Confidence measures are rarely evaluated as generalization measures in large-scale benchmarks.

We therefore revisit the empirical evaluation of generalization measures in the controlled image-shift setting defined by CIFAR-10-C/P, not distribution shift in general. We use "controlled covariate shift" operationally for these synthetic input changes and do not independently test formal invariance of $p(y \mid x)$. This scope is intentional. CIFAR-10-C and CIFAR-10-P keep the label space and task fixed while modifying the input images, so they allow us to ask whether IID-validated measures remain reliable for corruption and perturbation robustness without changing object categories or task semantics.

We follow the experimental philosophy of Jiang et al. (2020), which evaluates more than 40 generalization measures from multiple perspectives, and adapt it to CIFAR-10-C/P evaluation. In addition to conventional Baseline & Output-based, Norm & Margin-based, Sharpness-based, and Optimization-based measures, we include Calibration & Confidence and Information Criteria as candidate generalization-measure families. Through our evaluations, Sharpness-based measures are promising under IID settings, which is consistent with earlier observations that sharpness-related quantities can track model generalization. The main new finding is scoped to image corruptions and perturbations: some categories that are overlooked because they correlate weakly with the IID generalization gap, or because they have not usually been treated as generalization measures, can become more informative for CIFAR-10-C/P gaps than for the IID gap. At the decision level, several sharpness- and input-gradient-based measures are among the leading individual signals, while the leading family results are close and architecture dependent. Optimization-based measures, Information Criteria, and Sharpness-based measures provide additional regime-dependent signals under CIFAR-10-C/P. In addition, motivated by the concerns raised by Dziugaite et al. (2020), we analyze sign-error distributions to examine robustness across hyperparameter variations. The results show that several families can provide useful CIFAR-10-C/P ranking signals in some hyperparameter environments, but this does not make any family a uniformly reliable selector. The detailed definitions of the additional measure families are provided in Appendix A, and supplementary CIFAR-10 measure-level summaries and sign-error analyses are reported in Appendices C.1.2 and C.2. Thus, our results argue against treating any single generalization measure or measure family as a fixed proxy across clean IID and CIFAR-10-C/P regimes. They instead support a regime-aware view in which association, local reliability, and decision utility are evaluated separately for the intended corruption or perturbation setting.

Our key contributions are as follows:

- Systematic expansion of the experimental framework. We extend the benchmark of Jiang et al. (2020) from IID evaluation to CIFAR-10-C/P corruptions and perturbations, while adding Calibration & Confidence measures and Information Criteria to the conventional measure families.

- Empirical evidence of regime-dependent predictability. We show that IID predictivity does not reliably imply predictivity on CIFAR-10-C/P. Calibration & Confidence measures, Optimization-based measures, Information Criteria, and Sharpness-based measures can be useful candidates under these controlled image shifts, with different levels of correlation and local reliability.

- From correlation to model selection. We complement rank-correlation and sign-error analyses with a decision-level protocol that foregrounds individual selectors, compares them with source/clean baselines, and treats family averages as descriptive summaries rather than default selectors.

## 2 Related Work

### 2.1 Generalization Measures

To estimate the generalization performance of a trained model without relying on target test data, prior work has proposed many theoretical and empirical measures. Capacity-based measures, such as VC-dimension (Vapnik, 1991; Bartlett et al., 2019), depend mainly on the hypothesis space and do not account for the learned solution (Neyshabur et al., 2014). Norm- and margin-based measures instead use properties of the learned

classifier, including parameter scale and distance to the decision boundary (Krogh & Hertz, 1991; Neyshabur et al., 2015; Bartlett et al., 2017; Jiang et al., 2018). PAC-Bayes measures bound generalization risk through a posterior distribution over parameters relative to a prior (Dziugaite & Roy, 2017). Dynamics- and Sharpness-based measures use optimization behavior or the local curvature of the loss around the trained solution (Keskar et al., 2017; Foret et al., 2020). Our primary fixed selectors use source-domain information without shifted-target data, whereas ATC (Garg et al., 2022) uses predictions on unlabeled target examples to estimate target accuracy and therefore differs in both information budget and estimand. We do not perform a matched ATC comparison; Oracle and Random are evaluation references rather than substitutes for it.

Jiang et al. (2020) conducted a large-scale evaluation of such measures and found that PAC-Bayes and some Sharpness-based measures correlated well with IID generalization, while many traditional norm-based measures were less reliable. Dziugaite et al. (2020) further showed that these conclusions can depend on the experimental setup. The behavior of these measures is still less clear when the same image task is evaluated under corruptions and perturbations. Our study addresses this narrower question and adds Calibration & Confidence measures and Information Criteria to the candidate set.

## 2.2 Image Corruption, Perturbation, and Confidence Calibration

Clean IID evaluation does not fully describe the behavior of image classifiers under input degradation. Benchmarks such as ImageNet-C and CIFAR-10-C/P show that models with strong clean accuracy can lose accuracy under common corruptions or small perturbation sequences (Hendrycks et al., 2020; Hendrycks & Dietterich, 2019; Geirhos et al., 2018). These benchmarks are useful for our purpose because they modify the image input while keeping the task fixed.

Confidence calibration measures whether predicted probabilities match empirical correctness (Guo et al., 2017). Modern neural networks can be overconfident, and this overconfidence can become more visible under corrupted or perturbed inputs. Prior work has connected calibration with generalization behavior (Ovadia et al., 2019; Wald et al., 2021), but Calibration & Confidence measures (Naeini et al., 2015; Nixon et al., 2019) have rarely been evaluated as standalone predictors in large-scale generalization-measure benchmarks. We therefore include Calibration & Confidence metrics as candidate measures for CIFAR-10-C/P performance gaps.

# 3 Experimental Protocol

Our main experimental protocol centers on the CIFAR-10 suite. Building on the ranking-based benchmark of Jiang et al. (2020) and the robustness concerns raised by Dziugaite et al. (2020), we ask whether generalization measures induce model rankings that remain valid when evaluation moves from the clean CIFAR-10 test set to CIFAR-10-C/P. This design keeps the training distribution, label space, and model family fixed while changing the test images, allowing us to isolate whether a measure that is useful for IID model selection remains useful for corruption and perturbation robustness. We organize the evaluation into three stages: correlation analysis identifies promising measure families, local sign-error analysis checks whether their rankings are reliable along hyperparameter axes, and decision-level evaluation tests whether a fixed source-domain selector chooses a run with low OODGenGap before shifted-target outcomes are observed. We also broaden the set of measures by incorporating Calibration & Confidence measures and Information Criteria in addition to the traditional families of complexity, norm, sharpness, and Optimization-based measures. Details of the hyperparameter grids and model-specific settings are provided in Appendix B.

## 3.1 Primary CIFAR-10 Suite, Datasets, and Models

We adopt CIFAR-10 (Krizhevsky et al., 2009) as the primary IID benchmark and use CIFAR-10-C/P (Hendrycks & Dietterich, 2019) as the matched corruption and perturbation evaluation suite. The suite therefore consists of three matched evaluation targets: the standard CIFAR-10 test set for IID generalization, CIFAR-10-C for common corruptions, and CIFAR-10-P for perturbations. CIFAR-10-C and CIFAR-10-P preserve the same classification task while changing the test-time input images, making them suitable for testing whether measure-induced model rankings transfer from clean evaluation to controlled image shifts.

For this primary suite, we use a three-layer CNN (SimpleCNN), ResNetV2-32 (He et al., 2016), and Network in Network (NiN) (Lin et al., 2013).

CIFAR-10-C and CIFAR-10-P are related but statistically different targets: CIFAR-10-C applies corruptions to images, whereas CIFAR-10-P contains ordered perturbation sequences. This distinction affects the aggregation and uncertainty interpretation of target metrics, not the source-domain computation of generalization measures; pooled C/P results are therefore compact summaries rather than per-frame IID estimates. Target-separated decision results are reported in Appendix C.3.6.

We also report supplementary analyses in the appendix. Appendix C.1.2 provides the clean-to-shifted-test degradation scatter plot and measure-level summaries, and Appendix C.2 provides the full sign-error distributions across architectures and targets.

All hyperparameter grids, model-specific settings, and random seeds for the CIFAR-10 suite are provided in Appendix B.

### 3.2 Training Protocol

For the primary CIFAR-10 suite, each run is trained for 100 epochs. The sweep includes training hyper-parameters such as optimizer, learning rate, batch size, dropout, weight decay, and random seed; for NiN, depth and width are also swept. These controlled sweeps are central to our protocol because they create many candidate models whose rankings can be compared under clean, corrupted, and perturbed test targets. Additional training details are provided in Appendix B.

### 3.3 Generalization Measures

We classify generalization measures into six categories, partially adopting the taxonomy of Jiang et al. (2020).

**Baseline & Output-based Measures.** This category comprises structural capacity proxies and simple statistics of the predictive distribution that do not require gradient or optimization information. These serve as baselines reflecting architectural scale or output-shape properties. Representative metrics include VC-dimension approximations, total parameter counts, and output statistics such as cross-entropy or negative entropy.

**Norm & Margin-based Measures.** These measures capture the geometric properties of the learned function by combining decision boundary separation with parameter complexity. This family includes metrics quantifying the classifier margin, such as the inverse logit margin, as well as various measures of scale, such as parameter $\ell_2$ norms, spectral norms, and distances from initialization.

**Sharpness-based Measures.** Sharpness metrics quantify the sensitivity of the empirical loss to perturbations around the learned parameters, operationalizing the concept of local minima flatness. This category encompasses worst-case loss increases under structured perturbations, curvature approximations like the Hessian top eigenvalue, and PAC-Bayes bounds that interpret stability under stochastic perturbations (Foret et al., 2020).

**Optimization-based Measures.** These measures summarize the properties of gradients to capture training dynamics, stability, and local sensitivity. Unlike static baselines, these metrics explicitly leverage differential information. Examples include the variance of parameter gradients, which reflects the stochasticity of the training process, and the norms of gradients with respect to parameters or inputs.

The first four categories were included in the evaluations of Jiang et al. (2020) and Dziugaite et al. (2020). We add the following two categories because they may capture behavior under corrupted or perturbed inputs:

**Information Criteria.** Information criteria balance goodness-of-fit with complexity corrections. This family includes classical criteria like AIC, as well as measures adapted for deep learning that account for

posterior variability or local geometry, such as WAIC and TIC. These metrics penalize the model's negative log-likelihood by terms reflecting effective complexity.

**Calibration & Confidence.** Finally, we include measures that evaluate the alignment between predicted probabilities and empirical accuracy. This category focuses on the reliability of the model's uncertainty estimates, employing metrics such as Expected Calibration Error, Maximum Calibration Error, and adaptive binning variants.

All generalization measures are computed from source CIFAR-10 data before shifted-test evaluation; the training script partitions the CIFAR-10 training set into an 80% training split and a 20% clean validation split. Calibration & Confidence measures use the source training labels, while designated source/clean baselines may use clean validation labels. No main selector uses clean-test or CIFAR-10-C/P information; clean-test quantities appear only as separate references in the appendix.

More detailed descriptions of the metrics are provided in Appendix A.

### 3.4 Correlation-based Evaluation

We first evaluate each generalization measure as a ranking signal. The purpose of this stage is screening: we ask whether a measure ranks candidate models in the same order as a clean, corrupted, or perturbed target gap. This perspective follows Jiang et al. (2020): a useful generalization measure should help identify, among trained candidate models, which one is likely to suffer a larger or smaller train–test degradation before accessing the target test labels.

**Target quantities.** For the CIFAR-10 suite, we evaluate both clean and shifted target quantities. We denote the standard CIFAR-10 train–test gap by `GenGap_CIFAR10`. For CIFAR-10-C/P, we use two complementary targets. First, `OODGenGap_C` and `OODGenGap_P` are train-to-shifted-test gaps, defined as the difference between CIFAR-10 training accuracy and CIFAR-10-C/P test accuracy. Second, `OODTestGap_C` and `OODTestGap_P` are clean-to-shifted-test degradation targets, defined as the difference between clean CIFAR-10 test accuracy and CIFAR-10-C/P test accuracy. The first target matches the train–test generalization-gap perspective of prior work, while the second isolates shifted-test degradation by removing the training-accuracy term.

**Granulated Kendall score $\Psi$.** Following Jiang et al. (2020), and matching our correlation-analysis implementation, we avoid computing a single global correlation over all runs, since such a score can be dominated by large architectural or hyperparameter changes. Instead, we form local subspaces in which exactly one hyperparameter varies while all others are fixed. For a valid subspace $s$ with $n_s$ runs, measure values $\mu_i$, and target gaps $g_i$, we compute Kendall's tau-a as

$$\tau_s(\mu, g) = \frac{2}{n_s(n_s - 1)} \sum_{i < j} \text{sign}(\mu_i - \mu_j) \, \text{sign}(g_i - g_j), \tag{1}$$

where ties contribute zero through the sign function. We then average over valid subspaces:

$$\Psi(\mu, g) = \frac{1}{|\mathcal{S}|} \sum_{s \in \mathcal{S}} \tau_s(\mu, g). \tag{2}$$

A positive $\Psi$ indicates that larger measure values tend to rank models as having larger target gaps, while values near zero indicate little reliable ranking signal. Comparing $\Psi$ across clean and CIFAR-10-C/P targets tests whether the measure-induced ordering is preserved under image corruptions and perturbations.

### 3.5 Local Reliability Evaluation

The correlation stage identifies promising measure families on average, but an average Kendall score can hide local ranking failures. We therefore evaluate local reliability using sign-error distributions, following the robustness motivation of Dziugaite et al. (2020). The goal is to detect whether a measure fails along particular hyperparameter axes, such as learning rate or weight decay, even when its aggregate correlation is favorable.

**Sign-error distributions.** Matching our sign-error implementation, we define a hyperparameter configuration, excluding random seed, as a combo. An environment is a pair of combos $(h_1, h_2)$ that differ in exactly one hyperparameter. For each environment, we compare repeated runs from the two combos and compute the pairwise sign-error between the measure $\mu$ and target gap $g$:

$$\ell_{ij} = \frac{1 - \text{sign}(\mu_i - \mu_j)\,\text{sign}(g_i - g_j)}{2}.$$ (3)

Thus, $\ell_{ij} = 0$ when the measure and target agree on the ordering of the two runs, $\ell_{ij} = 1$ when they disagree, and ties contribute $1/2$. When noise filtering is enabled, pairs are weighted using the Hoeffding-based weight

$$w_{ij} = \max\left\{ 0, \left( 1 - 2\exp\left[ -2n\left( \frac{|g_i - g_j|}{2} \right)^2 \right] \right)^2 - 0.5 \right\},$$ (4)

where $n$ is the relevant test-set size. Environments with small effective sample size

$$n_{\text{eff}} = \frac{(\sum w_{ij})^2}{\sum w_{ij}^2}$$ (5)

are filtered out. The environment-level sign-error is then estimated as

$$\widehat{\text{SE}} = \frac{\sum w_{ij}\ell_{ij}}{\sum w_{ij}}.$$ (6)

We report the distribution of $\widehat{\text{SE}}$ across environments and hyperparameter axes. This exposes failure modes that can be hidden by a single mean score, such as measures that look predictive on average but fail badly for particular hyperparameter changes.

### 3.6 Decision-level Model Selection Evaluation

The third CIFAR-10 evaluation asks whether a fixed source-domain measure can select a run with a low OOD generalization gap. The held-out targets are `final/OODGenGap_C` and `final/OODGenGap_P`; both are minimized. Each target is training accuracy minus shifted-test accuracy. Minimizing it is not generally equivalent to maximizing shifted-test accuracy and can favor a run with lower source and target accuracy, so this task does not establish absolute shifted-accuracy or deployment utility.

**Candidate sets and selectors.** A candidate set contains finished runs with the same architecture, sweep identity, and shifted target. For the reported decision analysis, we restrict each set to runs with finite source `cross.entropy` $\leq 0.01$. Because the analysis does not establish that this threshold was prespecified or assess sensitivity to alternatives, and because cross-entropy is also a selector, all rankings are exploratory and conditional on the restricted pool. Each measure or baseline is evaluated separately. With the selector direction fixed before target evaluation, the selected run is

$$\hat{r}_q(C) = \begin{cases} \arg\max_{r \in C} s_q(r), & q \in \{\text{accuracy-based selectors, } \texttt{negative.entropy}\}, \\ \arg\min_{r \in C} s_q(r), & \text{otherwise,} \end{cases}$$ (7)

where the direction is fixed before target evaluation. All selectors are minimized except accuracy-based selectors and negative.entropy, which are maximized. Oracle OOD is the run with the smallest held-out OODGenGap and is used only as a reference.

**Family-level aggregation.** We also report a descriptive family-level summary by averaging individual-selector outcomes within each candidate set and then across candidate sets. This blind family-commitment summary neither selects the best member nor defines an ensemble; it remains conditional on family composition, selector support, and missingness.

**Source/clean baselines.** The main Source/clean baseline contains final-epoch training accuracy and loss and clean validation accuracy and loss. The validation quantities are read from `cifar10/val_acc` and `cifar10/val_loss`; accuracy is maximized and loss is minimized. The `cross.entropy` selector is a post-training evaluation-mode measure on the source training split and is minimized. Clean-test quantities are excluded from the main baseline and reported separately in Appendix C.3.1.

**Selection metrics.** No fixed selector uses CIFAR-10-C/P information when choosing a run. For a candidate set $C$, let $y(r)$ denote the held-out OODGenGap, let $r^\star = \arg\min_{r \in C} y(r)$ be Oracle OOD, let $r^- = \arg\max_{r \in C} y(r)$ be the worst run, and let $\hat{r}_q$ be the selected run. Regret is

$$R_q(C) = y(\hat{r}_q) - y(r^\star). \tag{8}$$

Normalized regret is

$$\bar{R}_q(C) = \frac{R_q(C)}{y(r^-) - y(r^\star)}, \tag{9}$$

with zero assigned when the oracle-to-worst range is zero. We report mean normalized regret, top-10% and top-20% hit rates, and the number of candidate sets. Appendix C.3.1 gives implementation details and the bootstrap procedure.

**Leakage boundary.** For fixed source-domain selector evaluation, the leakage boundary excludes shifted-target information, not labeled source information. Source training labels used by calibration measures and clean validation labels used by source baselines are allowed. Clean CIFAR-10 test quantities and all CIFAR-10-C/P target information are unavailable to these selectors; Oracle OOD is a target-aware evaluation reference, not a deployable selector.

**Proxy-supervised meta-selection.** Appendix C.3.7 studies a separate information regime that uses labeled proxy-shift outcomes to rank measures or families before evaluation on a held-out target set. The held-out target outcome is not used for that choice, but the procedure is proxy-supervised rather than source-domain-only, and the C↔P experiment is not literal leave-one-CIFAR-10-C-corruption-type-out validation.

## 4 Results

### 4.1 Regime-Dependent Associations under Corruptions and Perturbations

We begin with CIFAR-10 models evaluated on CIFAR-10-C and CIFAR-10-P. This setting keeps the training distribution, label space, and model families fixed while changing the test-time input images, making it a controlled regime for asking whether an IID generalization measure remains useful for corrupted or perturbed images. Figure 1 compares each measure family's summarized Granulated Kendall score for the IID generalization gap against its score for the train-to-shifted-test gaps, `OODGenGap_C` and `OODGenGap_P`. Each plotted point corresponds to one measure, architecture, and shifted target. The quadrant annotations in each panel summarize the global sign pattern: Q1 contains measures that are positively predictive in both clean and shifted regimes, whereas Q2 contains measures that are weak or negative under IID evaluation but become positive under CIFAR-10-C/P. We read Q1 and Q2 as shifted-test-favorable regions, with Q2 marking signals that IID-only analysis would tend to miss. We do not interpret the quadrant summaries as architecture-invariant effect sizes. They are descriptive screening summaries; the architecture-specific tables and decision-level results are needed to assess whether these signals persist beyond the aggregate visualization.

**Sharpness remains the main continuity with IID evaluation.** Sharpness-based measures provide the clearest continuity with the IID-centered picture. Many sharpness measures fall in the positive shifted-test region and often remain positive in IID as well, indicating that this family continues to capture a useful sensitivity signal under controlled corruptions and perturbations. This is an important consistency check rather than the main new point: sharpness-related quantities were already among the more informative families in prior IID-centered evaluations (Jiang et al., 2020).

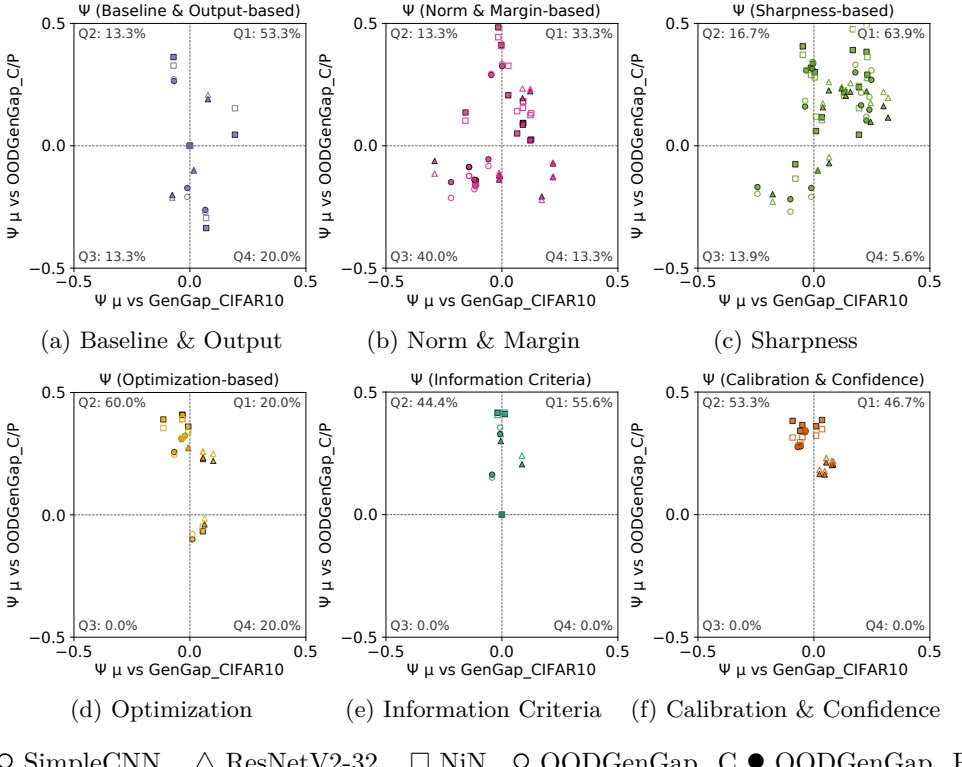

(a) Baseline & Output    (b) Norm & Margin    (c) Sharpness

(d) Optimization    (e) Information Criteria    (f) Calibration & Confidence

○ SimpleCNN    △ ResNetV2-32    □ NiN    ○ OODGenGap_C    ● OODGenGap_P

Figure 1: CIFAR-10 suite under corruptions and perturbations: relationship between IID generalization-gap correlation and shifted generalization-gap correlation. Each point is a measure by architecture by shifted target combination. The x-axis is $\Psi$ for `GenGap_CIFAR10`, and the y-axis is $\Psi$ for `OODGenGap_C` or `OODGenGap_P`. Q1 contains measures with positive IID and shifted-test correlations. Q2 contains measures that are weak or negative for IID but positive for CIFAR-10-C/P. Q1 and Q2 are the shifted-test-favorable regions. The main pattern is that Sharpness-based measures remain comparatively stable, while Optimization-based measures, Information Criteria, and Calibration & Confidence measures concentrate more strongly in Q1/Q2 than expected from their IID-only behavior.

**Corruptions and perturbations expose overlooked families.** The more distinctive finding is visible in Figures 1d–1f. Optimization-based measures, Information Criteria, and Calibration & Confidence measures move toward the shifted-test-favorable quadrants, especially Q1 and Q2, even though several of them are not strong IID predictors. This means that IID-only evaluation can miss measure families whose signal becomes visible when the target is corruption or perturbation robustness. For Calibration & Confidence, however, the main positive shifted-test correlation is concentrated in NiN: Table 6 reports OOD-C $\Psi = 0.355$ for NiN, whereas SimpleCNN and ResNetV2-32 are near zero with OOD-C $\Psi = 0.044$ and $0.092$, respectively. This architecture dependence prevents us from interpreting Calibration & Confidence as a uniformly strong correlation signal across all architectures. Baseline & Output-based and Norm & Margin-based families are mixed at the family level, but this family-level summary hides an important exception: `cross.entropy` and `negative.entropy` are competitive individual comparison points in the restricted decision analysis. Information Criteria should be interpreted here as comparatively informative correlation signals and candidates for further validation, not as uniformly strong decision-level selectors. The Calibration & Confidence values in this analysis are computed from labeled source CIFAR-10 data; shifted-test labels enter only when the precomputed measures are evaluated against CIFAR-10-C/P outcomes.

The correlation pattern is not solely driven by including training accuracy in the target gap. Appendix C.1.2 reports the clean-to-shifted degradation analysis, where `OODTestGap_C` and `OODTestGap_P` remove the training-accuracy term and compare clean CIFAR-10 test performance directly against CIFAR-10-C/P test performance.

The signal is generally weaker under this stricter target. Appendix C.1.4 provides the corresponding measure-level tables across architectures and targets, and Appendix C.1.1 separates CIFAR-10-C from CIFAR-10-P.

## 4.2 Local Reliability Depends on the Hyperparameter Axis

**From average association to local reliability.** The scatter plots above answer which measure families are promising on average, but model selection is usually a local ranking problem. A practitioner often compares models that differ along one hyperparameter axis, and a measure with a favorable average $\Psi$ can still be unreliable if it reverses these local orderings. We therefore audit the CIFAR-10-C/P signals with sign-error distributions. For NiN on CIFAR-10-C, Figure 2 reports environment-level sign-error rates separated by the hyperparameter axis that defines the local comparison. Lower values indicate that the measure preserves the target-gap ordering more reliably, whereas values near 0.5 indicate chance-level local ranking.

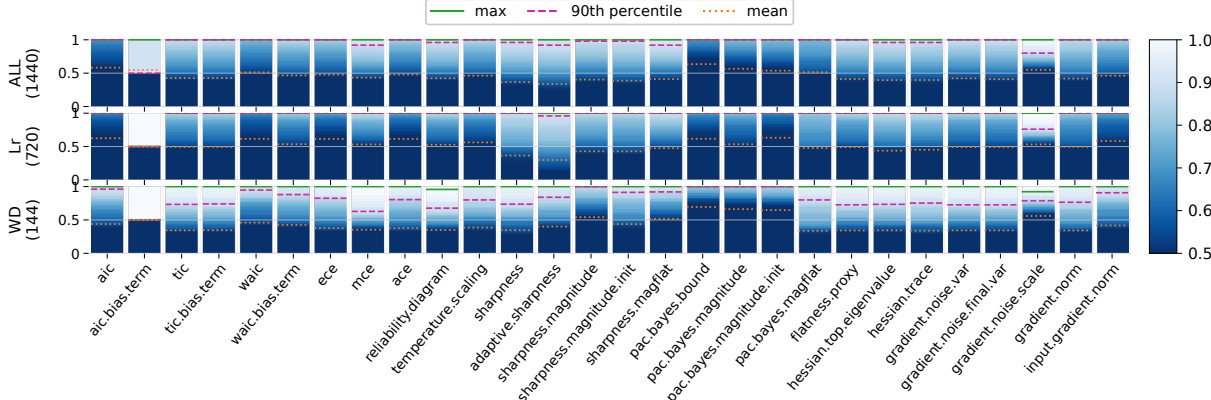

Figure 2: Local reliability on CIFAR-10-C: sign-error distributions for NiN. Each column shows the distribution of environment-level sign-error values for one generalization measure, separated by the hyperparameter axis that defines the local environment. The plot reveals that average shifted-test correlation is not enough: several promising measures remain difficult to use along the learning-rate axis but become much more reliable along weight-decay environments. The orange, magenta, and green lines indicate the mean, 90th percentile, and maximum, respectively.

**Hyperparameter-axis dependence.** The main lesson is that reliability is strongly axis-dependent. Learning-rate environments remain difficult: many measures have mean sign-errors close to or above chance level, and their upper tails indicate badly misranked local comparisons. Weight-decay environments show a different pattern, with lower sign-errors for a broader set of measures, including Calibration & Confidence measures, Information Criteria, and several sharpness- or optimization-related quantities. Thus, a high average CIFAR-10-C/P correlation is not sufficient for model selection; the measure must also be reliable over the hyperparameter direction actually being searched.

**Reproducibility across CIFAR-10-C/P.** The CIFAR-10-P results in Appendix C.2 show the same qualitative structure for NiN, and the appendix also reports the corresponding distributions for SimpleCNN, ResNetV2-32, and the clean-to-shifted degradation targets. This supports a narrow interpretation of the CIFAR-10-C/P results: Optimization-based measures, Information Criteria, and Calibration & Confidence measures are useful candidates, but they should be used only after checking local sign-error behavior in the relevant hyperparameter environment.

## 4.3 Decision-Level Model Selection Separates Individual and Family Results

**Selection task.** The correlation and sign-error analyses do not directly test run selection for a prespecified gap target. For each architecture–sweep–target set, a fixed source-domain selector chooses one run without

observing `OODGenGap_C/P`, and we compare the selected run with the minimum-gap Oracle OOD run. This comparison concerns low train-to-shifted-test OODGenGap, not maximum shifted accuracy. Calibration & Confidence selectors use labeled source CIFAR-10 data at selection time, but they do not use CIFAR-10-C/P labels, target ranks, or shifted-test metrics in this fixed-selector evaluation.

Table 1: Decision-level selection of runs with low CIFAR-10-C/P OOD generalization gap. Rows are selected from the full individual-selector analysis in Table 27. Lower mean normalized regret is better; higher top-20% hit rate is better.

| Selector | Mean normalized regret ↓ | 95% CI | Top-20% hit ↑ | # candidate sets |
|---|---|---|---|---|
| oracle.ood | 0.000 | [0.000, 0.000] | 1.000 | 26 |
| sharpness.magnitude.init | 0.132 | [0.084, 0.188] | 0.577 | 26 |
| input.gradient.norm | 0.135 | [0.088, 0.189] | 0.654 | 26 |
| sharpness | 0.144 | [0.097, 0.196] | 0.615 | 26 |
| clean.val.acc | 0.178 | [0.111, 0.255] | 0.577 | 26 |
| cross.entropy | 0.195 | [0.122, 0.275] | 0.500 | 26 |
| ece | 0.206 | [0.138, 0.283] | 0.423 | 26 |
| clean.val.loss | 0.435 | [0.302, 0.573] | 0.231 | 26 |
| random | 0.394 | [0.360, 0.426] | 0.203 | 26 |

Note: These are exploratory point estimates under an analysis restriction requiring finished runs with `cross.entropy` $\leq 0.01$; the reported analysis does not establish the threshold's provenance or sensitivity. Because cross-entropy is also evaluated as a selector, this filter restricts its evaluated range. Selectors choose one run from each architecture–sweep–target set without using CIFAR-10-C/P target information. Oracle OOD is the run with minimum `OODGenGap_C` or `OODGenGap_P`. Regret is normalized by the oracle-to-worst OODGenGap range. The intervals are candidate-set bootstrap summaries conditional on the observed benchmark; they do not model dependence among sets sharing architecture, sweep, or trained runs and are not intervals over independently sampled datasets.

**Individual-selector results.**    Table 1 reports the exploratory individual-selector ranking under the restriction in Section 3.6. The lowest non-oracle mean normalized regrets are 0.132 for `sharpness.magnitude.init`, 0.135 for `input.gradient.norm`, and 0.144 for `sharpness`; clean validation accuracy, source `cross.entropy`, and ECE obtain 0.178, 0.195, and 0.206. Residualized ECE and MCE are weaker at 0.307 and 0.382, separating residual association from incremental decision utility. On common support, source cross-entropy and negative entropy also have lower point-estimate regret than the Calibration & Confidence family average and ECE (Table 25).

**Descriptive family-level aggregation.**    Table 23 reports a descriptive blind-family-commitment summary, not an implementable selector. Optimization-based, Calibration & Confidence, and Information Criteria obtain mean regrets of 0.221, 0.223, and 0.242; the paired Calibration-minus-Optimization difference is 0.002 with interval $[-0.047, 0.060]$, so the data do not identify a dominant family. Calibration is lower than Random but is not separated from the Source/clean group, and the clean-test reference (0.326) does not improve on the Source/clean average (0.283). These summaries remain conditional on the observed, potentially dependent candidate sets.

Family results are secondary to named individual selectors: the observed Calibration spread depends on the included members and does not imply robustness to an unseen measure, and family orderings vary by architecture.

**Target and severity checks.**    Target-separated results show similar but non-identical selector orderings for CIFAR-10-C and CIFAR-10-P; the latter remains a sequence benchmark rather than a per-frame IID sample (Appendix C.3.6). The proxy-supervised C↔P stress test improves on Random for OODGenGap but not on the clean-selected reference, weakens for OODTestGap and ResNetV2-32, and is not literal leave-one-corruption-type-out validation (Appendix C.3.7). Severity-controlled rankings also vary across small, unequal candidate-set supports and are therefore exploratory (Appendix C.3.5).

Table 2: **Paired decision-level regret differences on CIFAR-10-C/P.** Each row reports the candidate-set paired difference in mean normalized regret between Calibration & Confidence and a comparator family. Negative values favor Calibration & Confidence. Intervals are percentile 95% paired bootstrap intervals obtained by resampling candidate sets.

| Comparison | $\Delta$ normalized regret | 95% paired CI | # candidate sets |
|---|---|---|---|
| Calibration - Source/clean baselines | -0.060 | [-0.135, 0.011] | 26 |
| Calibration - Optimization-based | 0.002 | [-0.047, 0.060] | 26 |
| Calibration - Random | -0.171 | [-0.251, -0.082] | 26 |

Note: These paired candidate-set bootstrap intervals are descriptive summaries conditional on the observed benchmark and do not model shared architecture/sweep/run dependence. Family averages are additionally conditional on the included, unequally sized family memberships specified in Appendix C.3.1.

## 5 Discussion

### 5.1 How Corruptions and Perturbations Change Measure Validity

The central empirical lesson is that the validity of a generalization measure depends on the evaluation regime. In the CIFAR-10 suite, the shifts are controlled: corruptions and perturbations alter the input while preserving the label space and the underlying task. Under the restricted OODGenGap decision analysis, sharpness-magnitude and input-gradient selectors have the lowest exploratory individual point estimates, while descriptive family averages are close and architecture dependent.

This contrast reframes the role of IID evaluation. The issue is not simply that IID correlations are noisy, but that they can point to measures that do not transfer to a corruption or perturbation target. A concise statement of the result is

$$\rho(m, G_{\mathrm{IID}}) \not\approx \rho(m, G_{\mathrm{OOD}}),$$

where $m$ denotes a generalization measure, $G_{\mathrm{IID}}$ is the IID generalization gap, and $G_{\mathrm{OOD}}$ is the relevant shifted-test gap. This notation summarizes an empirical mismatch in the observed benchmark rather than a population theorem. The implication is not that our benchmark identifies a universal measure family, but that correlation, local reliability, and low-gap decision evidence should be checked separately. For controlled corruptions and perturbations, IID evidence alone is not enough to decide which measure should guide model selection. This conclusion concerns controlled CIFAR-10-C/P corruptions and perturbations with the CNN-style architectures studied here; it is not evidence for the same ordering under natural or semantic shifts, larger datasets, transformers, or modern augmentation-heavy training.

### 5.2 Calibration and Incremental Decision Value

Calibration & Confidence measures are computed from labeled source CIFAR-10 data without shifted-target information. Their raw associations may combine source fit, confidence dispersion, and error–confidence alignment. After controlling for source cross-entropy and source error, some calibration association remains, but residualized calibration selectors do not retain a decision-level advantage. The benchmark therefore does not identify a causal calibration-specific mechanism or incremental actionable value beyond the evaluated source statistics.

### 5.3 Guidelines for Regime-Aware Model Selection

The consequence is that no single generalization measure should be trusted on CIFAR-10-C/P only because it works well under IID evaluation. The three evaluation stages answer distinct questions: association screens measures, sign error tests local rank consistency, and normalized regret evaluates selection for low OODGenGap. A regime-aware analysis should match the criterion and information budget to the intended decision rather than infer selection utility from correlation alone.

Thus, model-selection use should check not only whether a measure has a high average $\Psi$, but whether it preserves rankings in the local hyperparameter environment being searched. Named individual selectors and source/clean baselines are more directly interpretable than family averages, which depend on included members and support. The proxy experiment is a separate diagnostic because it uses labeled proxy-shift outcomes; its architecture- and objective-dependent transfer must be revalidated for the intended regime.

## 6 Limitations and Future Work

**Scope and external validity.** The evidence concerns SimpleCNN, ResNetV2-32, and NiN on controlled CIFAR-10-C/P corruptions and perturbations. It does not establish selector orderings for natural or semantic shifts, larger datasets, transformers, or modern training recipes. Broader shift regimes and model classes are therefore a primary extension.

**Decision target and eligible pool.** The decision analysis minimizes OODGenGap rather than shifted accuracy, so it does not establish deployment-accuracy selection. Rankings are also conditional on retaining finished runs with finite source `cross.entropy` $\leq 0.01$. The analysis does not establish that this threshold was prespecified or assess sensitivity to alternatives, and filtering on an evaluated selector restricts its range; we therefore report the rankings as exploratory for the retained pool. Future work should evaluate absolute shifted-accuracy objectives and threshold sensitivity.

**Aggregation and uncertainty.** Candidate-set bootstrap summaries do not model shared architectures, sweeps, or runs, and family summaries depend on member composition, support, and missingness. CIFAR-10-P additionally contains correlated ordered sequences: equal sequence lengths preserve equal source-image weighting within each evaluated cell, but we did not apply source-image block bootstrap or compare alternative sequence summaries (Appendix B.1.1). Dependence-aware and whole-sequence uncertainty analyses remain future work.

**Information regimes and missing comparisons.** Primary selectors use only source-domain information, including labels where required, but no CIFAR-10-C/P information, whereas the proxy analysis uses labeled proxy-shift outcomes and shows architecture- and objective-dependent transfer. The source/clean baseline set is non-exhaustive and does not include combined validation selectors. We did not perform matched ATC, which uses unlabeled target predictions to estimate accuracy, or literal leave-one-corruption-type-out validation; the C$\leftrightarrow$P study crosses two Gaussian-noise constructions rather than distinct CIFAR-10-C corruption types. These remain direct extensions for comparing information budgets and cross-type transfer.

**Mechanistic interpretation.** Correlation, sign error, and regret measure complementary empirical utility rather than causal mechanisms. Calibration residualization leaves some association but does not yield incremental decision utility over source-loss baselines under this protocol. Future work should isolate when calibration, source fit, confidence, information criteria, optimization dynamics, and sharpness provide distinct transferable information.

## 7 Conclusion

This work revisits generalization-measure benchmarking for controlled CIFAR-10-C/P corruptions and perturbations. Across controlled CIFAR-10-C/P shifts, IID predictivity does not reliably transfer to shifted-test predictivity. Calibration & Confidence, Information Criteria, and Optimization-based measures can provide useful regime-dependent correlation or local-reliability signals. The decision-level model-selection experiment shows that correlation-based predictivity is not sufficient by itself. Under the stated restriction, sharpness- and input-gradient-based selectors have the lowest exploratory individual point estimates, family averages are close and architecture dependent, and residualized ECE/MCE are weaker than their raw versions and source cross-entropy. The primary analysis uses only source-domain information, including labels where required, but no shifted-target information, whereas the C$\leftrightarrow$P stress test is proxy-supervised. These

results support regime-aware evaluation rather than a universal default and are limited to the evaluated CIFAR-10-C/P constructions and CNN-style architectures.

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

# A    More Details on Generalization Measures

Table 3: Summary of generalization measures evaluated in this study, including those applicable beyond the IID regime. The measures are categorized into six groups based on their source of signal and intended use as generalization predictors.

| Measure Name | Description |
| --- | --- |
| Baseline & Output-based Measures | |
| vcdim | Approximate VC-dimension bound based on network depth and width. |
| params | Total number of trainable model parameters. |
| magnitude | The $\ell_2$ norm of the parameter vector. |
| cross.entropy | Average cross-entropy loss on the split passed to the measure evaluator. |
| negative.entropy | Negative entropy of the predictive distribution (measure of confidence/peakiness). |
| Norm & Margin-based Measures | |
| inverse.margin.p10 | Inverse of the 10th percentile of the logit margin distribution. |
| l2.over.margin.p10 | Ratio of parameter $\ell_2$ norm to the 10th percentile margin. |
| l1.over.margin.p10 | Ratio of parameter $\ell_1$ norm to the 10th percentile margin. |
| margin.normalized.param.norm | Parameter norms scaled by the robust margin. |
| spectral.norm.per.layer | Sum of spectral norms computed per layer. |
| spec.sum | Sum of the spectral norms of the weight matrices. |
| spec.prod | Product of the spectral norms of the weight matrices. |
| frobenius.distance | Frobenius distance between the final weights and initialization. |
| path.norm | Path-norm; a scale-invariant capacity measure. |
| fisher.rao.norm | Fisher-Rao norm; geometry-aware norm based on the Fisher Information Matrix. |
| Sharpness-based Measures | |
| sharpness | Worst-case loss increase within a neighborhood (standard definition). |
| adaptive.sharpness | Sharpness measured with an adaptive neighborhood size. |
| sharpness.magnitude | Sharpness scaled by the magnitude of the parameters. |
| sharpness.magnitude.init | Sharpness scaled by the magnitude relative to initialization. |
| sharpness.magflat | Flatness measure using multiplicative perturbations proportional to weight magnitude. |
| pac.bayes.bound | PAC-Bayes generalization bound (interpreted as stability/flatness). |
| pac.bayes.magnitude | PAC-Bayes bound incorporating parameter magnitude. |
| pac.bayes.magnitude.init | PAC-Bayes bound incorporating distance from initialization. |
| pac.bayes.magflat | PAC-Bayes variant related to magnitude-proportional flatness. |
| flatness.proxy | Flatness proxy based on Elastic Weight Consolidation (EWC). |
| hessian.top.eigenvalue | The largest eigenvalue of the Hessian matrix (worst-case curvature). |
| hessian.trace | The trace of the Hessian matrix (average curvature). |
| Optimization-based Measures | |

Table 3 – continued from previous page

| Measure Name | Description |
|---|---|
| `gradient.noise.var` | Variance of the stochastic gradients during training. |
| `gradient.noise.final.var` | Variance of the gradients measured at the final solution. |
| `gradient.noise.scale` | Ratio of gradient noise variance to the mean squared gradient norm. |
| `gradient.norm` | Aggregate norm of the gradients across minibatches. |
| `input.gradient.norm` | Norm of the gradient of the loss with respect to the input (sensitivity). |
| Information Criteria | |
| `aic.bias.term` | The complexity penalty term of AIC (based on parameter count). |
| `tic.bias.term` | The complexity penalty term of TIC (trace term). |
| `waic.bias.term` | The complexity penalty term of WAIC (log-likelihood variance). |
| Calibration & Confidence Measures | |
| `ece` | Expected Calibration Error; weighted average difference between confidence and accuracy in bins. |
| `mce` | Maximum Calibration Error; the maximum gap between confidence and accuracy across bins. |
| `ace` | Adaptive Calibration Error; uses adaptive binning based on quantiles. |
| `reliability.diagram` | Calibration error derived from bin-wise reliability diagrams. |
| `temperature.scaling` | Calibration error calculated after post-hoc optimal temperature scaling. |

Table 3 presents the generalization measures adopted in this study. The following subsections describe the evaluated measures or their implementation families, including the Calibration & Confidence and Information Criteria families added to the original Jiang et al. (2020) taxonomy.

## A.1 Baseline & Output-based Measures

**vcdim.** We report an architecture-based VC-dimension proxy. Classical VC theory motivates capacity control via VC-style complexity measures (Vapnik, 1991). In our experiments, we use the standard parameter-count proxy commonly used as a simple capacity baseline in generalization-measure evaluations (Jiang et al., 2020):

$$\text{vcdim} = W \log W, \tag{10}$$

where $W$ is the number of trainable parameters. This proxy is also consistent with known scaling behavior of VC/pseudodimension upper bounds for piecewise-linear networks (Bartlett et al., 2019). For 4D image inputs, we instead use the $\mu_{\text{VC}}$ upper bound (Eq. 15) from Fantastic Generalization Measures and Where to Find Them by Jiang et al. (2020). We computed this measure to create a capacity-only baseline to test whether architecture scale alone predicts IID/OOD generalization gaps (Jiang et al., 2020; Dziugaite et al., 2020).

**params, magnitude, cross.entropy, and negative.entropy.** These measures are simple architecture- or output-based baselines. `params` is the number of trainable parameters, `magnitude` is the parameter $\ell_2$ norm, `cross.entropy` is the empirical cross-entropy on the data split used for measuring the model, and `negative.entropy` is the negative predictive entropy. They are included to distinguish signals that come from model size or output confidence from signals that require margins, curvature, or optimization dynamics.

## A.2 Norm & Margin-based Measures

**inverse.margin.p10.** For each sample $(x_n, y_n)$, define a margin $m_n$. For all classification runs in this study, $m_n$ is computed from logits. Margin distributions, especially lower-tail statistics, have been shown to correlate with generalization gaps and are used in practice as predictors (Jiang et al., 2018; 2020). Let $q_{0.10}$ be the empirical 10th percentile of $\{m_n\}_{n=1}^N$. We compute a sign-preserving, clamped inverse:

$$\text{inverse.margin.p10} = \frac{1}{\text{clip}(q_{0.10}, -\epsilon, \epsilon)} \quad \text{where} \quad \text{clip}(z, -\epsilon, \epsilon) = \text{sign}(z) \max(|z|, \epsilon). \tag{11}$$

We computed this measure to create a low-tail separation/robustness signal (small-magnitude lower-tail margins $\Rightarrow$ larger score) used as a predictor of generalization gaps (Jiang et al., 2018; Bartlett et al., 2017).

**l2.over.margin.p10.** This measure is an alias of `margin.normalized.param.norm` under default settings. Norm- and margin-based quantities are standard complexity surrogates for deep nets (Neyshabur et al., 2015; Bartlett et al., 2017; Jiang et al., 2020). Let $\theta$ be the vector of all trainable parameters (including biases by default) and let $\|\theta\|_2$ be its $\ell_2$ norm. Let $S$ denote the chosen lower-tail margin statistic (default: $p10$), computed from logits. The reported value is

$$\text{l2.over.margin.p10} = \frac{\|\theta\|_2}{\max(|S|, \epsilon)}. \tag{12}$$

We computed this measure to create a combined scale+tail-margin proxy to test whether larger parameter scale relative to worst-case margins correlates with poorer generalization (Neyshabur et al., 2015; Jiang et al., 2018; Bartlett et al., 2017).

**l1.over.margin.p10.** Same implementation family as above, but using $\ell_1$ norm. $\ell_1$-type norms appear in norm-based capacity control and can behave differently under sparsity/scale effects (Neyshabur et al., 2015; 2014).

$$\text{l1.over.margin.p10} = \frac{\|\theta\|_1}{\max(|S|, \epsilon)}, \tag{13}$$

with the same $S$ definition (default statistic $p10$). We computed this measure to probe whether absolute-scale/sparsity-sensitive norms behave differently as generalization predictors under IID and shift (Jiang et al., 2020; Dziugaite et al., 2020).

**margin.normalized.param.norm.** This is the canonical implementation used by `l2.over.margin.p10` by default. With $\|\theta\|_2$ including biases and $S$ the selected margin statistic (default $p10$), we compute

$$\text{margin.normalized.param.norm} = \frac{\|\theta\|_2}{\max(|S|, \epsilon)}. \tag{14}$$

We computed this measure to create a single scalar that merges parameter scale and lower-tail margin magnitude, evaluated as a candidate predictor of IID/OOD generalization gaps across architectures (Neyshabur et al., 2015; Bartlett et al., 2017; Jiang et al., 2018).

**spectral.norm.per.layer.** For each layer weight tensor $W_\ell$, we estimate the spectral norm via power iteration on a matrix flattening of the weights. Spectral norms are central to spectrally-normalized margin bounds and Lipschitz-style capacity control for deep networks (Bartlett et al., 2017; Neyshabur et al., 2015). For convolutions, kernels are flattened (not the full convolution operator). Let $\hat{\sigma}_\ell$ denote the resulting estimate. We aggregate to a scalar mean:

$$\text{spectral.norm.per.layer} = \frac{1}{L} \sum_{\ell=1}^{L} \hat{\sigma}_\ell. \tag{15}$$

We computed this measure to create an operator-scale proxy (approximate Lipschitz/conditioning signal) to test correlation with IID/OOD generalization gaps (Bartlett et al., 2017; Jiang et al., 2020).

**spec.sum, spec.prod, and frobenius.distance.** The `spec.sum` and `spec.prod` measures use the same per-layer spectral-norm estimates as `spectral.norm.per.layer`, but aggregate them by summation or product, respectively. `frobenius.distance` measures the Frobenius norm of the displacement from initialization, $\|\theta - \theta_0\|_F$, when an initialization snapshot is available. These quantities are included because spectral products/sums and distance-from-initialization terms appear in norm-based generalization bounds and in empirical generalization-measure benchmarks (Bartlett et al., 2017; Jiang et al., 2020; Dziugaite et al., 2020).

**path.norm.** This measure is implementation-dependent by model family. Path- and norm-based controls are standard complexity surrogates and appear in multiple generalization-measure catalogs (Neyshabur et al., 2014; 2015; Jiang et al., 2020). For the convolutional models in this study, we use a structured layer-wise aggregation:

$$\text{path.norm} = \sum_{\ell=1}^{L} \log \|W_\ell\|_F. \tag{16}$$

We computed this measure to create a structured weight-aggregation complexity proxy, since exact path norms are intractable at scale, intended to correlate with generalization behavior (Neyshabur et al., 2014; Jiang et al., 2020; Dziugaite et al., 2020).

**fisher.rao.norm.** Let $p_\theta(y \mid x)$ be the predictive distribution and define $g_n = \nabla_\theta \log p_\theta(y_n \mid x_n)$. Fisher-information geometry motivates measuring parameter sensitivity through Fisher-metric–weighted quantities (Amari, 1998; Kawaguchi et al., 2022). The Fisher–Rao quantity is computed as

$$\text{fisher.rao.norm} = \sqrt{\mathbb{E}\left[\left(\theta^\top g\right)^2\right]} \approx \sqrt{\frac{1}{N} \sum_{n=1}^{N} \left(\theta^\top g_n\right)^2} = \sqrt{\theta^\top \widehat{F} \theta}, \tag{17}$$

where $\widehat{F} = \frac{1}{N} \sum_{n=1}^{N} g_n g_n^\top$. We computed this measure to create an information-geometric complexity measure that weights parameters by their sensitivity in log-likelihood space (Amari, 1998; Jiang et al., 2020).

### A.3 Sharpness-based Measures

**sharpness.** Sharpness is computed in a SAM-style one-step perturbation per minibatch, then averaged over the data loader. Sharpness and sharp minima have been linked to generalization behavior (e.g., large-batch effects) (Keskar et al., 2017), and SAM operationalizes a sharpness-aware objective (Foret et al., 2020). For a batch loss $\hat{L}_b(\theta)$, define the normalized gradient direction

$$u_b = \frac{\nabla_\theta \hat{L}_b(\theta)}{\left\|\nabla_\theta \hat{L}_b(\theta)\right\|_2}. \tag{18}$$

The one-step perturbed parameters are

$$\theta_b^+ = \theta + \rho\, u_b, \tag{19}$$

and we report the average batch-wise loss increase

$$\text{sharpness} = \mathbb{E}_b\left[\hat{L}_b(\theta_b^+) - \hat{L}_b(\theta)\right]. \tag{20}$$

We computed this measure to create a scalable flatness/sensitivity proxy (one-step) that remains informative across architectures and evaluation regimes (Foret et al., 2020; Cha et al., 2021; Izmailov et al., 2018).

**adaptive.sharpness.** Adaptive sharpness scans over a log-spaced set of radii $\{\rho_k\}_{k=1}^K$, computes the SAM-style one-step sharpness at each radius, and by default returns the maximal normalized value:

$$\text{adaptive.sharpness} = \max_{k \in \{1,\ldots,K\}} \frac{\text{sharpness}(\rho_k)}{\rho_k}. \tag{21}$$

We computed this measure to create a radius-robust sharpness summary that reduces sensitivity to a single perturbation scale (Foret et al., 2020; Jiang et al., 2020; Cha et al., 2021).

**sharpness.magnitude.** Let the baseline empirical loss be $\hat{L}(\theta) = \frac{1}{N}\sum_{n=1}^{N}\mathrm{CE}_n(\theta)$. Magnitude-aware perturbations are commonly used in empirical generalization-measure evaluations to improve comparability across scales (Jiang et al., 2020; Dziugaite et al., 2020). For each perturbation sample $k$, and for each parameter tensor $p$, we apply additive Gaussian noise scaled by the tensor standard deviation:

$$p \leftarrow p + \epsilon, \qquad \epsilon \sim \mathcal{N}\left(0, \, (r \cdot \mathrm{std}(p))^2\right). \tag{22}$$

Let $\Delta_k = \hat{L}(\theta + \Delta\theta_k) - \hat{L}(\theta)$ be the loss increase. We aggregate across samples (default: max; optionally mean):

$$\mathrm{sharpness\_raw} = \max_k \Delta_k. \tag{23}$$

Define the magnitude factor

$$M_{\mathrm{orig}} = \frac{\|\theta\|_2}{\sqrt{d}}, \tag{24}$$

where $d$ is the total number of parameters. The final measure is

$$\mathrm{sharpness.magnitude} = \mathrm{sharpness\_raw} \cdot M_{\mathrm{orig}}. \tag{25}$$

We computed this measure to create a scale-aware sharpness signal that probes sensitivity of the learned solution to stochastic perturbations (Keskar et al., 2017; Jiang et al., 2020; He et al., 2019).

**sharpness.magnitude.init.** This measure matches **sharpness.magnitude** but uses an initialization-relative magnitude factor when an init snapshot $\theta_0$ is available:

$$M_{\mathrm{init}} = \frac{\|\theta - \theta_0\|_2}{\sqrt{d}}, \tag{26}$$

falling back to $M_{\mathrm{orig}}$ if $\theta_0$ is unavailable. The final measure is

$$\mathrm{sharpness.magnitude.init} = \mathrm{sharpness\_raw} \cdot M_{\mathrm{init}}. \tag{27}$$

We computed this measure to capture sharpness weighted by distance traveled during training, reducing sensitivity to raw parameter scaling (Jiang et al., 2020; Dziugaite et al., 2020; Izmailov et al., 2018).

**sharpness.magflat.** With the same baseline loss $\hat{L}(\theta)$, we sample magnitude-aware perturbations

$$\Delta\theta_k = r \cdot z_k \odot (|\theta| + \epsilon_{\mathrm{scale}}), \qquad z_k \sim \mathcal{N}(0, I), \tag{28}$$

and compute loss increases $\Delta_k = \hat{L}(\theta + \Delta\theta_k) - \hat{L}(\theta)$. We aggregate across samples (default: max; optionally mean):

$$\mathrm{sharpness.magflat} = \max_k \Delta_k. \tag{29}$$

We computed this measure to create a magnitude-aware flatness proxy intended to improve comparability across architectures and training scales (Jiang et al., 2020; Dziugaite et al., 2020; Keskar et al., 2017).

**pac.bayes.magflat.** PAC-Bayes analysis provides nonvacuous generalization bounds for deep stochastic predictors under suitable choices of priors/posteriors (Dziugaite & Roy, 2017; Dziugaite et al., 2020). We define a diagonal Gaussian posterior centered at the learned parameters with magnitude-scaled variance:

$$Q = \mathcal{N}\left(\theta, \, \mathrm{diag}(\sigma_{q,i}^2)\right), \qquad \sigma_{q,i}^2 = (\sigma_{\mathrm{post}}|\theta_i|)^2, \tag{30}$$

and an isotropic Gaussian prior

$$P = \mathcal{N}(0, \sigma_{\mathrm{prior}}^2 I). \tag{31}$$

We approximate the Gibbs empirical 0–1 risk via multiplicative noise:

$$\theta' = \theta \odot (1 + \sigma_{\mathrm{post}}\varepsilon), \qquad \varepsilon \sim \mathcal{N}(0, I), \qquad \hat{R}_G = \mathbb{E}_{\theta'}[\text{empirical 0–1 error}]. \tag{32}$$

The KL divergence is

$$\mathrm{KL}(Q\|P) = \frac{1}{2}\sum_i\left(\frac{\theta_i^2}{\sigma_p^2} + \frac{\sigma_{q,i}^2}{\sigma_p^2} - 1 - \log\frac{\sigma_{q,i}^2}{\sigma_p^2}\right), \qquad \sigma_p^2 = \sigma_{\mathrm{prior}}^2. \tag{33}$$

Using a McAllester-style bound, we report

$$\mathrm{pac.bayes.magflat} = \hat{R}_G + \sqrt{\frac{\mathrm{KL}(Q\|P) + \log\left(\frac{2\sqrt{n}}{\delta}\right)}{2n}}, \tag{34}$$

where $n$ is the sample count used for the bound and $\delta$ is the confidence parameter. We computed this measure in our experiment to see the uncertainty and scale-aware bound, which is intended to remain informative across model families and evaluation regimes (Dziugaite & Roy, 2017; Jiang et al., 2020; Dziugaite et al., 2020).

**`pac.bayes.bound`, `pac.bayes.magnitude`, and `pac.bayes.magnitude.init`.** These measures use the same PAC-Bayes template as `pac.bayes.magflat`, but differ in how the posterior perturbation scale is parameterized. `pac.bayes.bound` uses the base stochastic-perturbation bound, `pac.bayes.magnitude` incorporates the learned parameter magnitude, and `pac.bayes.magnitude.init` incorporates distance from initialization. They are grouped with Sharpness-based measures because they evaluate sensitivity of the learned solution under stochastic perturbations and convert that sensitivity into a bound-like scalar.

**`flatness.proxy`.** We estimate the diagonal of the empirical Fisher for each parameter coordinate $i$ using per-batch gradients:

$$F_i \approx \frac{1}{B}\sum_{b=1}^{B}\left(\frac{\partial L_b}{\partial w_i}\right)^2. \tag{35}$$

With prior precision $\lambda$ (a fixed hyperparameter), define

$$\pi_i = F_i + \lambda. \tag{36}$$

We aggregate across coordinates using a configured statistic (mean/median/harmonic mean):

$$\mathrm{flatness.proxy} = \mathrm{agg}_i(\pi_i). \tag{37}$$

We computed this measure to create a lightweight precision proxy that approximates local sensitivity without explicit Hessian computation, consistent with flatness/curvature-focused generalization analyses (Keskar et al., 2017; Cha et al., 2021; Jiang et al., 2020).

**`hessian.top.eigenvalue`.** Using a single-batch loss $L(\theta)$, we estimate the largest Hessian eigenvalue via power iteration with Hessian–vector products:

$$v_{t+1} = \frac{Hv_t}{\|Hv_t\|_2}, \tag{38}$$

and report the Rayleigh quotient

$$\lambda_{\max} \approx v^\top H v. \tag{39}$$

We computed this measure to create a direct curvature indicator capturing the dominant local curvature direction, aligning with curvature-based accounts of sharp/flat minima (Keskar et al., 2017; He et al., 2019; Jiang et al., 2020).

**`hessian.trace`.** We estimate the Hessian trace using the Hutchinson estimator with Rademacher vectors $v_s$:

$$\mathrm{tr}(H) \approx \frac{1}{S}\sum_{s=1}^{S} v_s^\top H v_s. \tag{40}$$

We computed this measure to create a scalar summary of total curvature that complements $\lambda_{\max}$ and is commonly used in curvature/flatness studies of generalization (Keskar et al., 2017; He et al., 2019; Jiang et al., 2020).

### A.4 Optimization-based Measures

**gradient.noise.var and gradient.noise.final.var.** These measures estimate the variance of stochastic gradients across minibatches. `gradient.noise.var` summarizes gradient variability during training or over the configured measurement loader, while `gradient.noise.final.var` measures the same type of variability at the final trained solution. They are included as optimization-dynamics proxies because high gradient variability can reflect instability or sensitivity of the learned solution.

**gradient.noise.scale.** Let $g_b$ be the per-batch gradient vector and let $d$ denote the number of gradient coordinates. Gradient noise and batch-size effects are central to several empirical accounts of optimization dynamics and generalization (Shallue et al., 2019; Zhang et al., 2019). For each coordinate $i$,

$$\bar{g}_i = \frac{1}{B}\sum_{b=1}^{B} g_{b,i}, \qquad \mathrm{Var}_i = \frac{1}{B}\sum_{b=1}^{B}(g_{b,i} - \bar{g}_i)^2. \tag{41}$$

We compute the noise scale

$$\text{gradient.noise.scale} = \frac{1}{d}\sum_{i=1}^{d}\frac{\mathrm{Var}_i}{\bar{g}_i^2 + \epsilon}. \tag{42}$$

We computed this measure for our experiments to create a training-dynamics stability statistic that summarizes gradient stochasticity as a candidate predictor of generalization behavior (Zhang et al., 2019; Shallue et al., 2019; Keskar et al., 2017).

**gradient.norm.** For each batch, compute the gradient of the mean loss and its global norm

$$\|g\|_2 = \left(\sum_p \|g_p\|_2^2\right)^{1/2} \qquad (\text{or } \|g\|_1, \ \|g\|_\infty \text{ if configured}), \tag{43}$$

then aggregate across batches using a configured statistic (mean/max/std/median):

$$\text{gradient.norm} = \mathrm{agg}_b\big(\|g^{(b)}\|\big). \tag{44}$$

We computed this measure to create a compact summary of the gradient scale over evaluation batches, which was used to relate optimization sensitivity to IID/OOD generalization gaps (Jiang et al., 2020; Zhang et al., 2019; Dziugaite et al., 2020).

**input.gradient.norm.** This measure computes the norm of the loss gradient with respect to the input, $\|\nabla_x L(x, y; \theta)\|$, and aggregates it across batches. It is included as a sensitivity measure: larger input gradients indicate that small input perturbations can change the loss more strongly, which is directly relevant when comparing IID behavior with corruption- or perturbation-based shifted targets.

### A.5 Information Criteria

**aic.bias.term.** The Akaike Information Criterion (AIC) (Akaike, 1974) is derived from maximum likelihood estimation under standard regularity conditions. It is defined as:

$$\mathrm{AIC} = 2\,\mathrm{NLL}(\hat{\theta}) + 2k, \tag{45}$$

where $\mathrm{NLL}(\hat{\theta}) = -\log L(\hat{\theta})$ is the negative log-likelihood evaluated at the MLE and $k$ is the number of parameters. In our implementation, NLL is computed as the sum over all samples. The AIC bias term quantifies model complexity to mitigate overfitting and is given by $2k$. We take $k$ to be the total number of trainable parameters. Optionally, when the sample size $N$ is small relative to $k$, we use the finite-sample correction (AICc), in which case the bias term becomes

$$2k + \frac{2k(k+1)}{N - k - 1}, \tag{46}$$

valid when $N > k + 1$.

**tic.bias.term.** The Takeuchi Information Criterion (TIC) (Takeuchi, 1976; Naganuma et al., 2022) generalizes AIC by relaxing the realizability assumption. It remains valid under model misspecification (i.e., when the true distribution is not contained in the model family), provided standard regularity conditions hold. It is defined as

$$\text{TIC} = \text{NLL}_{\text{mean}}(\theta) + \text{Tr}\big(I(\theta)^{-1}J(\theta)\big), \tag{47}$$

where $I(\theta)$ is the Fisher information based on the Hessian of the mean NLL, and $J(\theta)$ is the empirical Fisher based on per-sample scores. When the model is correctly specified, $J \approx I$ and the trace term reduces to the parameter count.

The TIC bias term is $\text{Tr}(I(\theta)^{-1}J(\theta))$. Since forming and inverting the full Fisher information matrix is infeasible at scale, we approximate the trace via a diagonal sandwich approximation:

$$\text{bias}_{\text{TIC}} = \sum_j \frac{\text{diag}(J)_j}{\text{diag}(I)_j + \epsilon}, \tag{48}$$

where $\text{diag}(I)$ and $\text{diag}(J)$ denote the diagonals of (respectively) the Hessian of the mean NLL (estimated via Hutchinson's method) and the empirical Fisher (estimated from per-sample gradients).

For completeness, the same diagonal approximation also admits a deterministic upper bound using the inequality $\sum_j \frac{b_j}{a_j} \leq \frac{\sum_j b_j}{\min_j a_j}$ for positive vectors $a, b$:

$$\text{bias}_{\text{TIC}}^{\text{bound}} = \frac{\sum_j \text{diag}(J)_j}{\max(\min_j \text{diag}(I)_j, \ \epsilon)}. \tag{49}$$

This bound is not reported as a separate measure in Table 3; the reported Information Criteria entry is `tic.bias.term`.

**waic.bias.term.** The Widely Applicable Information Criterion (WAIC) (Watanabe & Opper, 2010) is Bayesian and grounded in singular learning theory. Unlike AIC and TIC, WAIC remains valid for singular models (e.g., neural networks and mixture models) where regularity conditions fail, and it does not require realizability. WAIC is computed from the posterior predictive distribution as

$$\text{WAIC} = -2\,\text{LPPD} + 2k_{\text{WAIC}}, \tag{50}$$

where LPPD is the log pointwise predictive density and $k_{\text{WAIC}}$ is the effective number of parameters, typically estimated via the variance of the log-likelihood under the posterior. In our runs, posterior samples are generated by MC Dropout when the dropout probability is greater than 0; when it is 0, we use weight noise.

The WAIC bias term is twice the effective number of parameters and is computed as

$$\text{bias}_{\text{WAIC}} = 2\sum_{n=1}^{N} \text{Var}_{\theta \sim p(\theta|D)}\big[\ln p(x_n \mid \theta)\big], \tag{51}$$

where $N$ is the number of samples and the variance is taken over posterior samples of $\theta$.

### A.6 Calibration & Confidence Measures

**ece.** Expected Calibration Error (ECE) (Naeini et al., 2015) partitions model predictions into equally spaced confidence bins. It calculates the absolute difference between the average confidence and the accuracy within each bin, taking a weighted average based on the number of samples in each bin. Let $y$ denote the ground-truth label and $\tilde{p}$ denote the predicted probability for a given sample, with $\tilde{y}$ representing the final predicted class. Let $N$ be the total number of samples in the dataset used to calculate ECE, and let $B_m$ be the set of samples falling into the $m$-th confidence bin. Then, ECE is calculated as follows:

$$\text{ECE} = \sum_{m=1}^{M} \frac{|B_m|}{N} |Acc(B_m) - Conf(B_m)| \tag{52}$$

$Acc(B_m)$ and $Conf(B_m)$ are defined as follows

$$Acc(B_m) = \frac{1}{|B_m|} \sum_{b \in B_m} \mathbf{1}\left(\tilde{y}_b = y_b\right), \ Conf(B_m) = \frac{1}{|B_m|} \sum_{b \in B_m} \tilde{p}_b$$

`reliability diagram.` In contrast to ECE, which computes a sample-weighted average across bins, the reliability-diagram score used here calculates an unweighted average of bin-wise calibration gaps.

$$\text{Reliability Diagram} = \frac{1}{M} \sum_{m=1}^{M} |Acc(B_m) - Conf(B_m)| \tag{53}$$

`mce.` Maximum Calibration Error (MCE) represents the maximum discrepancy between accuracy and confidence across all bins. Formally:

$$\text{MCE} = \max_{m} |Acc(B_m) - Conf(B_m)| \tag{54}$$

`ace.` It has been pointed out that predictions from recent large-scale models tend to be concentrated in high-confidence ranges (Nixon et al., 2019). Consequently, the equally spaced binning used in ECE results in low-confidence bins containing almost no samples due to this skew. To address this, Adaptive Calibration Error (ACE) (Nixon et al., 2019) dynamically handles confidence bias by partitioning bins to contain an equal number of samples, rather than using equal intervals. Let $K$ be the number of classes. For each class $k$, the prediction confidences are sorted and partitioned into $M$ subsets of equal size. Let $B_{m,k}$ denote the $m$-th bin for class $k$.

$$\text{ACE} = \frac{1}{KM} \sum_{k=1}^{K} \sum_{m=1}^{M} |Acc(B_{m,k}) - Conf(B_{m,k})| \tag{55}$$

`temperature.scaling.` We calculate the ECE after calibrating the model's predicted distribution using Temperature Scaling (Guo et al., 2017). Specifically, Temperature Scaling involves dividing the model's output logits $z$ by a scalar constant $T$. In this study, we determined the optimal $T$ by minimizing the cross-entropy loss on the training data using the L-BFGS optimizer, and subsequently computed the ECE.

## B  Experimental Setup Details

All experiments were conducted on a single NVIDIA H100 GPU.

### B.1  CIFAR-10

Details of the training hyperparameter sweep ranges are provided in Tables 4 and 5. Each primary CIFAR-10-suite run is trained for 100 epochs, consistent with the protocol described in Section 3. We specified different ranges for the learning rate, as the optimal values vary depending on the optimizer. CIFAR-10-C corruption severity is an evaluation setting rather than a training hyperparameter; for SimpleCNN, we evaluated severity levels 1–5. Additionally, we provided a separate table for NiN, as it is the only architecture that includes width and depth as hyperparameters.

Table 4: Hyperparameter search space for SimpleCNN and ResNetV2-32 on CIFAR-10.

| Hyperparameter | SGD | RMSProp & Adam |
|---|---|---|
| Learning rate | $\{3.2 \times 10^{-1}, 1 \times 10^{-1}, 3.2 \times 10^{-2}, 1 \times 10^{-2}, 3.2 \times 10^{-3}\}$ | $\{3.2 \times 10^{-3}, 1 \times 10^{-3}, 3.2 \times 10^{-4}, 1 \times 10^{-4}, 3.2 \times 10^{-5}\}$ |
| Batch size | $\{32, 64, 128, 256\}$ | |
| Dropout | $\{0, 0.5\}$ | |
| Weight decay | $\{0, 1, 5 \times 10^{-4}\}$ | |
| Random seed | $\{0, 1, 2, 3, 4\}$ | |

Table 5: Hyperparameter search space for NiN on CIFAR-10.

| Hyperparameter | SGD | RMSProp & Adam |
|---|---|---|
| Learning rate | $\{1 \times 10^{-1}, 3.2 \times 10^{-2}, 1 \times 10^{-2}\}$ | $\{1 \times 10^{-3}, 3.2 \times 10^{-4}, 1 \times 10^{-4}\}$ |
| Batch size | | $\{32, 64, 128\}$ |
| Dropout | | $\{0, 0.5\}$ |
| Depth | | $\{2, 8\}$ |
| Width | | $\{2, 8\}$ |
| Weight decay | | $\{0, 5 \times 10^{-4}\}$ |
| Random seed | | $\{0, 1, 2, 3, 4\}$ |

### B.1.1 CIFAR-10-P sequence aggregation

Generalization measures are computed only from source CIFAR-10 data. CIFAR-10-P frames are not used to compute ECE, MCE, margins, cross-entropy, or any other generalization measure. The dependence among perturbation frames therefore concerns the aggregation and uncertainty interpretation of the target metric, not measure-side aggregation.

The CIFAR-10-P loader represents each source image as one ordered perturbation sequence and returns one source label for that sequence. Each type–severity file is loaded as one dense array, so its sequence-length dimension is fixed across the full cell, not only within a minibatch. Sequence length can differ between perturbation types. The Gaussian-noise, severity-3 cell used in the primary correlation sweeps contains $N = 10{,}000$ sequences with $T = 31$ frames. A minibatch has shape $[B, T, C, H, W]$. The evaluator flattens only the first two dimensions for the model forward pass, repeats each source label $T$ times, and then reshapes the outputs to $[B, T, K]$. Thus, sequence boundaries are retained by the loader and restored before stability metrics are computed.

Top-1 accuracy, top-5 accuracy, and cross-entropy loss are summed over all frames in a cell and divided by the total number of frames. Because every sequence within a cell has the same length, this frame mean gives equal weight to each source image and is algebraically identical to first averaging within each sequence and then averaging the sequence means:

$$\text{Acc}_P = \frac{1}{NT} \sum_{i=1}^{N} \sum_{t=1}^{T} \mathbf{1}[\hat{y}_{it} = y_i] = \frac{1}{N} \sum_{i=1}^{N} \left\{ \frac{1}{T} \sum_{t=1}^{T} \mathbf{1}[\hat{y}_{it} = y_i] \right\}. \tag{56}$$

The same identity holds for cross-entropy after replacing the indicator by the per-frame loss. It depends on the common sequence length $T$; a pooled frame mean would not provide equal source-image weighting if sequence lengths differed.

Flip Probability and mT5D use the restored sequence tensor. Flip Probability compares top-1 predictions only at adjacent frames $(t, t+1)$ within the same source-image sequence. mT5D likewise computes its top-5 ranking distance only for adjacent frames within a sequence. Neither calculation crosses a sequence boundary. Their final values divide the accumulated statistic by the number of within-sequence adjacent pairs, $N(T-1)$ for one cell. These metrics are therefore explicitly sequence-aware and pair-weighted; with common $T$, pair weighting also gives equal weight to each source sequence.

For each evaluated cell, the implementation defines `OODGenGap_P` as source CIFAR-10 training accuracy in evaluation mode minus CIFAR-10-P top-1 accuracy, and `OODTestGap_P` as clean CIFAR-10 test accuracy minus CIFAR-10-P top-1 accuracy. The correlation-level, residual, and decision-level analyses read `final/OODGenGap_P`; all decision-level targets are minimized. When a run contains multiple available type–severity cells, the final summary key is the unweighted arithmetic mean of the cell-level metrics, rather than a frame-count-weighted pool.

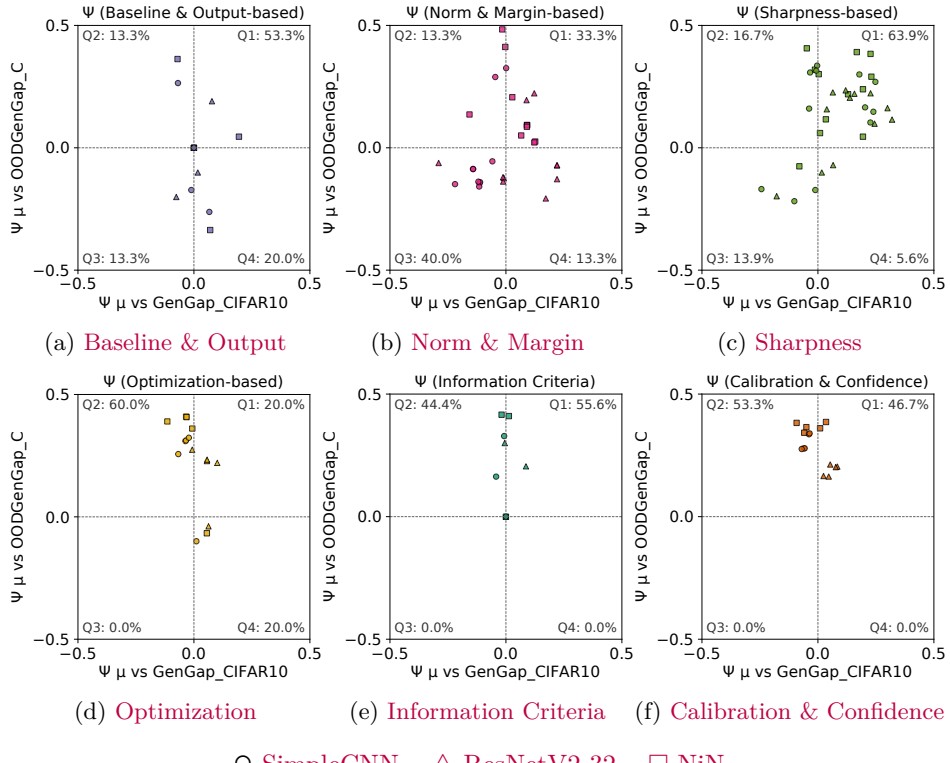

(a) Baseline & Output  (b) Norm & Margin  (c) Sharpness

(d) Optimization  (e) Information Criteria  (f) Calibration & Confidence

○ SimpleCNN  △ ResNetV2-32  □ NiN

Figure 3: CIFAR-10-C train-to-shifted-test split of Figure 1. Each point is a measure-by-architecture pair. The x-axis is Ψ for `GenGap_CIFAR10`, and the y-axis is Ψ for `OODGenGap_C`. The split view isolates corruption robustness from perturbation robustness and shows that several families retain positive shifted-gap association under corruptions.

## C Additional Results

This appendix reports target-specific correlations, local sign errors, decision-level diagnostics, and calibration residualization for the CIFAR-10 suite.

### C.1 CIFAR-10 Shifted-Gap Correlation Analyses

#### C.1.1 Train-to-shifted-test split by benchmark

Figures 3 and 4 split Figure 1 by target. Both show shifted-test-favorable mass for several families while exposing benchmark-specific dispersion.

#### C.1.2 Clean-to-shifted test degradation

Figure 5 reports the clean-to-shifted degradation scatter analysis used as a robustness check for the main train-to-shifted-test analysis in Figure 1. This target removes the training-accuracy term by comparing clean CIFAR-10 test performance directly against CIFAR-10-C/P test performance. The tables below provide the corresponding measure-level summaries for both train-to-shifted-test gaps and clean-to-shifted degradation targets.

#### C.1.3 Granulated-score uncertainty

Table 6 reports a descriptive uncertainty audit behind the family-level Ψ scatter plots in Figure 1. For each architecture and measure family, Ψ is the unweighted family mean of the measure-level "ALL" granulated scores. We report percentile bootstrap 95% intervals using measure-level resampling and subspace-level resampling

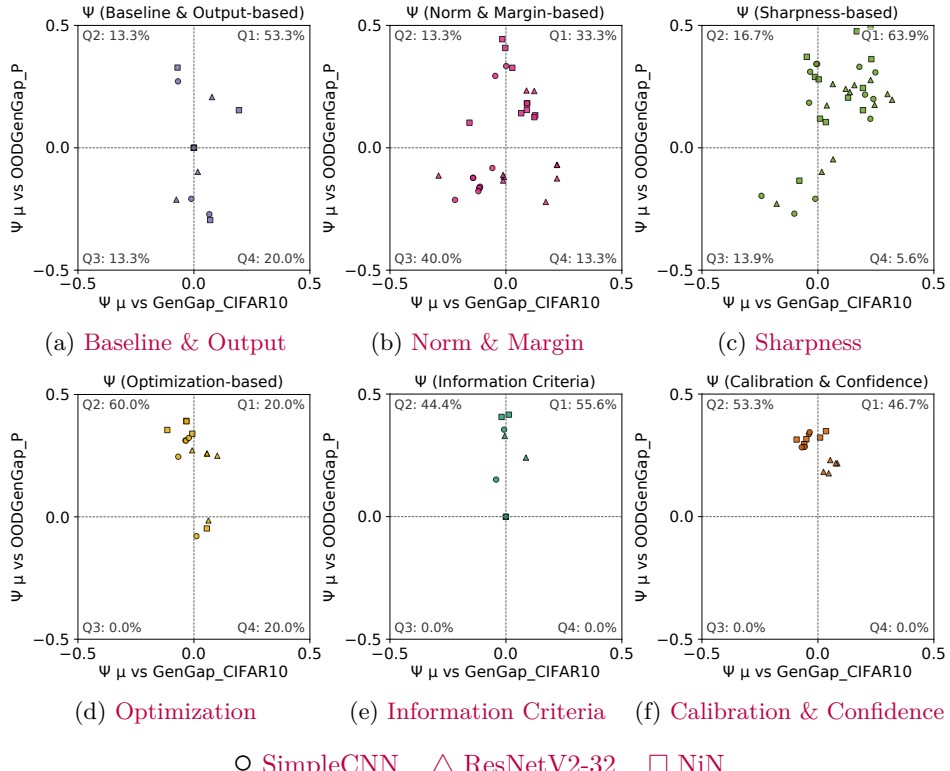

(a) Baseline & Output    (b) Norm & Margin    (c) Sharpness

(d) Optimization    (e) Information Criteria    (f) Calibration & Confidence

○ SimpleCNN    △ ResNetV2-32    □ NiN

Figure 4: CIFAR-10-P train-to-shifted-test split of Figure 1. Each point is a measure-by-architecture pair. The x-axis is Ψ for `GenGap_CIFAR10`, and the y-axis is Ψ for `OODGenGap_P`. The split view isolates perturbation robustness and complements the CIFAR-10-C corruption-only view in Figure 3.

within each selected measure, together with the valid-subspace count used in each family aggregate. These intervals should be interpreted as descriptive stability summaries rather than formal inferential confidence intervals, because different subspaces can share underlying trained runs and hyperparameter configurations and therefore are not guaranteed to be independent samples. The measure families and exclusions match the scatter-plot construction: raw `aic`, `tic`, and `waic` are omitted, while their bias-term variants are retained.

Table 6: Family-level granulated score Ψ with descriptive bootstrap 95% intervals and valid-subspace counts on the CIFAR-10 suite. The intervals summarize stability under measure- and subspace-level resampling, but should not be interpreted as formal inferential confidence intervals because subspaces may share trained runs and hyperparameter configurations. Valid subspaces are reported as total count across measures, followed by median per measure and range in parentheses.

| Model | Family | # Measures | Valid subspaces | IID Ψ [95% boot.] | OOD-C Ψ [95% boot.] | OOD-P Ψ [95% boot.] |
|---|---|---|---|---|---|---|
| SimpleCNN | Information Criteria | 3 | 1746 (582; 582–582) | -0.031 [-0.094, 0.013] | 0.026 [0.000, 0.056] | 0.082 [0.000, 0.173] |
| SimpleCNN | Calibration & Confidence | 5 | 2910 (582; 582–582) | -0.005 [-0.034, 0.026] | 0.044 [0.030, 0.059] | 0.153 [0.120, 0.186] |
| SimpleCNN | Baseline & Output-based | 5 | 2910 (582; 582–582) | 0.000 [-0.044, 0.044] | -0.000 [-0.005, 0.004] | -0.012 [-0.058, 0.035] |
| SimpleCNN | Norm & Margin-based | 10 | 5820 (582; 582–582) | -0.055 [-0.079, -0.028] | 0.002 [-0.014, 0.023] | -0.019 [-0.067, 0.040] |
| SimpleCNN | Sharpness-based | 12 | 6984 (582; 582–582) | 0.054 [-0.012, 0.119] | 0.034 [0.006, 0.058] | 0.110 [0.024, 0.187] |
| SimpleCNN | Optimization-based | 5 | 2910 (582; 582–582) | 0.006 [-0.016, 0.030] | 0.043 [0.032, 0.052] | 0.121 [0.058, 0.163] |
| ResNetV2-32 | Information Criteria | 3 | 1746 (582; 582–582) | 0.007 [-0.060, 0.078] | 0.062 [0.000, 0.112] | 0.103 [0.000, 0.172] |
| ResNetV2-32 | Calibration & Confidence | 5 | 2910 (582; 582–582) | 0.062 [0.033, 0.091] | 0.092 [0.043, 0.139] | 0.135 [0.092, 0.174] |
| ResNetV2-32 | Baseline & Output-based | 5 | 2910 (582; 582–582) | 0.008 [-0.021, 0.038] | 0.016 [-0.015, 0.053] | 0.014 [-0.015, 0.045] |
| ResNetV2-32 | Norm & Margin-based | 10 | 5281 (582; 43–582) | 0.070 [-0.022, 0.166] | 0.029 [-0.022, 0.073] | 0.032 [-0.011, 0.078] |
| ResNetV2-32 | Sharpness-based | 12 | 6982 (582; 581–582) | 0.126 [0.068, 0.180] | 0.101 [0.056, 0.143] | 0.142 [0.086, 0.194] |
| ResNetV2-32 | Optimization-based | 5 | 2910 (582; 582–582) | 0.064 [0.035, 0.094] | 0.108 [0.050, 0.151] | 0.151 [0.070, 0.204] |
| NiN | Information Criteria | 3 | 3888 (1296; 1296–1296) | -0.085 [-0.123, -0.049] | 0.133 [-0.171, 0.325] | 0.136 [-0.208, 0.352] |
| NiN | Calibration & Confidence | 5 | 6480 (1296; 1296–1296) | -0.016 [-0.099, 0.072] | 0.355 [0.317, 0.392] | 0.362 [0.323, 0.395] |
| NiN | Baseline & Output-based | 5 | 6480 (1296; 1296–1296) | -0.013 [-0.114, 0.091] | -0.113 [-0.231, 0.078] | -0.131 [-0.262, 0.085] |
| NiN | Norm & Margin-based | 10 | 12382 (1296; 718–1296) | -0.048 [-0.110, 0.028] | -0.037 [-0.142, 0.094] | -0.041 [-0.154, 0.097] |
| NiN | Sharpness-based | 12 | 15552 (1296; 1296–1296) | -0.016 [-0.081, 0.048] | 0.172 [0.041, 0.290] | 0.192 [0.035, 0.332] |
| NiN | Optimization-based | 5 | 6480 (1296; 1296–1296) | -0.086 [-0.128, -0.032] | 0.237 [0.049, 0.343] | 0.256 [0.047, 0.375] |

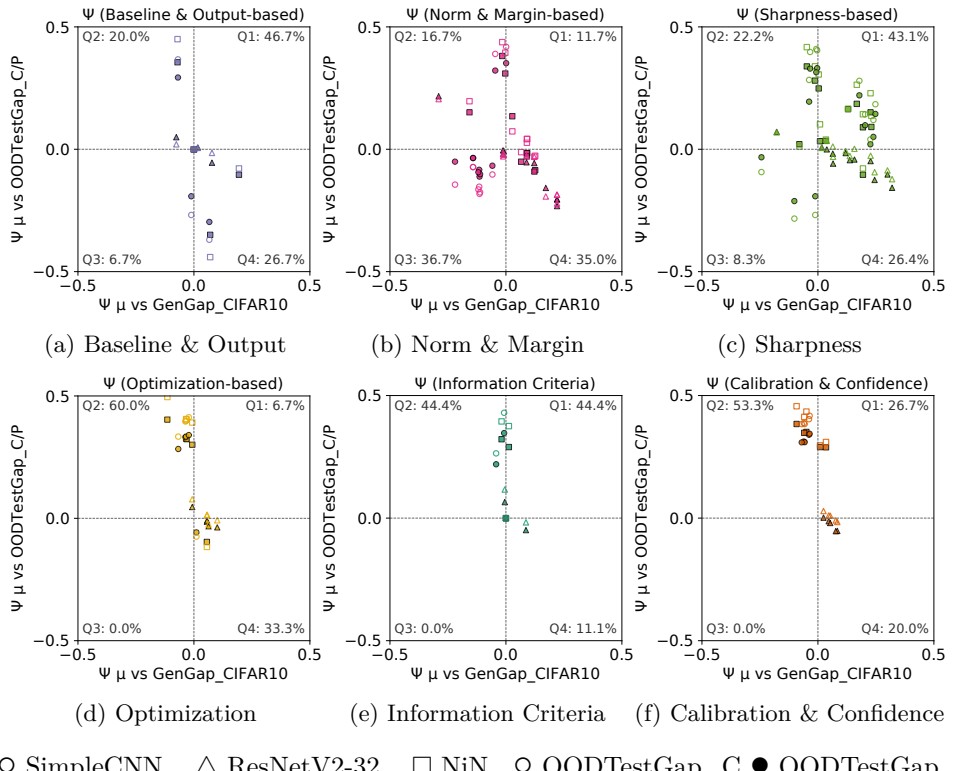

(a) Baseline & Output     (b) Norm & Margin     (c) Sharpness

(d) Optimization     (e) Information Criteria     (f) Calibration & Confidence

○ SimpleCNN    △ ResNetV2-32    □ NiN    ○ OODTestGap_C ● OODTestGap_P

Figure 5: CIFAR-10 suite clean-to-shifted degradation robustness check. Each panel compares IID generalization-gap sensitivity against the degradation from clean CIFAR-10 test accuracy to CIFAR-10-C/P test accuracy. The quadrant annotations show that the main CIFAR-10-C/P pattern from Figure 1 is attenuated but still visible for several Optimization-based measures, Information Criteria, and Calibration & Confidence measures.

### C.1.4 Measure-level $\Psi$ summaries

Tables 7–22 report the measure-level $\Psi$ summaries used in the scatter analyses above. The tables are grouped first by pooled architecture results and then by architecture-specific results for NiN, ResNetV2-32, and SimpleCNN.

Table 7: Average summary of IID correlations and OODGenGap_C OOD sensitivities across input directories.

| Measure | $lr^{IID}$ | $lr^{OOD}$ | $wd^{IID}$ | $wd^{OOD}$ | $\tau^{IID}$ | $\tau^{OOD}$ | $\Psi^{IID}$ | $\Psi^{OOD}$ |
|---|---|---|---|---|---|---|---|---|
| spec.prod | -0.1743 | -0.1827 | 0.2927 | -0.1783 | -0.1413 | -0.2420 | -0.0598 | -0.0772 |
| frobenius.distance | -0.3710 | -0.1168 | 0.1300 | -0.0169 | -0.3393 | -0.2375 | -0.1300 | 0.0013 |
| spec.sum | -0.2089 | -0.1573 | 0.2964 | -0.1832 | -0.1339 | -0.2087 | -0.0565 | -0.0793 |
| vcdim | 0.0005 | 0.0005 | 0.0006 | 0.0006 | -0.0337 | -0.1573 | -0.0168 | -0.0593 |
| pac.bayes.bound | -0.2112 | -0.1689 | 0.1902 | -0.2437 | -0.0789 | -0.1410 | -0.1503 | -0.1379 |
| pac.bayes.magnitude | -0.1135 | -0.0988 | 0.2947 | -0.1817 | -0.0402 | -0.1212 | 0.0303 | -0.0405 |
| magnitude | -0.1135 | -0.0988 | 0.2947 | -0.1817 | -0.0402 | -0.1212 | 0.0303 | -0.0405 |
| negative.entropy | -0.1802 | -0.1207 | 0.3254 | -0.1216 | 0.0503 | -0.1092 | 0.0900 | -0.0795 |
| spectral.norm.per.layer | -0.2090 | -0.1575 | 0.2930 | -0.1824 | -0.1111 | -0.1085 | -0.0301 | -0.0250 |
| pac.bayes.magnitude.init | -0.3319 | -0.0998 | 0.2730 | -0.1568 | -0.2925 | -0.0952 | -0.0415 | 0.0355 |
| aic.bias.term | 0.0000 | 0.0000 | 0.0000 | 0.0000 | -0.0044 | -0.0755 | -0.0170 | -0.0595 |
| params | 0.0000 | 0.0000 | 0.0000 | 0.0000 | -0.0044 | -0.0755 | -0.0170 | -0.0595 |
| l1.over.margin.p10 | 0.0776 | -0.0323 | 0.3251 | -0.1762 | -0.0012 | -0.0434 | 0.0043 | -0.0706 |
| gradient.noise.scale | 0.1085 | 0.0550 | 0.1000 | -0.0587 | 0.0838 | -0.0328 | 0.0389 | -0.0379 |
| l2.over.margin.p10 | 0.0359 | 0.0059 | 0.2949 | -0.1633 | -0.0708 | -0.0083 | 0.0007 | -0.0270 |
| margin.normalized.param.norm | 0.0359 | 0.0059 | 0.2949 | -0.1633 | -0.0708 | -0.0083 | 0.0007 | -0.0270 |
| pac.bayes.magflat | 0.0182 | 0.1013 | -0.2787 | 0.1987 | 0.0243 | 0.0126 | -0.0367 | -0.0154 |
| path.norm | 0.1671 | -0.1081 | 0.4915 | 0.1584 | 0.1126 | 0.0178 | 0.1379 | -0.0166 |
| cross.entropy | 0.1743 | 0.1173 | -0.3214 | 0.1131 | -0.0564 | 0.1042 | -0.0957 | 0.0769 |
| waic.bias.term | 0.2257 | 0.1462 | -0.1596 | 0.1306 | -0.0987 | 0.1146 | -0.0833 | 0.1259 |
| temperature.scaling | 0.2038 | 0.1681 | -0.2368 | 0.1545 | -0.0383 | 0.1266 | -0.0236 | 0.1487 |
| fisher.rao.norm | 0.2175 | 0.1639 | -0.1920 | 0.1257 | 0.0135 | 0.1388 | -0.0270 | 0.1162 |
| ace | 0.2562 | 0.1573 | -0.2454 | 0.1878 | -0.0102 | 0.1434 | -0.0254 | 0.1315 |
| ece | 0.2544 | 0.1595 | -0.2429 | 0.1917 | -0.0107 | 0.1436 | -0.0222 | 0.1352 |
| mce | 0.3258 | 0.2379 | -0.0996 | 0.1805 | 0.0636 | 0.1750 | 0.0709 | 0.2015 |
| gradient.norm | 0.2823 | 0.1810 | -0.2501 | 0.1911 | 0.0407 | 0.1782 | -0.0003 | 0.1661 |
| gradient.noise.var | 0.2904 | 0.1780 | -0.2268 | 0.1809 | 0.0412 | 0.1800 | -0.0176 | 0.1676 |
| reliability.diagram | 0.3413 | 0.2287 | -0.0953 | 0.1864 | 0.0684 | 0.1842 | 0.0697 | 0.2009 |
| flatness.proxy | 0.2878 | 0.1800 | -0.2362 | 0.1810 | 0.0570 | 0.1938 | -0.0115 | 0.1738 |
| gradient.noise.final.var | 0.2894 | 0.1779 | -0.2279 | 0.1767 | 0.0590 | 0.1956 | -0.0142 | 0.1750 |
| tic.bias.term | 0.3198 | 0.2049 | -0.2270 | 0.1924 | 0.0805 | 0.1988 | -0.0080 | 0.1558 |
| input.gradient.norm | 0.2565 | 0.2030 | -0.2509 | 0.1735 | 0.0437 | 0.2106 | -0.0319 | 0.1754 |
| hessian.top.eigenvalue | 0.3152 | 0.2282 | -0.2313 | 0.2008 | 0.1503 | 0.2448 | 0.0250 | 0.1710 |
| hessian.trace | 0.3168 | 0.2334 | -0.2448 | 0.1978 | 0.1491 | 0.2460 | 0.0338 | 0.1687 |
| inverse.margin.p10 | 0.3616 | 0.3138 | -0.2205 | 0.2166 | 0.1745 | 0.2600 | 0.0508 | 0.1895 |
| sharpness.magnitude | 0.3544 | 0.2580 | 0.2065 | -0.0033 | 0.2967 | 0.2744 | 0.1673 | 0.1603 |
| sharpness.magnitude.init | 0.3376 | 0.2834 | 0.1583 | 0.1036 | 0.2550 | 0.2834 | 0.1530 | 0.2084 |
| sharpness | 0.3301 | 0.2531 | -0.1329 | 0.2253 | 0.2527 | 0.2898 | 0.1250 | 0.1899 |
| sharpness.magflat | 0.3970 | 0.2748 | 0.1354 | 0.0283 | 0.3571 | 0.3310 | 0.1716 | 0.1734 |
| adaptive.sharpness | 0.3381 | 0.2824 | 0.0118 | 0.1357 | 0.3488 | 0.3505 | 0.1940 | 0.1429 |

Table 8: Average summary of IID correlations and OODGenGap_P OOD sensitivities across input directories.

| Measure | $lr^{IID}$ | $lr^{OOD}$ | $wd^{IID}$ | $wd^{OOD}$ | $\tau^{IID}$ | $\tau^{OOD}$ | $\Psi^{IID}$ | $\Psi^{OOD}$ |
|---|---|---|---|---|---|---|---|---|
| frobenius.distance | -0.3710 | -0.3217 | 0.1300 | -0.0602 | -0.3393 | -0.3842 | -0.1300 | -0.0555 |
| spec.prod | -0.1743 | -0.3157 | 0.2927 | -0.1869 | -0.1413 | -0.3338 | -0.0598 | -0.1156 |
| spec.sum | -0.2089 | -0.2918 | 0.2964 | -0.1859 | -0.1339 | -0.2940 | -0.0565 | -0.1032 |
| pac.bayes.magnitude.init | -0.3319 | -0.3116 | 0.2730 | -0.1690 | -0.2925 | -0.2422 | -0.0415 | -0.0227 |
| spectral.norm.per.layer | -0.2090 | -0.2920 | 0.2930 | -0.1883 | -0.1111 | -0.1905 | -0.0301 | -0.0428 |
| negative.entropy | -0.1802 | -0.2559 | 0.3254 | -0.1297 | 0.0503 | -0.1890 | 0.0900 | -0.1327 |
| magnitude | -0.1135 | -0.2131 | 0.2947 | -0.1873 | -0.0402 | -0.1847 | 0.0303 | -0.0720 |
| pac.bayes.magnitude | -0.1135 | -0.2131 | 0.2947 | -0.1873 | -0.0402 | -0.1847 | 0.0303 | -0.0720 |
| vcdim | 0.0005 | 0.0002 | 0.0006 | 0.0006 | -0.0337 | -0.1730 | -0.0168 | -0.0704 |
| pac.bayes.bound | -0.2112 | -0.1858 | 0.1902 | -0.2500 | -0.0789 | -0.1451 | -0.1503 | -0.1805 |
| aic.bias.term | 0.0000 | 0.0000 | 0.0000 | 0.0000 | -0.0044 | -0.0919 | -0.0170 | -0.0706 |
| params | 0.0000 | 0.0000 | 0.0000 | 0.0000 | -0.0044 | -0.0919 | -0.0170 | -0.0706 |
| path.norm | 0.1671 | -0.2180 | 0.4915 | 0.1049 | 0.1126 | -0.0584 | 0.1379 | -0.0581 |
| gradient.noise.scale | 0.1085 | 0.0848 | 0.1000 | -0.0574 | 0.0838 | -0.0368 | 0.0389 | -0.0532 |
| l1.over.margin.p10 | 0.0776 | 0.0216 | 0.3251 | -0.1591 | -0.0012 | -0.0095 | 0.0043 | -0.0700 |
| l2.over.margin.p10 | 0.0359 | 0.0354 | 0.2949 | -0.1278 | -0.0708 | 0.0173 | 0.0007 | -0.0238 |
| margin.normalized.param.norm | 0.0359 | 0.0354 | 0.2949 | -0.1278 | -0.0708 | 0.0173 | 0.0007 | -0.0238 |
| pac.bayes.magflat | 0.0182 | 0.1473 | -0.2787 | 0.1939 | 0.0243 | 0.0375 | -0.0367 | -0.0049 |
| waic.bias.term | 0.2257 | 0.2702 | -0.1596 | 0.1143 | -0.0987 | 0.1600 | -0.0833 | 0.1629 |
| cross.entropy | 0.1743 | 0.2494 | -0.3214 | 0.1227 | -0.0564 | 0.1821 | -0.0957 | 0.1309 |
| temperature.scaling | 0.2038 | 0.2836 | -0.2368 | 0.1621 | -0.0383 | 0.2011 | -0.0236 | 0.1958 |
| mce | 0.3258 | 0.3082 | -0.0996 | 0.1870 | 0.0636 | 0.2140 | 0.0709 | 0.2499 |
| fisher.rao.norm | 0.2175 | 0.2932 | -0.1920 | 0.1597 | 0.0135 | 0.2202 | -0.0270 | 0.1671 |
| ece | 0.2544 | 0.2824 | -0.2429 | 0.2084 | -0.0107 | 0.2228 | -0.0222 | 0.1931 |
| ace | 0.2562 | 0.2817 | -0.2454 | 0.2074 | -0.0102 | 0.2231 | -0.0254 | 0.1901 |
| reliability.diagram | 0.3413 | 0.2983 | -0.0953 | 0.1938 | 0.0684 | 0.2271 | 0.0697 | 0.2538 |
| inverse.margin.p10 | 0.3616 | 0.3923 | -0.2205 | 0.2245 | 0.1745 | 0.2779 | 0.0508 | 0.2325 |
| gradient.norm | 0.2823 | 0.3624 | -0.2501 | 0.2081 | 0.0407 | 0.2800 | -0.0003 | 0.2352 |
| gradient.noise.var | 0.2904 | 0.3602 | -0.2268 | 0.1983 | 0.0412 | 0.2801 | -0.0176 | 0.2320 |
| input.gradient.norm | 0.2565 | 0.3554 | -0.2509 | 0.1988 | 0.0437 | 0.2957 | -0.0319 | 0.2249 |
| flatness.proxy | 0.2878 | 0.3606 | -0.2362 | 0.1967 | 0.0570 | 0.2961 | -0.0115 | 0.2410 |
| gradient.noise.final.var | 0.2894 | 0.3603 | -0.2279 | 0.1967 | 0.0590 | 0.2983 | -0.0142 | 0.2413 |
| tic.bias.term | 0.3198 | 0.3885 | -0.2270 | 0.2116 | 0.0805 | 0.3053 | -0.0080 | 0.2286 |
| hessian.trace | 0.3168 | 0.4360 | -0.2448 | 0.2091 | 0.1491 | 0.3678 | 0.0338 | 0.2470 |
| hessian.top.eigenvalue | 0.3152 | 0.4245 | -0.2313 | 0.2086 | 0.1503 | 0.3693 | 0.0250 | 0.2406 |
| sharpness.magnitude.init | 0.3376 | 0.4064 | 0.1583 | 0.1581 | 0.2550 | 0.4044 | 0.1530 | 0.2918 |
| sharpness.magnitude | 0.3544 | 0.4128 | 0.2065 | 0.0448 | 0.2967 | 0.4169 | 0.1673 | 0.2411 |
| sharpness | 0.3301 | 0.4594 | -0.1329 | 0.2552 | 0.2527 | 0.4437 | 0.1250 | 0.2919 |
| sharpness.magflat | 0.3970 | 0.4433 | 0.1354 | 0.0694 | 0.3571 | 0.4836 | 0.1716 | 0.2659 |
| adaptive.sharpness | 0.3381 | 0.4759 | 0.0118 | 0.1603 | 0.3488 | 0.5025 | 0.1940 | 0.2367 |

Table 9: Average summary of IID correlations and OODTestGap_C OOD sensitivities across input directories.

| Measure | $lr^{IID}$ | $lr^{OOD}$ | $wd^{IID}$ | $wd^{OOD}$ | $\tau^{IID}$ | $\tau^{OOD}$ | $\Psi^{IID}$ | $\Psi^{OOD}$ |
|---|---|---|---|---|---|---|---|---|
| vcdim | 0.0005 | 0.0008 | 0.0006 | 0.0000 | -0.0337 | -0.1444 | -0.0168 | -0.0550 |
| pac.bayes.bound | -0.2112 | -0.0842 | 0.1902 | -0.2810 | -0.0789 | -0.1429 | -0.1503 | -0.0628 |
| pac.bayes.magflat | 0.0182 | -0.0630 | -0.2787 | 0.3093 | 0.0243 | -0.1250 | -0.0367 | -0.0291 |
| spec.prod | -0.1743 | 0.0282 | 0.2927 | -0.2947 | -0.1413 | -0.0922 | -0.0598 | -0.0153 |
| l1.over.margin.p10 | 0.0776 | -0.1048 | 0.3251 | -0.2857 | -0.0012 | -0.0839 | 0.0043 | -0.0745 |
| aic.bias.term | 0.0000 | 0.0000 | 0.0000 | 0.0000 | -0.0044 | -0.0779 | -0.0170 | -0.0552 |
| params | 0.0000 | 0.0000 | 0.0000 | 0.0000 | -0.0044 | -0.0779 | -0.0170 | -0.0552 |
| spec.sum | -0.2089 | 0.0617 | 0.2964 | -0.3020 | -0.1339 | -0.0640 | -0.0565 | -0.0274 |
| gradient.noise.scale | 0.1085 | 0.0029 | 0.1000 | -0.1083 | 0.0838 | -0.0624 | 0.0389 | -0.0542 |
| l2.over.margin.p10 | 0.0359 | -0.0891 | 0.2949 | -0.2615 | -0.0708 | -0.0379 | 0.0007 | -0.0402 |
| margin.normalized.param.norm | 0.0359 | -0.0891 | 0.2949 | -0.2615 | -0.0708 | -0.0379 | 0.0007 | -0.0402 |
| pac.bayes.magnitude | -0.1135 | 0.0660 | 0.2947 | -0.2941 | -0.0402 | -0.0266 | 0.0303 | -0.0593 |
| magnitude | -0.1135 | 0.0660 | 0.2947 | -0.2941 | -0.0402 | -0.0266 | 0.0303 | -0.0593 |
| cross.entropy | 0.1743 | -0.1410 | -0.3214 | 0.2090 | -0.0564 | -0.0085 | -0.0957 | 0.0844 |
| negative.entropy | -0.1802 | 0.1374 | 0.3254 | -0.2210 | 0.0503 | 0.0040 | 0.0900 | -0.0873 |
| fisher.rao.norm | 0.2175 | -0.1048 | -0.1920 | 0.1736 | 0.0135 | 0.0074 | -0.0270 | 0.0732 |
| temperature.scaling | 0.2038 | -0.0976 | -0.2368 | 0.2240 | -0.0383 | 0.0077 | -0.0236 | 0.1064 |
| path.norm | 0.1671 | -0.1464 | 0.4915 | 0.1135 | 0.1126 | 0.0135 | 0.1379 | -0.0481 |
| frobenius.distance | -0.3710 | 0.2114 | 0.1300 | -0.0473 | -0.3393 | 0.0214 | -0.1300 | 0.0715 |
| ace | 0.2562 | -0.1302 | -0.2454 | 0.2604 | -0.0102 | 0.0237 | -0.0254 | 0.1111 |
| ece | 0.2544 | -0.1277 | -0.2429 | 0.2613 | -0.0107 | 0.0237 | -0.0222 | 0.1110 |
| spectral.norm.per.layer | -0.2090 | 0.0611 | 0.2930 | -0.2990 | -0.1111 | 0.0244 | -0.0301 | 0.0315 |
| gradient.norm | 0.2823 | -0.1269 | -0.2501 | 0.2832 | 0.0407 | 0.0338 | -0.0003 | 0.1225 |
| gradient.noise.var | 0.2904 | -0.1301 | -0.2268 | 0.2610 | 0.0412 | 0.0358 | -0.0176 | 0.1269 |
| tic.bias.term | 0.3198 | -0.1050 | -0.2270 | 0.2739 | 0.0805 | 0.0459 | -0.0080 | 0.1112 |
| flatness.proxy | 0.2878 | -0.1243 | -0.2362 | 0.2645 | 0.0570 | 0.0465 | -0.0115 | 0.1361 |
| gradient.noise.final.var | 0.2894 | -0.1300 | -0.2279 | 0.2613 | 0.0590 | 0.0472 | -0.0142 | 0.1359 |
| mce | 0.3258 | -0.0496 | -0.0996 | 0.1980 | 0.0636 | 0.0511 | 0.0709 | 0.1156 |
| sharpness.magnitude | 0.3544 | -0.0620 | 0.2065 | -0.0957 | 0.2967 | 0.0538 | 0.1673 | 0.0499 |
| reliability.diagram | 0.3413 | -0.0579 | -0.0953 | 0.2057 | 0.0684 | 0.0617 | 0.0697 | 0.1230 |
| hessian.trace | 0.3168 | -0.0860 | -0.2448 | 0.2842 | 0.1491 | 0.0639 | 0.0338 | 0.1121 |
| hessian.top.eigenvalue | 0.3152 | -0.0873 | -0.2313 | 0.2774 | 0.1503 | 0.0658 | 0.0250 | 0.1111 |
| sharpness | 0.3301 | -0.0812 | -0.1329 | 0.2503 | 0.2527 | 0.0662 | 0.1250 | 0.0924 |
| sharpness.magflat | 0.3970 | -0.0606 | 0.1354 | -0.0237 | 0.3571 | 0.0750 | 0.1716 | 0.0600 |
| waic.bias.term | 0.2257 | -0.1544 | -0.1596 | 0.1687 | -0.0987 | 0.0797 | -0.0833 | 0.1081 |
| input.gradient.norm | 0.2565 | -0.0821 | -0.2509 | 0.2535 | 0.0437 | 0.0842 | -0.0319 | 0.1503 |
| sharpness.magnitude.init | 0.3376 | -0.0147 | 0.1583 | 0.0047 | 0.2550 | 0.0896 | 0.1530 | 0.0896 |
| inverse.margin.p10 | 0.3616 | 0.0322 | -0.2205 | 0.2615 | 0.1745 | 0.0963 | 0.0508 | 0.1009 |
| adaptive.sharpness | 0.3381 | -0.0352 | 0.0118 | 0.1691 | 0.3488 | 0.1009 | 0.1940 | 0.0551 |
| pac.bayes.magnitude.init | -0.3319 | 0.2310 | 0.2730 | -0.2836 | -0.2925 | 0.1536 | -0.0415 | 0.0882 |

Table 10: Average summary of IID correlations and OODTestGap_P OOD sensitivities across input directories.

| Measure | $lr^{IID}$ | $lr^{OOD}$ | $wd^{IID}$ | $wd^{OOD}$ | $\tau^{IID}$ | $\tau^{OOD}$ | $\Psi^{IID}$ | $\Psi^{OOD}$ |
|---|---|---|---|---|---|---|---|---|
| spec.prod | -0.1743 | -0.1036 | 0.2927 | -0.3644 | -0.1413 | -0.1949 | -0.0598 | -0.0543 |
| pac.bayes.bound | -0.2112 | -0.1146 | 0.1902 | -0.3377 | -0.0789 | -0.1603 | -0.1503 | -0.0795 |
| vcdim | 0.0005 | 0.0008 | 0.0006 | 0.0000 | -0.0337 | -0.1599 | -0.0168 | -0.0675 |
| spec.sum | -0.2089 | -0.0693 | 0.2964 | -0.3686 | -0.1339 | -0.1523 | -0.0565 | -0.0736 |
| magnitude | -0.1135 | -0.0703 | 0.2947 | -0.3637 | -0.0402 | -0.0940 | 0.0303 | -0.1169 |
| pac.bayes.magnitude | -0.1135 | -0.0703 | 0.2947 | -0.3637 | -0.0402 | -0.0940 | 0.0303 | -0.1169 |
| aic.bias.term | 0.0000 | 0.0000 | 0.0000 | 0.0000 | -0.0044 | -0.0924 | -0.0170 | -0.0677 |
| params | 0.0000 | 0.0000 | 0.0000 | 0.0000 | -0.0044 | -0.0924 | -0.0170 | -0.0677 |
| frobenius.distance | -0.3710 | 0.0771 | 0.1300 | -0.1295 | -0.3393 | -0.0867 | -0.1300 | 0.0244 |
| pac.bayes.magflat | 0.0182 | 0.0299 | -0.2787 | 0.3739 | 0.0243 | -0.0856 | -0.0367 | -0.0013 |
| path.norm | 0.1671 | -0.3101 | 0.4915 | 0.0274 | 0.1126 | -0.0807 | 0.1379 | -0.0862 |
| l1.over.margin.p10 | 0.0776 | -0.1182 | 0.3251 | -0.3335 | -0.0012 | -0.0782 | 0.0043 | -0.0698 |
| gradient.noise.scale | 0.1085 | 0.0290 | 0.1000 | -0.1152 | 0.0838 | -0.0639 | 0.0389 | -0.0808 |
| negative.entropy | -0.1802 | 0.0116 | 0.3254 | -0.3359 | 0.0503 | -0.0635 | 0.0900 | -0.1763 |
| spectral.norm.per.layer | -0.2090 | -0.0701 | 0.2930 | -0.3655 | -0.1111 | -0.0594 | -0.0301 | -0.0118 |
| l2.over.margin.p10 | 0.0359 | -0.0976 | 0.2949 | -0.2792 | -0.0708 | -0.0379 | 0.0007 | -0.0253 |
| margin.normalized.param.norm | 0.0359 | -0.0976 | 0.2949 | -0.2792 | -0.0708 | -0.0379 | 0.0007 | -0.0253 |
| pac.bayes.magnitude.init | -0.3319 | 0.0766 | 0.2730 | -0.3439 | -0.2925 | 0.0478 | -0.0415 | 0.0532 |
| cross.entropy | 0.1743 | -0.0231 | -0.3214 | 0.3250 | -0.0564 | 0.0558 | -0.0957 | 0.1749 |
| temperature.scaling | 0.2038 | -0.0075 | -0.2368 | 0.3167 | -0.0383 | 0.0660 | -0.0236 | 0.1844 |
| fisher.rao.norm | 0.2175 | 0.0039 | -0.1920 | 0.2771 | 0.0135 | 0.0694 | -0.0270 | 0.1531 |
| mce | 0.3258 | -0.0217 | -0.0996 | 0.2568 | 0.0636 | 0.0729 | 0.0709 | 0.1599 |
| reliability.diagram | 0.3413 | -0.0322 | -0.0953 | 0.2539 | 0.0684 | 0.0853 | 0.0697 | 0.1621 |
| ece | 0.2544 | -0.0155 | -0.2429 | 0.3764 | -0.0107 | 0.0899 | -0.0222 | 0.1958 |
| ace | 0.2562 | -0.0177 | -0.2454 | 0.3722 | -0.0102 | 0.0901 | -0.0254 | 0.1943 |
| waic.bias.term | 0.2257 | -0.0511 | -0.1596 | 0.2178 | -0.0987 | 0.1214 | -0.0833 | 0.1952 |
| gradient.noise.var | 0.2904 | 0.0195 | -0.2268 | 0.3456 | 0.0412 | 0.1215 | -0.0176 | 0.2088 |
| gradient.norm | 0.2823 | 0.0269 | -0.2501 | 0.3775 | 0.0407 | 0.1217 | -0.0003 | 0.2086 |
| tic.bias.term | 0.3198 | 0.0472 | -0.2270 | 0.3607 | 0.0805 | 0.1335 | -0.0080 | 0.1995 |
| flatness.proxy | 0.2878 | 0.0256 | -0.2362 | 0.3496 | 0.0570 | 0.1348 | -0.0115 | 0.2208 |
| gradient.noise.final.var | 0.2894 | 0.0200 | -0.2279 | 0.3435 | 0.0590 | 0.1352 | -0.0142 | 0.2176 |
| inverse.margin.p10 | 0.3616 | 0.1211 | -0.2205 | 0.3502 | 0.1745 | 0.1414 | 0.0508 | 0.1622 |
| input.gradient.norm | 0.2565 | 0.0518 | -0.2509 | 0.3676 | 0.0437 | 0.1587 | -0.0319 | 0.2211 |
| sharpness.magnitude | 0.3544 | 0.0546 | 0.2065 | -0.0945 | 0.2967 | 0.1648 | 0.1673 | 0.0811 |
| hessian.trace | 0.3168 | 0.0816 | -0.2448 | 0.3633 | 0.1491 | 0.1694 | 0.0338 | 0.1888 |
| hessian.top.eigenvalue | 0.3152 | 0.0836 | -0.2313 | 0.3516 | 0.1503 | 0.1746 | 0.0250 | 0.1883 |
| sharpness.magflat | 0.3970 | 0.0563 | 0.1354 | -0.0300 | 0.3571 | 0.1917 | 0.1716 | 0.1076 |
| sharpness.magnitude.init | 0.3376 | 0.0703 | 0.1583 | 0.0188 | 0.2550 | 0.1918 | 0.1530 | 0.1271 |
| sharpness | 0.3301 | 0.0915 | -0.1329 | 0.3274 | 0.2527 | 0.2003 | 0.1250 | 0.1566 |
| adaptive.sharpness | 0.3381 | 0.1214 | 0.0118 | 0.2032 | 0.3488 | 0.2258 | 0.1940 | 0.0837 |

Table 11: nin summary of IID correlations and OODGenGap_C OOD sensitivities for each measure.

| Measure | $lr^{IID}$ | $lr^{OOD}$ | $wd^{IID}$ | $wd^{OOD}$ | $\tau^{IID}$ | $\tau^{OOD}$ | $\Psi^{IID}$ | $\Psi^{OOD}$ |
|---|---|---|---|---|---|---|---|---|
| vcdim | 0.0014 | 0.0014 | 0.0019 | 0.0019 | -0.1010 | -0.4718 | -0.0503 | -0.1780 |
| frobenius.distance | -0.0917 | -0.2671 | 0.0704 | 0.0111 | -0.1200 | -0.4046 | -0.1719 | -0.1150 |
| spec.sum | 0.0963 | -0.2699 | 0.2778 | -0.3148 | -0.0134 | -0.3476 | -0.0625 | -0.2117 |
| spec.prod | 0.1185 | -0.2532 | 0.2741 | -0.2963 | -0.0179 | -0.3453 | -0.0441 | -0.1719 |
| pac.bayes.bound | -0.2269 | -0.3542 | 0.1796 | -0.3852 | -0.0957 | -0.2616 | -0.2208 | -0.2615 |
| gradient.noise.scale | 0.0898 | 0.0125 | 0.1259 | -0.1389 | 0.0245 | -0.2422 | 0.0172 | -0.1308 |
| aic.bias.term | 0.0000 | 0.0000 | 0.0000 | 0.0000 | -0.0133 | -0.2265 | -0.0510 | -0.1785 |
| params | 0.0000 | 0.0000 | 0.0000 | 0.0000 | -0.0133 | -0.2265 | -0.0510 | -0.1785 |
| pac.bayes.magflat | -0.1213 | 0.2051 | -0.2889 | 0.3241 | -0.0509 | -0.2066 | -0.1421 | -0.1162 |
| magnitude | 0.1389 | -0.2097 | 0.2981 | -0.3148 | 0.0665 | -0.2047 | 0.0409 | -0.1990 |
| pac.bayes.magnitude | 0.1389 | -0.2097 | 0.2981 | -0.3148 | 0.0665 | -0.2047 | 0.0409 | -0.1990 |
| l1.over.margin.p10 | 0.0500 | -0.1551 | 0.3463 | -0.2796 | -0.3049 | -0.1943 | -0.1466 | -0.1817 |
| negative.entropy | 0.0824 | -0.1218 | 0.3444 | -0.1500 | 0.3927 | -0.1371 | 0.1724 | -0.2641 |
| l2.over.margin.p10 | 0.1130 | -0.0995 | 0.3352 | -0.2870 | -0.3264 | -0.0809 | -0.1077 | -0.0795 |
| margin.normalized.param.norm | 0.1130 | -0.0995 | 0.3352 | -0.2870 | -0.3264 | -0.0809 | -0.1077 | -0.0795 |
| spectral.norm.per.layer | 0.0954 | -0.2708 | 0.2722 | -0.3056 | 0.0546 | -0.0471 | 0.0133 | -0.0507 |
| pac.bayes.magnitude.init | -0.0120 | -0.2338 | 0.2667 | -0.3000 | -0.0161 | 0.0066 | 0.0041 | 0.0482 |
| fisher.rao.norm | -0.0657 | 0.1218 | -0.1759 | 0.1500 | -0.3872 | 0.0650 | -0.1345 | 0.2130 |
| waic.bias.term | 0.0398 | 0.1644 | -0.0907 | 0.1704 | -0.4471 | 0.1091 | -0.0918 | 0.2527 |
| cross.entropy | -0.0713 | 0.1181 | -0.3370 | 0.1556 | -0.4115 | 0.1241 | -0.1786 | 0.2556 |
| temperature.scaling | 0.0412 | 0.2389 | -0.2352 | 0.2204 | -0.3946 | 0.1460 | -0.1250 | 0.3033 |
| path.norm | 0.2694 | -0.2049 | 0.2611 | -0.4963 | 0.1767 | 0.1518 | 0.1889 | -0.1252 |
| gradient.noise.var | 0.0380 | 0.2347 | -0.1981 | 0.2852 | -0.3597 | 0.1651 | -0.1189 | 0.3163 |
| tic.bias.term | 0.0389 | 0.2505 | -0.1981 | 0.3000 | -0.3530 | 0.1679 | -0.1111 | 0.3254 |
| ece | 0.1037 | 0.1884 | -0.2083 | 0.2787 | -0.3472 | 0.1698 | -0.0782 | 0.3380 |
| ace | 0.1106 | 0.1833 | -0.2148 | 0.2759 | -0.3430 | 0.1710 | -0.0811 | 0.3343 |
| gradient.norm | 0.0176 | 0.2477 | -0.2222 | 0.2963 | -0.3498 | 0.1717 | -0.1019 | 0.3301 |
| sharpness.magnitude | 0.2352 | 0.2588 | 0.3106 | -0.0218 | 0.0055 | 0.1730 | 0.1052 | 0.2972 |
| sharpness.magnitude.init | 0.2833 | 0.3042 | 0.2056 | 0.1870 | -0.0352 | 0.1734 | 0.0885 | 0.3425 |
| mce | 0.2968 | 0.3074 | -0.0370 | 0.3111 | -0.1667 | 0.1909 | 0.0917 | 0.3888 |
| input.gradient.norm | -0.0269 | 0.1699 | -0.2556 | 0.1870 | -0.3227 | 0.2045 | -0.1214 | 0.3227 |
| flatness.proxy | 0.0296 | 0.2394 | -0.2222 | 0.2944 | -0.3110 | 0.2067 | -0.1083 | 0.3417 |
| gradient.noise.final.var | 0.0352 | 0.2338 | -0.1981 | 0.2852 | -0.3060 | 0.2119 | -0.1028 | 0.3445 |
| reliability.diagram | 0.3370 | 0.2894 | -0.0278 | 0.3167 | -0.1612 | 0.2230 | 0.1114 | 0.4098 |
| hessian.top.eigenvalue | 0.0287 | 0.3088 | -0.2167 | 0.2981 | -0.2270 | 0.2348 | -0.0927 | 0.3356 |
| inverse.margin.p10 | 0.2269 | 0.3162 | -0.1630 | 0.3204 | 0.0791 | 0.2511 | 0.0943 | 0.4359 |
| sharpness | 0.0009 | 0.3366 | -0.0500 | 0.3278 | -0.1193 | 0.2523 | -0.0401 | 0.3062 |
| hessian.trace | 0.0204 | 0.3208 | -0.2389 | 0.3056 | -0.2230 | 0.2591 | -0.0843 | 0.3530 |
| sharpness.magflat | 0.2046 | 0.2412 | 0.1722 | 0.0333 | 0.0594 | 0.3165 | 0.1473 | 0.3105 |
| adaptive.sharpness | -0.0389 | 0.3208 | 0.0000 | 0.1889 | 0.0758 | 0.4181 | 0.1138 | 0.3042 |

Table 12: nin summary of IID correlations and OODGenGap_P OOD sensitivities for each measure.

| Measure | $lr^{IID}$ | $lr^{OOD}$ | $wd^{IID}$ | $wd^{OOD}$ | $\tau^{IID}$ | $\tau^{OOD}$ | $\Psi^{IID}$ | $\Psi^{OOD}$ |
|---|---|---|---|---|---|---|---|---|
| vcdim | 0.0014 | 0.0005 | 0.0019 | 0.0019 | -0.1010 | -0.5189 | -0.0503 | -0.2112 |
| frobenius.distance | -0.0917 | -0.3417 | 0.0704 | 0.0352 | -0.1200 | -0.4658 | -0.1719 | -0.1796 |
| spec.prod | 0.1185 | -0.2296 | 0.2741 | -0.2167 | -0.0179 | -0.3720 | -0.0441 | -0.2000 |
| spec.sum | 0.0963 | -0.2481 | 0.2778 | -0.2241 | -0.0134 | -0.3700 | -0.0625 | -0.2301 |
| pac.bayes.bound | -0.2269 | -0.2954 | 0.1796 | -0.2981 | -0.0957 | -0.3125 | -0.2208 | -0.2949 |
| gradient.noise.scale | 0.0898 | 0.0065 | 0.1259 | -0.1148 | 0.0245 | -0.2785 | 0.0172 | -0.1545 |
| aic.bias.term | 0.0000 | 0.0000 | 0.0000 | 0.0000 | -0.0133 | -0.2758 | -0.0510 | -0.2118 |
| params | 0.0000 | 0.0000 | 0.0000 | 0.0000 | -0.0133 | -0.2758 | -0.0510 | -0.2118 |
| pac.bayes.magflat | -0.1213 | 0.2065 | -0.2889 | 0.2222 | -0.0509 | -0.2488 | -0.1421 | -0.1558 |
| magnitude | 0.1389 | -0.2167 | 0.2981 | -0.2130 | 0.0665 | -0.2273 | 0.0409 | -0.2207 |
| pac.bayes.magnitude | 0.1389 | -0.2167 | 0.2981 | -0.2130 | 0.0665 | -0.2273 | 0.0409 | -0.2207 |
| l1.over.margin.p10 | 0.0500 | -0.1056 | 0.3463 | -0.1667 | -0.3049 | -0.2182 | -0.1466 | -0.1751 |
| negative.entropy | 0.0824 | -0.1991 | 0.3444 | -0.1000 | 0.3927 | -0.1618 | 0.1724 | -0.2980 |
| l2.over.margin.p10 | 0.1130 | -0.0333 | 0.3352 | -0.1741 | -0.3264 | -0.0772 | -0.1077 | -0.0623 |
| margin.normalized.param.norm | 0.1130 | -0.0333 | 0.3352 | -0.1741 | -0.3264 | -0.0772 | -0.1077 | -0.0623 |
| spectral.norm.per.layer | 0.0954 | -0.2491 | 0.2722 | -0.2185 | 0.0546 | -0.0598 | 0.0133 | -0.0463 |
| pac.bayes.magnitude.init | -0.0120 | -0.3213 | 0.2667 | -0.2241 | -0.0161 | -0.0291 | 0.0041 | 0.0106 |
| fisher.rao.norm | -0.0657 | 0.1935 | -0.1759 | 0.1444 | -0.3872 | 0.0840 | -0.1345 | 0.2260 |
| waic.bias.term | 0.0398 | 0.2398 | -0.0907 | 0.1500 | -0.4471 | 0.1077 | -0.0918 | 0.2606 |
| path.norm | 0.2694 | -0.1847 | 0.2611 | -0.5111 | 0.1767 | 0.1095 | 0.1889 | -0.1365 |
| cross.entropy | -0.0713 | 0.1991 | -0.3370 | 0.1093 | -0.4115 | 0.1467 | -0.1786 | 0.2881 |
| temperature.scaling | 0.0412 | 0.2903 | -0.2352 | 0.1556 | -0.3946 | 0.1593 | -0.1250 | 0.2960 |
| mce | 0.2968 | 0.2903 | -0.0370 | 0.2481 | -0.1667 | 0.1740 | 0.0917 | 0.3723 |
| gradient.noise.var | 0.0380 | 0.3269 | -0.1981 | 0.2278 | -0.3597 | 0.1881 | -0.1189 | 0.3460 |
| ece | 0.1037 | 0.2324 | -0.2083 | 0.2398 | -0.3472 | 0.1912 | -0.0782 | 0.3684 |
| tic.bias.term | 0.0389 | 0.3389 | -0.1981 | 0.2389 | -0.3530 | 0.1926 | -0.1111 | 0.3587 |
| ace | 0.1106 | 0.2301 | -0.2148 | 0.2370 | -0.3430 | 0.1937 | -0.0811 | 0.3662 |
| gradient.norm | 0.0176 | 0.3324 | -0.2222 | 0.2352 | -0.3498 | 0.2004 | -0.1019 | 0.3644 |
| reliability.diagram | 0.3370 | 0.2630 | -0.0278 | 0.2648 | -0.1612 | 0.2118 | 0.1114 | 0.4082 |
| input.gradient.norm | -0.0269 | 0.2417 | -0.2556 | 0.1556 | -0.3227 | 0.2258 | -0.1214 | 0.3451 |
| flatness.proxy | 0.0296 | 0.3259 | -0.2222 | 0.2259 | -0.3110 | 0.2368 | -0.1083 | 0.3722 |
| inverse.margin.p10 | 0.2269 | 0.3528 | -0.1630 | 0.2593 | 0.0791 | 0.2381 | 0.0943 | 0.4540 |
| sharpness.magnitude.init | 0.2833 | 0.3833 | 0.2056 | 0.2815 | -0.0352 | 0.2408 | 0.0885 | 0.4078 |
| gradient.noise.final.var | 0.0352 | 0.3259 | -0.1981 | 0.2315 | -0.3060 | 0.2424 | -0.1028 | 0.3779 |
| sharpness.magnitude | 0.2352 | 0.3481 | 0.3106 | 0.0949 | 0.0055 | 0.2450 | 0.1052 | 0.3669 |
| hessian.top.eigenvalue | 0.0287 | 0.3917 | -0.2167 | 0.2370 | -0.2270 | 0.2728 | -0.0927 | 0.3608 |
| hessian.trace | 0.0204 | 0.4130 | -0.2389 | 0.2333 | -0.2230 | 0.2961 | -0.0843 | 0.3888 |
| sharpness | 0.0009 | 0.4009 | -0.0500 | 0.3037 | -0.1193 | 0.2994 | -0.0401 | 0.3482 |
| sharpness.magflat | 0.2046 | 0.3269 | 0.1722 | 0.0611 | 0.0594 | 0.3837 | 0.1473 | 0.3669 |
| adaptive.sharpness | -0.0389 | 0.3630 | 0.0000 | 0.1537 | 0.0758 | 0.4604 | 0.1138 | 0.3523 |

Table 13: nin summary of IID correlations and OODTestGap_C OOD sensitivities for each measure.

| Measure | $lr^{IID}$ | $lr^{OOD}$ | $wd^{IID}$ | $wd^{OOD}$ | $\tau^{IID}$ | $\tau^{OOD}$ | $\Psi^{IID}$ | $\Psi^{OOD}$ |
|---|---|---|---|---|---|---|---|---|
| vcdim | 0.0014 | 0.0023 | 0.0019 | 0.0000 | -0.1010 | -0.4333 | -0.0503 | -0.1650 |
| frobenius.distance | -0.0917 | -0.1023 | 0.0704 | -0.0120 | -0.1200 | -0.3678 | -0.1719 | -0.0506 |
| spec.sum | 0.0963 | -0.2199 | 0.2778 | -0.5546 | -0.0134 | -0.3274 | -0.0625 | -0.2132 |
| negative.entropy | 0.0824 | -0.0699 | 0.3444 | -0.3083 | 0.3927 | -0.3204 | 0.1724 | -0.3433 |
| spec.prod | 0.1185 | -0.2051 | 0.2741 | -0.5398 | -0.0179 | -0.3027 | -0.0441 | -0.1572 |
| gradient.noise.scale | 0.0898 | -0.0653 | 0.1259 | -0.2231 | 0.0245 | -0.2632 | 0.0172 | -0.1628 |
| aic.bias.term | 0.0000 | 0.0000 | 0.0000 | 0.0000 | -0.0133 | -0.2338 | -0.0510 | -0.1656 |
| params | 0.0000 | 0.0000 | 0.0000 | 0.0000 | -0.0133 | -0.2338 | -0.0510 | -0.1656 |
| magnitude | 0.1389 | -0.1588 | 0.2981 | -0.5380 | 0.0665 | -0.2264 | 0.0409 | -0.2736 |
| pac.bayes.magnitude | 0.1389 | -0.1588 | 0.2981 | -0.5380 | 0.0665 | -0.2264 | 0.0409 | -0.2736 |
| pac.bayes.bound | -0.2269 | -0.1264 | 0.1796 | -0.4713 | -0.0957 | -0.2021 | -0.2208 | -0.1417 |
| pac.bayes.magflat | -0.1213 | 0.1338 | -0.2889 | 0.5546 | -0.0509 | -0.1973 | -0.1421 | -0.0308 |
| spectral.norm.per.layer | 0.0954 | -0.2208 | 0.2722 | -0.5435 | 0.0546 | -0.0622 | 0.0133 | -0.0353 |
| l1.over.margin.p10 | 0.0500 | -0.0940 | 0.3463 | -0.5102 | -0.3049 | -0.0446 | -0.1466 | -0.0840 |
| pac.bayes.magnitude.init | -0.0120 | -0.0597 | 0.2667 | -0.5472 | -0.0161 | 0.0259 | 0.0041 | 0.0952 |
| l2.over.margin.p10 | 0.1130 | -0.1069 | 0.3352 | -0.4954 | -0.3264 | 0.0860 | -0.1077 | -0.0026 |
| margin.normalized.param.norm | 0.1130 | -0.1069 | 0.3352 | -0.4954 | -0.3264 | 0.0861 | -0.1077 | -0.0026 |
| path.norm | 0.2694 | -0.2764 | 0.2611 | -0.5963 | 0.1767 | 0.1426 | 0.1889 | -0.1554 |
| inverse.margin.p10 | 0.2269 | 0.0921 | -0.1630 | 0.3565 | 0.0791 | 0.1497 | 0.0943 | 0.2775 |
| sharpness.magnitude | 0.2352 | 0.0125 | 0.3106 | -0.2523 | 0.0055 | 0.2085 | 0.1052 | 0.1739 |
| sharpness.magnitude.init | 0.2833 | 0.0255 | 0.2056 | -0.0324 | -0.0352 | 0.2204 | 0.0885 | 0.2291 |
| fisher.rao.norm | -0.0657 | 0.0440 | -0.1759 | 0.2009 | -0.3872 | 0.2438 | -0.1345 | 0.2347 |
| mce | 0.2968 | 0.0500 | -0.0370 | 0.3213 | -0.1667 | 0.2697 | 0.0917 | 0.2809 |
| sharpness.magflat | 0.2046 | 0.0171 | 0.1722 | -0.0657 | 0.0594 | 0.2868 | 0.1473 | 0.1805 |
| sharpness | 0.0009 | 0.1505 | -0.0500 | 0.3546 | -0.1193 | 0.2978 | -0.0401 | 0.2992 |
| reliability.diagram | 0.3370 | 0.0282 | -0.0278 | 0.3389 | -0.1612 | 0.3002 | 0.1114 | 0.2994 |
| waic.bias.term | 0.0398 | -0.0106 | -0.0907 | 0.1676 | -0.4471 | 0.3048 | -0.0918 | 0.2474 |
| cross.entropy | -0.0713 | 0.0588 | -0.3370 | 0.3028 | -0.4115 | 0.3119 | -0.1786 | 0.3324 |
| temperature.scaling | 0.0412 | 0.1551 | -0.2352 | 0.3537 | -0.3946 | 0.3323 | -0.1250 | 0.3527 |
| gradient.noise.var | 0.0380 | 0.0838 | -0.1981 | 0.4343 | -0.3597 | 0.3357 | -0.1189 | 0.3609 |
| tic.bias.term | 0.0389 | 0.0977 | -0.1981 | 0.4454 | -0.3530 | 0.3369 | -0.1111 | 0.3604 |
| gradient.norm | 0.0176 | 0.1079 | -0.2222 | 0.4620 | -0.3498 | 0.3414 | -0.1019 | 0.3783 |
| ace | 0.1106 | 0.0426 | -0.2148 | 0.4009 | -0.3430 | 0.3419 | -0.0811 | 0.3825 |
| ece | 0.1037 | 0.0477 | -0.2083 | 0.3944 | -0.3472 | 0.3421 | -0.0782 | 0.3817 |
| hessian.top.eigenvalue | 0.0287 | 0.1523 | -0.2167 | 0.4602 | -0.2270 | 0.3528 | -0.0927 | 0.3644 |
| adaptive.sharpness | -0.0389 | 0.1866 | 0.0000 | 0.2954 | 0.0758 | 0.3609 | 0.1138 | 0.2344 |
| flatness.proxy | 0.0296 | 0.0995 | -0.2222 | 0.4472 | -0.3110 | 0.3669 | -0.1083 | 0.3974 |
| gradient.noise.final.var | 0.0352 | 0.0829 | -0.1981 | 0.4343 | -0.3060 | 0.3691 | -0.1028 | 0.3934 |
| input.gradient.norm | -0.0269 | 0.0773 | -0.2556 | 0.3083 | -0.3227 | 0.3693 | -0.1214 | 0.3818 |
| hessian.trace | 0.0204 | 0.1569 | -0.2389 | 0.4731 | -0.2230 | 0.3744 | -0.0843 | 0.3894 |

Table 14: nin summary of IID correlations and OODTestGap_P OOD sensitivities for each measure.

| Measure | $lr^{IID}$ | $lr^{OOD}$ | $wd^{IID}$ | $wd^{OOD}$ | $\tau^{IID}$ | $\tau^{OOD}$ | $\Psi^{IID}$ | $\Psi^{OOD}$ |
|---|---|---|---|---|---|---|---|---|
| vcdim | 0.0014 | 0.0023 | 0.0019 | 0.0000 | -0.1010 | -0.4798 | -0.0503 | -0.2026 |
| frobenius.distance | -0.0917 | -0.1352 | 0.0704 | -0.0389 | -0.1200 | -0.4116 | -0.1719 | -0.1002 |
| negative.entropy | 0.0824 | -0.2148 | 0.3444 | -0.3963 | 0.3927 | -0.3563 | 0.1724 | -0.4266 |
| spec.sum | 0.0963 | -0.2472 | 0.2778 | -0.5130 | -0.0134 | -0.3556 | -0.0625 | -0.2485 |
| spec.prod | 0.1185 | -0.2361 | 0.2741 | -0.5056 | -0.0179 | -0.3329 | -0.0441 | -0.1958 |
| gradient.noise.scale | 0.0898 | -0.0926 | 0.1259 | -0.2259 | 0.0245 | -0.3027 | 0.0172 | -0.2030 |
| aic.bias.term | 0.0000 | 0.0000 | 0.0000 | 0.0000 | -0.0133 | -0.2772 | -0.0510 | -0.2031 |
| params | 0.0000 | 0.0000 | 0.0000 | 0.0000 | -0.0133 | -0.2772 | -0.0510 | -0.2031 |
| magnitude | 0.1389 | -0.2454 | 0.2981 | -0.4981 | 0.0665 | -0.2543 | 0.0409 | -0.3255 |
| pac.bayes.magnitude | 0.1389 | -0.2454 | 0.2981 | -0.4981 | 0.0665 | -0.2543 | 0.0409 | -0.3255 |
| pac.bayes.bound | -0.2269 | -0.0722 | 0.1796 | -0.4315 | -0.0957 | -0.2428 | -0.2208 | -0.1629 |
| pac.bayes.magflat | -0.1213 | 0.2296 | -0.2889 | 0.5111 | -0.0509 | -0.2325 | -0.1421 | -0.0533 |
| spectral.norm.per.layer | 0.0954 | -0.2481 | 0.2722 | -0.5037 | 0.0546 | -0.0768 | 0.0133 | -0.0610 |
| l1.over.margin.p10 | 0.0500 | -0.1250 | 0.3463 | -0.4741 | -0.3049 | -0.0681 | -0.1466 | -0.1007 |
| pac.bayes.magnitude.init | -0.0120 | -0.1481 | 0.2667 | -0.5167 | -0.0161 | 0.0054 | 0.0041 | 0.0574 |
| l2.over.margin.p10 | 0.1130 | -0.0750 | 0.3352 | -0.4593 | -0.3264 | 0.0837 | -0.1077 | 0.0025 |
| margin.normalized.param.norm | 0.1130 | -0.0750 | 0.3352 | -0.4593 | -0.3264 | 0.0837 | -0.1077 | 0.0025 |
| path.norm | 0.2694 | -0.3972 | 0.2611 | -0.6185 | 0.1767 | 0.1095 | 0.1889 | -0.2139 |
| inverse.margin.p10 | 0.2269 | 0.1333 | -0.1630 | 0.3667 | 0.0791 | 0.1428 | 0.0943 | 0.2937 |
| mce | 0.2968 | 0.0477 | -0.0370 | 0.3167 | -0.1667 | 0.2560 | 0.0917 | 0.2732 |
| sharpness.magnitude | 0.2352 | 0.1509 | 0.3106 | -0.1718 | 0.0055 | 0.2562 | 0.1052 | 0.2471 |
| fisher.rao.norm | -0.0657 | 0.1815 | -0.1759 | 0.2815 | -0.3872 | 0.2671 | -0.1345 | 0.2998 |
| sharpness.magnitude.init | 0.2833 | 0.1213 | 0.2056 | 0.0296 | -0.0352 | 0.2684 | 0.0885 | 0.2898 |
| reliability.diagram | 0.3370 | 0.0148 | -0.0278 | 0.3185 | -0.1612 | 0.2894 | 0.1114 | 0.2904 |
| waic.bias.term | 0.0398 | 0.1241 | -0.0907 | 0.2389 | -0.4471 | 0.3151 | -0.0918 | 0.3091 |
| sharpness | 0.0009 | 0.2759 | -0.0500 | 0.3704 | -0.1193 | 0.3349 | -0.0401 | 0.3665 |
| cross.entropy | -0.0713 | 0.2000 | -0.3370 | 0.3907 | -0.4115 | 0.3450 | -0.1786 | 0.4123 |
| sharpness.magflat | 0.2046 | 0.1315 | 0.1722 | -0.0463 | 0.0594 | 0.3468 | 0.1473 | 0.2426 |
| temperature.scaling | 0.0412 | 0.2431 | -0.2352 | 0.3944 | -0.3946 | 0.3540 | -0.1250 | 0.4089 |
| gradient.noise.var | 0.0380 | 0.2204 | -0.1981 | 0.4500 | -0.3597 | 0.3647 | -0.1189 | 0.4196 |
| tic.bias.term | 0.0389 | 0.2324 | -0.1981 | 0.4611 | -0.3530 | 0.3669 | -0.1111 | 0.4237 |
| ece | 0.1037 | 0.1676 | -0.2083 | 0.4676 | -0.3472 | 0.3741 | -0.0782 | 0.4530 |
| ace | 0.1106 | 0.1616 | -0.2148 | 0.4667 | -0.3430 | 0.3745 | -0.0811 | 0.4510 |
| gradient.norm | 0.0176 | 0.2444 | -0.2222 | 0.4870 | -0.3498 | 0.3749 | -0.1019 | 0.4471 |
| adaptive.sharpness | -0.0389 | 0.2935 | 0.0000 | 0.2907 | 0.0758 | 0.3892 | 0.1138 | 0.2729 |
| hessian.top.eigenvalue | 0.0287 | 0.2870 | -0.2167 | 0.4519 | -0.2270 | 0.3920 | -0.0927 | 0.4230 |
| input.gradient.norm | -0.0269 | 0.2185 | -0.2556 | 0.3963 | -0.3227 | 0.4010 | -0.1214 | 0.4537 |
| flatness.proxy | 0.0296 | 0.2361 | -0.2222 | 0.4667 | -0.3110 | 0.4033 | -0.1083 | 0.4617 |
| gradient.noise.final.var | 0.0352 | 0.2194 | -0.1981 | 0.4500 | -0.3060 | 0.4051 | -0.1028 | 0.4542 |
| hessian.trace | 0.0204 | 0.2806 | -0.2389 | 0.4667 | -0.2230 | 0.4157 | -0.0843 | 0.4520 |

Table 15: resnet summary of IID correlations and OODGenGap_C OOD sensitivities for each measure.

| Measure | $lr^{IID}$ | $lr^{OOD}$ | $wd^{IID}$ | $wd^{OOD}$ | $\tau^{IID}$ | $\tau^{OOD}$ | $\Psi^{IID}$ | $\Psi^{OOD}$ |
|---|---|---|---|---|---|---|---|---|
| negative.entropy | -0.4478 | -0.2058 | 0.4039 | -0.2017 | -0.2661 | -0.1924 | 0.0243 | 0.0279 |
| spec.prod | -0.3106 | -0.1119 | 0.3717 | -0.2128 | -0.1297 | -0.1648 | -0.0626 | -0.0476 |
| spec.sum | -0.4039 | -0.0197 | 0.3939 | -0.2094 | -0.1910 | -0.1033 | -0.0717 | -0.0196 |
| spectral.norm.per.layer | -0.4033 | -0.0192 | 0.3894 | -0.2161 | -0.1910 | -0.1033 | -0.0693 | -0.0175 |
| frobenius.distance | -0.6822 | 0.0919 | 0.1317 | -0.0417 | -0.5833 | -0.0814 | -0.1413 | 0.1229 |
| pac.bayes.magnitude.init | -0.6600 | 0.0997 | 0.3650 | -0.1506 | -0.5285 | -0.0791 | -0.0034 | 0.0714 |
| magnitude | -0.2872 | 0.0325 | 0.3772 | -0.2028 | -0.1449 | -0.0753 | 0.0445 | 0.0777 |
| pac.bayes.magnitude | -0.2872 | 0.0325 | 0.3772 | -0.2028 | -0.1449 | -0.0753 | 0.0445 | 0.0777 |
| pac.bayes.bound | -0.1217 | 0.0503 | 0.2894 | -0.2706 | 0.0364 | -0.0108 | -0.0685 | -0.0818 |
| aic.bias.term | 0.0000 | 0.0000 | 0.0000 | 0.0000 | 0.0000 | 0.0000 | 0.0000 | 0.0000 |
| params | 0.0000 | 0.0000 | 0.0000 | 0.0000 | 0.0000 | 0.0000 | 0.0000 | 0.0000 |
| vcdim | 0.0000 | 0.0000 | 0.0000 | 0.0000 | 0.0000 | 0.0000 | 0.0000 | 0.0000 |
| gradient.noise.scale | 0.0622 | 0.0064 | 0.1439 | -0.0483 | 0.0814 | 0.0152 | 0.0547 | -0.0050 |
| l1.over.margin.p10 | 0.1544 | 0.0797 | 0.4339 | -0.2228 | 0.2338 | 0.0399 | 0.2190 | -0.0224 |
| path.norm | 0.4524 | 0.0048 | 1.0000 | 1.0000 | 0.3136 | 0.0484 | 0.3016 | 0.0829 |
| l2.over.margin.p10 | 0.1017 | 0.2169 | 0.4106 | -0.1850 | 0.1890 | 0.1230 | 0.2068 | 0.0224 |
| margin.normalized.param.norm | 0.1017 | 0.2169 | 0.4106 | -0.1850 | 0.1890 | 0.1230 | 0.2068 | 0.0224 |
| waic.bias.term | 0.5178 | 0.2219 | -0.3672 | 0.2094 | 0.1949 | 0.1651 | -0.0599 | 0.1029 |
| cross.entropy | 0.4339 | 0.2108 | -0.3883 | 0.1739 | 0.2671 | 0.1972 | -0.0309 | -0.0252 |
| ace | 0.4511 | 0.2203 | -0.3450 | 0.2350 | 0.2817 | 0.2088 | 0.0330 | 0.0279 |
| ece | 0.4528 | 0.2219 | -0.3394 | 0.2439 | 0.2832 | 0.2110 | 0.0410 | 0.0348 |
| pac.bayes.magflat | 0.2456 | 0.0925 | -0.3583 | 0.2450 | 0.1907 | 0.2238 | 0.0595 | 0.0775 |
| temperature.scaling | 0.4153 | 0.2428 | -0.3017 | 0.1994 | 0.2873 | 0.2281 | 0.0833 | 0.1138 |
| gradient.norm | 0.5250 | 0.1819 | -0.3383 | 0.2361 | 0.3442 | 0.2489 | 0.0992 | 0.1216 |
| reliability.diagram | 0.4614 | 0.3078 | -0.1872 | 0.1928 | 0.2632 | 0.2505 | 0.0621 | 0.1290 |
| hessian.trace | 0.5700 | 0.1836 | -0.3106 | 0.2383 | 0.4038 | 0.2508 | 0.1396 | 0.0866 |
| tic.bias.term | 0.5533 | 0.1936 | -0.2939 | 0.2339 | 0.4025 | 0.2520 | 0.0814 | 0.0846 |
| flatness.proxy | 0.5178 | 0.1903 | -0.3039 | 0.2061 | 0.3456 | 0.2596 | 0.0784 | 0.1325 |
| gradient.noise.var | 0.5189 | 0.1892 | -0.3006 | 0.2128 | 0.3468 | 0.2598 | 0.0738 | 0.1396 |
| gradient.noise.final.var | 0.5183 | 0.1897 | -0.3039 | 0.2006 | 0.3465 | 0.2598 | 0.0674 | 0.1336 |
| mce | 0.4667 | 0.3203 | -0.1794 | 0.1861 | 0.2659 | 0.2601 | 0.0928 | 0.1530 |
| fisher.rao.norm | 0.4694 | 0.2686 | -0.2461 | 0.1739 | 0.3490 | 0.2639 | 0.0797 | 0.0893 |
| hessian.top.eigenvalue | 0.5611 | 0.1903 | -0.2939 | 0.2606 | 0.3966 | 0.2642 | 0.1322 | 0.1171 |
| adaptive.sharpness | 0.5928 | 0.2497 | 0.0672 | 0.1872 | 0.4458 | 0.2676 | 0.2471 | 0.0527 |
| input.gradient.norm | 0.4689 | 0.2947 | -0.3217 | 0.2772 | 0.3094 | 0.2846 | 0.0270 | 0.1500 |
| sharpness | 0.5994 | 0.2108 | -0.2161 | 0.3017 | 0.4510 | 0.3019 | 0.2428 | 0.1872 |
| inverse.margin.p10 | 0.4300 | 0.3869 | -0.3106 | 0.2683 | 0.1560 | 0.3023 | 0.0310 | 0.0617 |
| sharpness.magflat | 0.5394 | 0.3958 | 0.1061 | 0.0161 | 0.5047 | 0.3696 | 0.1958 | 0.1301 |
| sharpness.magnitude | 0.4435 | 0.3923 | 0.1550 | 0.0106 | 0.4434 | 0.3971 | 0.2316 | 0.1357 |
| sharpness.magnitude.init | 0.3728 | 0.4425 | 0.1439 | 0.1083 | 0.3885 | 0.4430 | 0.2158 | 0.2283 |

Table 16: resnet summary of IID correlations and OODGenGap_P OOD sensitivities for each measure.

| Measure | $lr^{IID}$ | $lr^{OOD}$ | $wd^{IID}$ | $wd^{OOD}$ | $\tau^{IID}$ | $\tau^{OOD}$ | $\Psi^{IID}$ | $\Psi^{OOD}$ |
|---|---|---|---|---|---|---|---|---|
| negative.entropy | -0.4478 | -0.3878 | 0.4039 | -0.1667 | -0.2661 | -0.3257 | 0.0243 | -0.0233 |
| frobenius.distance | -0.6822 | -0.1189 | 0.1317 | -0.0733 | -0.5833 | -0.2646 | -0.1413 | 0.0732 |
| pac.bayes.magnitude.init | -0.6600 | -0.1122 | 0.3650 | -0.1444 | -0.5285 | -0.2599 | -0.0034 | 0.0266 |
| spec.prod | -0.3106 | -0.2250 | 0.3717 | -0.1800 | -0.1297 | -0.2124 | -0.0626 | -0.0712 |
| spectral.norm.per.layer | -0.4033 | -0.1422 | 0.3894 | -0.1833 | -0.1910 | -0.1768 | -0.0693 | -0.0381 |
| spec.sum | -0.4039 | -0.1428 | 0.3939 | -0.1722 | -0.1910 | -0.1768 | -0.0717 | -0.0364 |
| magnitude | -0.2872 | -0.0550 | 0.3772 | -0.1700 | -0.1449 | -0.1361 | 0.0445 | 0.0595 |
| pac.bayes.magnitude | -0.2872 | -0.0550 | 0.3772 | -0.1700 | -0.1449 | -0.1361 | 0.0445 | 0.0595 |
| aic.bias.term | 0.0000 | 0.0000 | 0.0000 | 0.0000 | 0.0000 | 0.0000 | 0.0000 | 0.0000 |
| params | 0.0000 | 0.0000 | 0.0000 | 0.0000 | 0.0000 | 0.0000 | 0.0000 | 0.0000 |
| vcdim | 0.0000 | 0.0000 | 0.0000 | 0.0000 | 0.0000 | 0.0000 | 0.0000 | 0.0000 |
| gradient.noise.scale | 0.0622 | 0.0144 | 0.1439 | -0.0444 | 0.0814 | 0.0149 | 0.0547 | -0.0054 |
| path.norm | 0.4524 | -0.0619 | 1.0000 | 1.0000 | 0.3136 | 0.0402 | 0.3016 | 0.0597 |
| pac.bayes.bound | -0.1217 | 0.1117 | 0.2894 | -0.2133 | 0.0364 | 0.0653 | -0.0685 | -0.0619 |
| l1.over.margin.p10 | 0.1544 | 0.2067 | 0.4339 | -0.1611 | 0.2338 | 0.1560 | 0.2190 | 0.0069 |
| l2.over.margin.p10 | 0.1017 | 0.3350 | 0.4106 | -0.1167 | 0.1890 | 0.2289 | 0.2068 | 0.0538 |
| margin.normalized.param.norm | 0.1017 | 0.3350 | 0.4106 | -0.1167 | 0.1890 | 0.2289 | 0.2068 | 0.0538 |
| waic.bias.term | 0.5178 | 0.4222 | -0.3672 | 0.1856 | 0.1949 | 0.2666 | -0.0599 | 0.1609 |
| inverse.margin.p10 | 0.4300 | 0.3911 | -0.3106 | 0.2378 | 0.1560 | 0.2672 | 0.0310 | 0.0831 |
| pac.bayes.magflat | 0.2456 | 0.1867 | -0.3583 | 0.1989 | 0.1907 | 0.2837 | 0.0595 | 0.1110 |
| cross.entropy | 0.4339 | 0.3961 | -0.3883 | 0.1522 | 0.2671 | 0.3318 | -0.0309 | 0.0322 |
| ace | 0.4511 | 0.4133 | -0.3450 | 0.2111 | 0.2817 | 0.3435 | 0.0330 | 0.0802 |
| reliability.diagram | 0.4614 | 0.4531 | -0.1872 | 0.1633 | 0.2632 | 0.3452 | 0.0621 | 0.1568 |
| ece | 0.4528 | 0.4139 | -0.3394 | 0.2178 | 0.2832 | 0.3461 | 0.0410 | 0.0895 |
| mce | 0.4667 | 0.4622 | -0.1794 | 0.1700 | 0.2659 | 0.3517 | 0.0928 | 0.1858 |
| temperature.scaling | 0.4153 | 0.4342 | -0.3017 | 0.1856 | 0.2873 | 0.3673 | 0.0833 | 0.1612 |
| gradient.norm | 0.5250 | 0.3894 | -0.3383 | 0.2300 | 0.3442 | 0.3906 | 0.0992 | 0.1824 |
| adaptive.sharpness | 0.5928 | 0.4306 | 0.0672 | 0.2300 | 0.4458 | 0.3914 | 0.2471 | 0.1434 |
| flatness.proxy | 0.5178 | 0.3978 | -0.3039 | 0.2067 | 0.3456 | 0.4022 | 0.0784 | 0.1933 |
| hessian.trace | 0.5700 | 0.3900 | -0.3106 | 0.2256 | 0.4038 | 0.4025 | 0.1396 | 0.1417 |
| gradient.noise.final.var | 0.5183 | 0.3972 | -0.3039 | 0.1989 | 0.3465 | 0.4031 | 0.0674 | 0.1903 |
| gradient.noise.var | 0.5189 | 0.3967 | -0.3006 | 0.2089 | 0.3468 | 0.4032 | 0.0738 | 0.1956 |
| hessian.top.eigenvalue | 0.5611 | 0.3911 | -0.2939 | 0.2322 | 0.3966 | 0.4102 | 0.1322 | 0.1571 |
| tic.bias.term | 0.5533 | 0.4022 | -0.2939 | 0.2322 | 0.4025 | 0.4112 | 0.0814 | 0.1487 |
| fisher.rao.norm | 0.4694 | 0.4706 | -0.2461 | 0.1678 | 0.3490 | 0.4141 | 0.0797 | 0.1402 |
| input.gradient.norm | 0.4689 | 0.4922 | -0.3217 | 0.2644 | 0.3094 | 0.4154 | 0.0270 | 0.1942 |
| sharpness | 0.5994 | 0.4072 | -0.2161 | 0.2733 | 0.4510 | 0.4302 | 0.2428 | 0.2555 |
| sharpness.magflat | 0.5394 | 0.5706 | 0.1061 | 0.0667 | 0.5047 | 0.5096 | 0.1958 | 0.1937 |
| sharpness.magnitude | 0.4435 | 0.5398 | 0.1550 | 0.0811 | 0.4434 | 0.5122 | 0.2316 | 0.2033 |
| sharpness.magnitude.init | 0.3728 | 0.5417 | 0.1439 | 0.1700 | 0.3885 | 0.5189 | 0.2158 | 0.2812 |

Table 17: resnet summary of IID correlations and OODTestGap_C OOD sensitivities for each measure.

| Measure | $lr^{IID}$ | $lr^{OOD}$ | $wd^{IID}$ | $wd^{OOD}$ | $\tau^{IID}$ | $\tau^{OOD}$ | $\Psi^{IID}$ | $\Psi^{OOD}$ |
|---|---|---|---|---|---|---|---|---|
| hessian.trace | 0.5700 | -0.5456 | -0.3106 | 0.3044 | 0.4038 | -0.3619 | 0.1396 | -0.1133 |
| tic.bias.term | 0.5533 | -0.5211 | -0.2939 | 0.3056 | 0.4025 | -0.3453 | 0.0814 | -0.0881 |
| hessian.top.eigenvalue | 0.5611 | -0.5356 | -0.2939 | 0.3033 | 0.3966 | -0.3402 | 0.1322 | -0.0873 |
| sharpness | 0.5994 | -0.5200 | -0.2161 | 0.3300 | 0.4510 | -0.3343 | 0.2428 | -0.0600 |
| adaptive.sharpness | 0.5928 | -0.4644 | 0.0672 | 0.1689 | 0.4458 | -0.3267 | 0.2471 | -0.0974 |
| gradient.norm | 0.5250 | -0.5406 | -0.3383 | 0.3189 | 0.3442 | -0.3226 | 0.0992 | -0.0577 |
| cross.entropy | 0.4339 | -0.4628 | -0.3883 | 0.2767 | 0.2671 | -0.3194 | -0.0309 | -0.1006 |
| gradient.noise.var | 0.5189 | -0.5211 | -0.3006 | 0.2778 | 0.3468 | -0.3110 | 0.0738 | -0.0283 |
| gradient.noise.final.var | 0.5183 | -0.5206 | -0.3039 | 0.2789 | 0.3465 | -0.3105 | 0.0674 | -0.0334 |
| flatness.proxy | 0.5178 | -0.5200 | -0.3039 | 0.2778 | 0.3456 | -0.3104 | 0.0784 | -0.0368 |
| ace | 0.4511 | -0.4589 | -0.3450 | 0.3044 | 0.2817 | -0.3093 | 0.0330 | -0.0931 |
| ece | 0.4528 | -0.4561 | -0.3394 | 0.3133 | 0.2832 | -0.3087 | 0.0410 | -0.0930 |
| temperature.scaling | 0.4153 | -0.4297 | -0.3017 | 0.2506 | 0.2873 | -0.3056 | 0.0833 | -0.0697 |
| fisher.rao.norm | 0.4694 | -0.4139 | -0.2461 | 0.2456 | 0.3490 | -0.2967 | 0.0797 | -0.0718 |
| sharpness.magflat | 0.5394 | -0.2933 | 0.1061 | -0.0122 | 0.5047 | -0.2801 | 0.1958 | -0.0437 |
| input.gradient.norm | 0.4689 | -0.4111 | -0.3217 | 0.3711 | 0.3094 | -0.2357 | 0.0270 | 0.0196 |
| sharpness.magnitude | 0.4435 | -0.2344 | 0.1550 | -0.0156 | 0.4434 | -0.2180 | 0.2316 | -0.0365 |
| l1.over.margin.p10 | 0.1544 | -0.1833 | 0.4339 | -0.2900 | 0.2338 | -0.2174 | 0.2190 | -0.1450 |
| pac.bayes.magflat | 0.2456 | -0.3422 | -0.3583 | 0.3144 | 0.1907 | -0.2099 | 0.0595 | -0.0646 |
| reliability.diagram | 0.4614 | -0.2258 | -0.1872 | 0.2233 | 0.2632 | -0.1667 | 0.0621 | 0.0246 |
| mce | 0.4667 | -0.2222 | -0.1794 | 0.2200 | 0.2659 | -0.1654 | 0.0928 | 0.0194 |
| waic.bias.term | 0.5178 | -0.4822 | -0.3672 | 0.3167 | 0.1949 | -0.1609 | -0.0599 | 0.0286 |
| l2.over.margin.p10 | 0.1017 | -0.0794 | 0.4106 | -0.2489 | 0.1890 | -0.1509 | 0.2068 | -0.1162 |
| margin.normalized.param.norm | 0.1017 | -0.0794 | 0.4106 | -0.2489 | 0.1890 | -0.1509 | 0.2068 | -0.1162 |
| pac.bayes.bound | -0.1217 | 0.0239 | 0.2894 | -0.2889 | 0.0364 | -0.1092 | -0.0685 | -0.0107 |
| sharpness.magnitude.init | 0.3728 | -0.0904 | 0.1439 | 0.0522 | 0.3885 | -0.1086 | 0.2158 | 0.0209 |
| gradient.noise.scale | 0.0622 | -0.0394 | 0.1439 | -0.1100 | 0.0814 | -0.0251 | 0.0547 | -0.0133 |
| inverse.margin.p10 | 0.4300 | -0.1394 | -0.3106 | 0.3444 | 0.1560 | -0.0249 | 0.0310 | -0.0409 |
| aic.bias.term | 0.0000 | 0.0000 | 0.0000 | 0.0000 | 0.0000 | 0.0000 | 0.0000 | 0.0000 |
| params | 0.0000 | 0.0000 | 0.0000 | 0.0000 | 0.0000 | 0.0000 | 0.0000 | 0.0000 |
| vcdim | 0.0000 | 0.0000 | 0.0000 | 0.0000 | 0.0000 | 0.0000 | 0.0000 | 0.0000 |
| path.norm | 0.4524 | -0.0857 | 1.0000 | 1.0000 | 0.3136 | 0.0079 | 0.3016 | 0.0101 |
| spec.prod | -0.3106 | 0.3950 | 0.3717 | -0.2833 | -0.1297 | 0.1799 | -0.0626 | 0.1149 |
| magnitude | -0.2872 | 0.4361 | 0.3772 | -0.2822 | -0.1449 | 0.2154 | 0.0445 | 0.1109 |
| pac.bayes.magnitude | -0.2872 | 0.4361 | 0.3772 | -0.2822 | -0.1449 | 0.2154 | 0.0445 | 0.1109 |
| spectral.norm.per.layer | -0.4033 | 0.5133 | 0.3894 | -0.2933 | -0.1910 | 0.2616 | -0.0693 | 0.1426 |
| spec.sum | -0.4039 | 0.5139 | 0.3939 | -0.2911 | -0.1910 | 0.2616 | -0.0717 | 0.1435 |
| negative.entropy | -0.4478 | 0.4722 | 0.4039 | -0.3044 | -0.2661 | 0.3207 | 0.0243 | 0.1042 |
| pac.bayes.magnitude.init | -0.6600 | 0.8389 | 0.3650 | -0.2522 | -0.5285 | 0.5757 | -0.0034 | 0.1665 |
| frobenius.distance | -0.6822 | 0.8322 | 0.1317 | -0.0767 | -0.5833 | 0.5924 | -0.1413 | 0.2695 |

Table 18: resnet summary of IID correlations and OODTestGap_P OOD sensitivities for each measure.

| Measure | $lr^{IID}$ | $lr^{OOD}$ | $wd^{IID}$ | $wd^{OOD}$ | $\tau^{IID}$ | $\tau^{OOD}$ | $\Psi^{IID}$ | $\Psi^{OOD}$ |
|---|---|---|---|---|---|---|---|---|
| hessian.trace | 0.5700 | -0.4861 | -0.3106 | 0.3211 | 0.4038 | -0.3017 | 0.1396 | -0.1007 |
| adaptive.sharpness | 0.5928 | -0.4122 | 0.0672 | 0.1856 | 0.4458 | -0.2829 | 0.2471 | -0.0812 |
| tic.bias.term | 0.5533 | -0.4583 | -0.2939 | 0.3211 | 0.4025 | -0.2817 | 0.0814 | -0.0511 |
| hessian.top.eigenvalue | 0.5611 | -0.4728 | -0.2939 | 0.3300 | 0.3966 | -0.2807 | 0.1322 | -0.0663 |
| sharpness | 0.5994 | -0.4611 | -0.2161 | 0.3444 | 0.4510 | -0.2804 | 0.2428 | -0.0274 |
| gradient.norm | 0.5250 | -0.4767 | -0.3383 | 0.3456 | 0.3442 | -0.2612 | 0.0992 | -0.0178 |
| cross.entropy | 0.4339 | -0.3989 | -0.3883 | 0.3144 | 0.2671 | -0.2575 | -0.0309 | -0.0619 |
| ace | 0.4511 | -0.3939 | -0.3450 | 0.3444 | 0.2817 | -0.2492 | 0.0330 | -0.0612 |
| gradient.noise.var | 0.5189 | -0.4572 | -0.3006 | 0.2956 | 0.3468 | -0.2492 | 0.0738 | 0.0104 |
| gradient.noise.final.var | 0.5183 | -0.4567 | -0.3039 | 0.2878 | 0.3465 | -0.2487 | 0.0674 | 0.0032 |
| flatness.proxy | 0.5178 | -0.4561 | -0.3039 | 0.2933 | 0.3456 | -0.2485 | 0.0784 | 0.0056 |
| ece | 0.4528 | -0.3922 | -0.3394 | 0.3578 | 0.2832 | -0.2481 | 0.0410 | -0.0584 |
| temperature.scaling | 0.4153 | -0.3653 | -0.3017 | 0.2933 | 0.2873 | -0.2420 | 0.0833 | -0.0343 |
| sharpness.magflat | 0.5394 | -0.2389 | 0.1061 | -0.0089 | 0.5047 | -0.2302 | 0.1958 | -0.0316 |
| fisher.rao.norm | 0.4694 | -0.3489 | -0.2461 | 0.2611 | 0.3490 | -0.2297 | 0.0797 | -0.0455 |
| input.gradient.norm | 0.4689 | -0.3461 | -0.3217 | 0.3956 | 0.3094 | -0.1755 | 0.0270 | 0.0571 |
| l1.over.margin.p10 | 0.1544 | -0.1328 | 0.4339 | -0.2678 | 0.2338 | -0.1674 | 0.2190 | -0.1317 |
| pac.bayes.magflat | 0.2456 | -0.2994 | -0.3583 | 0.3344 | 0.1907 | -0.1618 | 0.0595 | -0.0452 |
| sharpness.magnitude | 0.4435 | -0.1696 | 0.1550 | -0.0033 | 0.4434 | -0.1600 | 0.2316 | -0.0375 |
| reliability.diagram | 0.4614 | -0.1864 | -0.1872 | 0.2244 | 0.2632 | -0.1334 | 0.0621 | 0.0298 |
| mce | 0.4667 | -0.1839 | -0.1794 | 0.2367 | 0.2659 | -0.1328 | 0.0928 | 0.0334 |
| waic.bias.term | 0.5178 | -0.4161 | -0.3672 | 0.3367 | 0.1949 | -0.1190 | -0.0599 | 0.0662 |
| l2.over.margin.p10 | 0.1017 | -0.0200 | 0.4106 | -0.2211 | 0.1890 | -0.0955 | 0.2068 | -0.0977 |
| margin.normalized.param.norm | 0.1017 | -0.0200 | 0.4106 | -0.2211 | 0.1890 | -0.0955 | 0.2068 | -0.0977 |
| pac.bayes.bound | -0.1217 | 0.0589 | 0.2894 | -0.2911 | 0.0364 | -0.0723 | -0.0685 | 0.0015 |
| sharpness.magnitude.init | 0.3728 | -0.0467 | 0.1439 | 0.0767 | 0.3885 | -0.0623 | 0.2158 | 0.0267 |
| gradient.noise.scale | 0.0622 | -0.0439 | 0.1439 | -0.1111 | 0.0814 | -0.0220 | 0.0547 | -0.0298 |
| path.norm | 0.4524 | -0.1238 | 1.0000 | 1.0000 | 0.3136 | -0.0137 | 0.3016 | 0.0163 |
| inverse.margin.p10 | 0.4300 | -0.1039 | -0.3106 | 0.3689 | 0.1560 | -0.0000 | 0.0310 | -0.0158 |
| aic.bias.term | 0.0000 | 0.0000 | 0.0000 | 0.0000 | 0.0000 | 0.0000 | 0.0000 | 0.0000 |
| params | 0.0000 | 0.0000 | 0.0000 | 0.0000 | 0.0000 | 0.0000 | 0.0000 | 0.0000 |
| vcdim | 0.0000 | 0.0000 | 0.0000 | 0.0000 | 0.0000 | 0.0000 | 0.0000 | 0.0000 |
| spec.prod | -0.3106 | 0.3567 | 0.3717 | -0.2956 | -0.1297 | 0.1457 | -0.0626 | 0.0965 |
| magnitude | -0.2872 | 0.4144 | 0.3772 | -0.2922 | -0.1449 | 0.1912 | 0.0445 | 0.1004 |
| pac.bayes.magnitude | -0.2872 | 0.4144 | 0.3772 | -0.2922 | -0.1449 | 0.1912 | 0.0445 | 0.1004 |
| spectral.norm.per.layer | -0.4033 | 0.4772 | 0.3894 | -0.3056 | -0.1910 | 0.2278 | -0.0693 | 0.1277 |
| spec.sum | -0.4039 | 0.4778 | 0.3939 | -0.3056 | -0.1910 | 0.2279 | -0.0717 | 0.1281 |
| negative.entropy | -0.4478 | 0.4094 | 0.4039 | -0.3289 | -0.2661 | 0.2607 | 0.0243 | 0.0766 |
| pac.bayes.magnitude.init | -0.6600 | 0.7961 | 0.3650 | -0.2689 | -0.5285 | 0.5261 | -0.0034 | 0.1444 |
| frobenius.distance | -0.6822 | 0.7906 | 0.1317 | -0.0844 | -0.5833 | 0.5409 | -0.1413 | 0.2564 |

Table 19: simplecnn summary of IID correlations and OODGenGap_C OOD sensitivities for each measure.

| Measure | $lr^{IID}$ | $lr^{OOD}$ | $wd^{IID}$ | $wd^{OOD}$ | $\tau^{IID}$ | $\tau^{OOD}$ | $\Psi^{IID}$ | $\Psi^{OOD}$ |
|---|---|---|---|---|---|---|---|---|
| frobenius.distance | -0.3391 | -0.1753 | 0.1880 | -0.0200 | -0.3146 | -0.2264 | -0.0768 | -0.0040 |
| spec.prod | -0.3307 | -0.1828 | 0.2325 | -0.0258 | -0.2764 | -0.2160 | -0.0727 | -0.0123 |
| pac.bayes.magnitude.init | -0.3238 | -0.1652 | 0.1872 | -0.0198 | -0.3330 | -0.2132 | -0.1252 | -0.0130 |
| spec.sum | -0.3191 | -0.1823 | 0.2174 | -0.0252 | -0.1973 | -0.1751 | -0.0353 | -0.0066 |
| spectral.norm.per.layer | -0.3191 | -0.1825 | 0.2174 | -0.0254 | -0.1971 | -0.1750 | -0.0345 | -0.0068 |
| pac.bayes.bound | -0.2851 | -0.2029 | 0.1015 | -0.0755 | -0.1772 | -0.1507 | -0.1616 | -0.0703 |
| path.norm | -0.2206 | -0.1242 | 0.2134 | -0.0284 | -0.1524 | -0.1469 | -0.0767 | -0.0077 |
| pac.bayes.magnitude | -0.1921 | -0.1192 | 0.2087 | -0.0274 | -0.0421 | -0.0837 | 0.0055 | -0.0003 |
| magnitude | -0.1921 | -0.1192 | 0.2087 | -0.0274 | -0.0421 | -0.0837 | 0.0055 | -0.0003 |
| l2.over.margin.p10 | -0.1069 | -0.0999 | 0.1388 | -0.0179 | -0.0749 | -0.0671 | -0.0972 | -0.0238 |
| margin.normalized.param.norm | -0.1069 | -0.0999 | 0.1388 | -0.0179 | -0.0749 | -0.0671 | -0.0972 | -0.0238 |
| cross.entropy | 0.1602 | 0.0230 | -0.2389 | 0.0099 | -0.0248 | -0.0087 | -0.0776 | 0.0004 |
| aic.bias.term | 0.0000 | 0.0000 | 0.0000 | 0.0000 | 0.0000 | 0.0000 | 0.0000 | 0.0000 |
| params | 0.0000 | 0.0000 | 0.0000 | 0.0000 | 0.0000 | 0.0000 | 0.0000 | 0.0000 |
| vcdim | 0.0000 | 0.0000 | 0.0000 | 0.0000 | 0.0000 | 0.0000 | 0.0000 | 0.0000 |
| negative.entropy | -0.1751 | -0.0345 | 0.2278 | -0.0130 | 0.0242 | 0.0019 | 0.0733 | -0.0024 |
| temperature.scaling | 0.1549 | 0.0226 | -0.1735 | 0.0436 | -0.0076 | 0.0058 | -0.0292 | 0.0291 |
| pac.bayes.magflat | -0.0696 | 0.0062 | -0.1888 | 0.0270 | -0.0670 | 0.0205 | -0.0275 | -0.0074 |
| l1.over.margin.p10 | 0.0284 | -0.0214 | 0.1952 | -0.0263 | 0.0674 | 0.0241 | -0.0595 | -0.0076 |
| ece | 0.2068 | 0.0681 | -0.1810 | 0.0525 | 0.0320 | 0.0499 | -0.0292 | 0.0329 |
| ace | 0.2070 | 0.0684 | -0.1763 | 0.0523 | 0.0308 | 0.0505 | -0.0283 | 0.0325 |
| waic.bias.term | 0.1196 | 0.0522 | -0.0208 | 0.0120 | -0.0439 | 0.0697 | -0.0981 | 0.0221 |
| mce | 0.2139 | 0.0859 | -0.0822 | 0.0444 | 0.0916 | 0.0739 | 0.0282 | 0.0627 |
| reliability.diagram | 0.2253 | 0.0890 | -0.0710 | 0.0499 | 0.1032 | 0.0792 | 0.0356 | 0.0640 |
| fisher.rao.norm | 0.2487 | 0.1013 | -0.1541 | 0.0533 | 0.0786 | 0.0874 | -0.0263 | 0.0464 |
| gradient.norm | 0.3044 | 0.1135 | -0.1897 | 0.0409 | 0.1276 | 0.1140 | 0.0018 | 0.0466 |
| gradient.noise.var | 0.3143 | 0.1100 | -0.1818 | 0.0448 | 0.1365 | 0.1150 | -0.0078 | 0.0470 |
| gradient.noise.final.var | 0.3145 | 0.1101 | -0.1818 | 0.0444 | 0.1364 | 0.1150 | -0.0071 | 0.0469 |
| flatness.proxy | 0.3159 | 0.1103 | -0.1826 | 0.0423 | 0.1364 | 0.1152 | -0.0046 | 0.0472 |
| gradient.noise.scale | 0.1735 | 0.1461 | 0.0302 | 0.0112 | 0.1454 | 0.1284 | 0.0448 | 0.0222 |
| input.gradient.norm | 0.3274 | 0.1442 | -0.1755 | 0.0561 | 0.1446 | 0.1426 | -0.0014 | 0.0535 |
| tic.bias.term | 0.3673 | 0.1705 | -0.1889 | 0.0432 | 0.1919 | 0.1766 | 0.0057 | 0.0573 |
| inverse.margin.p10 | 0.4279 | 0.2384 | -0.1880 | 0.0612 | 0.2885 | 0.2266 | 0.0272 | 0.0709 |
| hessian.trace | 0.3600 | 0.1956 | -0.1850 | 0.0494 | 0.2665 | 0.2281 | 0.0463 | 0.0667 |
| sharpness.magnitude.init | 0.3567 | 0.1035 | 0.1253 | 0.0154 | 0.4116 | 0.2337 | 0.1546 | 0.0544 |
| hessian.top.eigenvalue | 0.3558 | 0.1855 | -0.1834 | 0.0437 | 0.2814 | 0.2353 | 0.0356 | 0.0605 |
| sharpness.magnitude | 0.3844 | 0.1228 | 0.1538 | 0.0014 | 0.4412 | 0.2530 | 0.1651 | 0.0481 |
| sharpness.magflat | 0.4470 | 0.1874 | 0.1277 | 0.0353 | 0.5071 | 0.3068 | 0.1716 | 0.0794 |
| sharpness | 0.3899 | 0.2120 | -0.1325 | 0.0464 | 0.4265 | 0.3152 | 0.1725 | 0.0765 |
| adaptive.sharpness | 0.4603 | 0.2767 | -0.0317 | 0.0309 | 0.5249 | 0.3659 | 0.2212 | 0.0717 |

Table 20: simplecnn summary of IID correlations and OODGenGap_P OOD sensitivities for each measure.

| Measure | $lr^{IID}$ | $lr^{OOD}$ | $wd^{IID}$ | $wd^{OOD}$ | $\tau^{IID}$ | $\tau^{OOD}$ | $\Psi^{IID}$ | $\Psi^{OOD}$ |
|---|---|---|---|---|---|---|---|---|
| pac.bayes.magnitude.init | -0.3238 | -0.5013 | 0.1872 | -0.1385 | -0.3330 | -0.4376 | -0.1252 | -0.1054 |
| frobenius.distance | -0.3391 | -0.5045 | 0.1880 | -0.1425 | -0.3146 | -0.4220 | -0.0768 | -0.0601 |
| spec.prod | -0.3307 | -0.4925 | 0.2325 | -0.1639 | -0.2764 | -0.4170 | -0.0727 | -0.0755 |
| spec.sum | -0.3191 | -0.4846 | 0.2174 | -0.1615 | -0.1973 | -0.3352 | -0.0353 | -0.0431 |
| spectral.norm.per.layer | -0.3191 | -0.4846 | 0.2174 | -0.1631 | -0.1971 | -0.3350 | -0.0345 | -0.0442 |
| path.norm | -0.2206 | -0.4073 | 0.2134 | -0.1742 | -0.1524 | -0.3249 | -0.0767 | -0.0974 |
| magnitude | -0.1921 | -0.3677 | 0.2087 | -0.1790 | -0.0421 | -0.1906 | 0.0055 | -0.0550 |
| pac.bayes.magnitude | -0.1921 | -0.3677 | 0.2087 | -0.1790 | -0.0421 | -0.1906 | 0.0055 | -0.0550 |
| pac.bayes.bound | -0.2851 | -0.3737 | 0.1015 | -0.2386 | -0.1772 | -0.1882 | -0.1616 | -0.1847 |
| l2.over.margin.p10 | -0.1069 | -0.1955 | 0.1388 | -0.0926 | -0.0749 | -0.0996 | -0.0972 | -0.0629 |
| margin.normalized.param.norm | -0.1069 | -0.1955 | 0.1388 | -0.0926 | -0.0749 | -0.0996 | -0.0972 | -0.0629 |
| negative.entropy | -0.1751 | -0.1808 | 0.2278 | -0.1225 | 0.0242 | -0.0795 | 0.0733 | -0.0769 |
| aic.bias.term | 0.0000 | 0.0000 | 0.0000 | 0.0000 | 0.0000 | 0.0000 | 0.0000 | 0.0000 |
| params | 0.0000 | 0.0000 | 0.0000 | 0.0000 | 0.0000 | 0.0000 | 0.0000 | 0.0000 |
| vcdim | 0.0000 | 0.0000 | 0.0000 | 0.0000 | 0.0000 | 0.0000 | 0.0000 | 0.0000 |
| l1.over.margin.p10 | 0.0284 | -0.0362 | 0.1952 | -0.1496 | 0.0674 | 0.0336 | -0.0595 | -0.0418 |
| cross.entropy | 0.1602 | 0.1531 | -0.2389 | 0.1066 | -0.0248 | 0.0677 | -0.0776 | 0.0724 |
| temperature.scaling | 0.1549 | 0.1265 | -0.1735 | 0.1451 | -0.0076 | 0.0766 | -0.0292 | 0.1301 |
| pac.bayes.magflat | -0.0696 | 0.0487 | -0.1888 | 0.1607 | -0.0670 | 0.0776 | -0.0275 | 0.0303 |
| waic.bias.term | 0.1196 | 0.1484 | -0.0208 | 0.0075 | -0.0439 | 0.1056 | -0.0981 | 0.0673 |
| mce | 0.2139 | 0.1721 | -0.0822 | 0.1430 | 0.0916 | 0.1163 | 0.0282 | 0.1917 |
| reliability.diagram | 0.2253 | 0.1789 | -0.0710 | 0.1534 | 0.1032 | 0.1242 | 0.0356 | 0.1962 |
| ece | 0.2068 | 0.2010 | -0.1810 | 0.1677 | 0.0320 | 0.1311 | -0.0292 | 0.1212 |
| ace | 0.2070 | 0.2017 | -0.1763 | 0.1741 | 0.0308 | 0.1322 | -0.0283 | 0.1239 |
| gradient.noise.scale | 0.1735 | 0.2334 | 0.0302 | -0.0130 | 0.1454 | 0.1531 | 0.0448 | 0.0004 |
| fisher.rao.norm | 0.2487 | 0.2156 | -0.1541 | 0.1670 | 0.0786 | 0.1625 | -0.0263 | 0.1352 |
| input.gradient.norm | 0.3274 | 0.3323 | -0.1755 | 0.1765 | 0.1446 | 0.2457 | -0.0014 | 0.1355 |
| gradient.norm | 0.3044 | 0.3652 | -0.1897 | 0.1590 | 0.1276 | 0.2491 | 0.0018 | 0.1588 |
| gradient.noise.var | 0.3143 | 0.3571 | -0.1818 | 0.1582 | 0.1365 | 0.2491 | -0.0078 | 0.1546 |
| gradient.noise.final.var | 0.3145 | 0.3578 | -0.1818 | 0.1598 | 0.1364 | 0.2493 | -0.0071 | 0.1557 |
| flatness.proxy | 0.3159 | 0.3582 | -0.1826 | 0.1574 | 0.1364 | 0.2494 | -0.0046 | 0.1575 |
| tic.bias.term | 0.3673 | 0.4244 | -0.1889 | 0.1638 | 0.1919 | 0.3120 | 0.0057 | 0.1783 |
| inverse.margin.p10 | 0.4279 | 0.4331 | -0.1880 | 0.1766 | 0.2885 | 0.3285 | 0.0272 | 0.1603 |
| hessian.trace | 0.3600 | 0.5050 | -0.1850 | 0.1685 | 0.2665 | 0.4047 | 0.0463 | 0.2103 |
| hessian.top.eigenvalue | 0.3558 | 0.4907 | -0.1834 | 0.1566 | 0.2814 | 0.4250 | 0.0356 | 0.2039 |
| sharpness.magnitude.init | 0.3567 | 0.2941 | 0.1253 | 0.0227 | 0.4116 | 0.4534 | 0.1546 | 0.1862 |
| sharpness.magnitude | 0.3844 | 0.3504 | 0.1538 | -0.0415 | 0.4412 | 0.4934 | 0.1651 | 0.1529 |
| sharpness.magflat | 0.4470 | 0.4324 | 0.1277 | 0.0806 | 0.5071 | 0.5575 | 0.1716 | 0.2370 |
| sharpness | 0.3899 | 0.5702 | -0.1325 | 0.1885 | 0.4265 | 0.6016 | 0.1725 | 0.2718 |
| adaptive.sharpness | 0.4603 | 0.6341 | -0.0317 | 0.0972 | 0.5249 | 0.6557 | 0.2212 | 0.2145 |

Table 21: simplecnn summary of IID correlations and OODTestGap_C OOD sensitivities for each measure.

| Measure | $lr^{IID}$ | $lr^{OOD}$ | $wd^{IID}$ | $wd^{OOD}$ | $\tau^{IID}$ | $\tau^{OOD}$ | $\Psi^{IID}$ | $\Psi^{OOD}$ |
|---|---|---|---|---|---|---|---|---|
| frobenius.distance | -0.3391 | -0.0957 | 0.1880 | -0.0531 | -0.3146 | -0.1603 | -0.0768 | -0.0042 |
| spec.prod | -0.3307 | -0.1054 | 0.2325 | -0.0610 | -0.2764 | -0.1539 | -0.0727 | -0.0037 |
| pac.bayes.magnitude.init | -0.3238 | -0.0860 | 0.1872 | -0.0513 | -0.3330 | -0.1408 | -0.1252 | 0.0030 |
| spec.sum | -0.3191 | -0.1089 | 0.2174 | -0.0603 | -0.1973 | -0.1263 | -0.0353 | -0.0125 |
| spectral.norm.per.layer | -0.3191 | -0.1091 | 0.2174 | -0.0600 | -0.1971 | -0.1262 | -0.0345 | -0.0129 |
| pac.bayes.bound | -0.2851 | -0.1501 | 0.1015 | -0.0828 | -0.1772 | -0.1173 | -0.1616 | -0.0358 |
| path.norm | -0.2206 | -0.0770 | 0.2134 | -0.0632 | -0.1524 | -0.1099 | -0.0767 | 0.0011 |
| pac.bayes.magnitude | -0.1921 | -0.0794 | 0.2087 | -0.0620 | -0.0421 | -0.0689 | 0.0055 | -0.0151 |
| magnitude | -0.1921 | -0.0794 | 0.2087 | -0.0620 | -0.0421 | -0.0689 | 0.0055 | -0.0151 |
| l2.over.margin.p10 | -0.1069 | -0.0810 | 0.1388 | -0.0401 | -0.0749 | -0.0488 | -0.0972 | -0.0019 |
| margin.normalized.param.norm | -0.1069 | -0.0810 | 0.1388 | -0.0401 | -0.0749 | -0.0488 | -0.0972 | -0.0019 |
| cross.entropy | 0.1602 | -0.0191 | -0.2389 | 0.0476 | -0.0248 | -0.0178 | -0.0776 | 0.0214 |
| temperature.scaling | 0.1549 | -0.0180 | -0.1735 | 0.0679 | -0.0076 | -0.0036 | -0.0292 | 0.0363 |
| aic.bias.term | 0.0000 | 0.0000 | 0.0000 | 0.0000 | 0.0000 | 0.0000 | 0.0000 | 0.0000 |
| params | 0.0000 | 0.0000 | 0.0000 | 0.0000 | 0.0000 | 0.0000 | 0.0000 | 0.0000 |
| vcdim | 0.0000 | 0.0000 | 0.0000 | 0.0000 | 0.0000 | 0.0000 | 0.0000 | 0.0000 |
| l1.over.margin.p10 | 0.0284 | -0.0371 | 0.1952 | -0.0570 | 0.0674 | 0.0104 | -0.0595 | 0.0056 |
| negative.entropy | -0.1751 | 0.0097 | 0.2278 | -0.0501 | 0.0242 | 0.0115 | 0.0733 | -0.0228 |
| pac.bayes.magflat | -0.0696 | 0.0195 | -0.1888 | 0.0589 | -0.0670 | 0.0322 | -0.0275 | 0.0080 |
| ece | 0.2068 | 0.0254 | -0.1810 | 0.0761 | 0.0320 | 0.0377 | -0.0292 | 0.0443 |
| ace | 0.2070 | 0.0256 | -0.1763 | 0.0760 | 0.0308 | 0.0384 | -0.0283 | 0.0439 |
| mce | 0.2139 | 0.0235 | -0.0822 | 0.0526 | 0.0916 | 0.0489 | 0.0282 | 0.0464 |
| reliability.diagram | 0.2253 | 0.0239 | -0.0710 | 0.0548 | 0.1032 | 0.0516 | 0.0356 | 0.0450 |
| fisher.rao.norm | 0.2487 | 0.0555 | -0.1541 | 0.0744 | 0.0786 | 0.0751 | -0.0263 | 0.0568 |
| gradient.norm | 0.3044 | 0.0520 | -0.1897 | 0.0687 | 0.1276 | 0.0827 | 0.0018 | 0.0469 |
| gradient.noise.var | 0.3143 | 0.0472 | -0.1818 | 0.0710 | 0.1365 | 0.0827 | -0.0078 | 0.0482 |
| gradient.noise.final.var | 0.3145 | 0.0476 | -0.1818 | 0.0707 | 0.1364 | 0.0829 | -0.0071 | 0.0476 |
| flatness.proxy | 0.3159 | 0.0474 | -0.1826 | 0.0686 | 0.1364 | 0.0831 | -0.0046 | 0.0476 |
| waic.bias.term | 0.1196 | 0.0296 | -0.0208 | 0.0217 | -0.0439 | 0.0952 | -0.0981 | 0.0483 |
| gradient.noise.scale | 0.1735 | 0.1134 | 0.0302 | 0.0082 | 0.1454 | 0.1011 | 0.0448 | 0.0135 |
| input.gradient.norm | 0.3274 | 0.0874 | -0.1755 | 0.0811 | 0.1446 | 0.1190 | -0.0014 | 0.0493 |
| tic.bias.term | 0.3673 | 0.1083 | -0.1889 | 0.0708 | 0.1919 | 0.1461 | 0.0057 | 0.0613 |
| sharpness.magnitude.init | 0.3567 | 0.0208 | 0.1253 | -0.0058 | 0.4116 | 0.1571 | 0.1546 | 0.0186 |
| inverse.margin.p10 | 0.4279 | 0.1438 | -0.1880 | 0.0836 | 0.2885 | 0.1643 | 0.0272 | 0.0662 |
| sharpness.magnitude | 0.3844 | 0.0360 | 0.1538 | -0.0191 | 0.4412 | 0.1710 | 0.1651 | 0.0122 |
| hessian.trace | 0.3600 | 0.1305 | -0.1850 | 0.0751 | 0.2665 | 0.1792 | 0.0463 | 0.0600 |
| hessian.top.eigenvalue | 0.3558 | 0.1214 | -0.1834 | 0.0688 | 0.2814 | 0.1849 | 0.0356 | 0.0561 |
| sharpness.magflat | 0.4470 | 0.0943 | 0.1277 | 0.0069 | 0.5071 | 0.2182 | 0.1716 | 0.0431 |
| sharpness | 0.3899 | 0.1261 | -0.1325 | 0.0663 | 0.4265 | 0.2350 | 0.1725 | 0.0381 |
| adaptive.sharpness | 0.4603 | 0.1723 | -0.0317 | 0.0429 | 0.5249 | 0.2686 | 0.2212 | 0.0281 |

Table 22: simplecnn summary of IID correlations and OODTestGap_P OOD sensitivities for each measure.

| Measure | $lr^{IID}$ | $lr^{OOD}$ | $wd^{IID}$ | $wd^{OOD}$ | $\tau^{IID}$ | $\tau^{OOD}$ | $\Psi^{IID}$ | $\Psi^{OOD}$ |
|---|---|---|---|---|---|---|---|---|
| spec.prod | -0.3307 | -0.4314 | 0.2325 | -0.2921 | -0.2764 | -0.3975 | -0.0727 | -0.0637 |
| frobenius.distance | -0.3391 | -0.4240 | 0.1880 | -0.2651 | -0.3146 | -0.3895 | -0.0768 | -0.0829 |
| pac.bayes.magnitude.init | -0.3238 | -0.4181 | 0.1872 | -0.2460 | -0.3330 | -0.3881 | -0.1252 | -0.0423 |
| path.norm | -0.2206 | -0.4091 | 0.2134 | -0.2992 | -0.1524 | -0.3379 | -0.0767 | -0.0609 |
| spec.sum | -0.3191 | -0.4383 | 0.2174 | -0.2873 | -0.1973 | -0.3293 | -0.0353 | -0.1005 |
| spectral.norm.per.layer | -0.3191 | -0.4393 | 0.2174 | -0.2873 | -0.1971 | -0.3291 | -0.0345 | -0.1022 |
| magnitude | -0.1921 | -0.3800 | 0.2087 | -0.3008 | -0.0421 | -0.2190 | 0.0055 | -0.1257 |
| pac.bayes.magnitude | -0.1921 | -0.3800 | 0.2087 | -0.3008 | -0.0421 | -0.2190 | 0.0055 | -0.1257 |
| pac.bayes.bound | -0.2851 | -0.3306 | 0.1015 | -0.2906 | -0.1772 | -0.1657 | -0.1616 | -0.0771 |
| l2.over.margin.p10 | -0.1069 | -0.1977 | 0.1388 | -0.1573 | -0.0749 | -0.1019 | -0.0972 | 0.0192 |
| margin.normalized.param.norm | -0.1069 | -0.1977 | 0.1388 | -0.1573 | -0.0749 | -0.1019 | -0.0972 | 0.0192 |
| negative.entropy | -0.1751 | -0.1597 | 0.2278 | -0.2824 | 0.0242 | -0.0950 | 0.0733 | -0.1789 |
| aic.bias.term | 0.0000 | 0.0000 | 0.0000 | 0.0000 | 0.0000 | 0.0000 | 0.0000 | 0.0000 |
| params | 0.0000 | 0.0000 | 0.0000 | 0.0000 | 0.0000 | 0.0000 | 0.0000 | 0.0000 |
| vcdim | 0.0000 | 0.0000 | 0.0000 | 0.0000 | 0.0000 | 0.0000 | 0.0000 | 0.0000 |
| l1.over.margin.p10 | 0.0284 | -0.0968 | 0.1952 | -0.2587 | 0.0674 | 0.0008 | -0.0595 | 0.0229 |
| cross.entropy | 0.1602 | 0.1296 | -0.2389 | 0.2697 | -0.0248 | 0.0801 | -0.0776 | 0.1744 |
| temperature.scaling | 0.1549 | 0.0998 | -0.1735 | 0.2622 | -0.0076 | 0.0859 | -0.0292 | 0.1786 |
| mce | 0.2139 | 0.0710 | -0.0822 | 0.2172 | 0.0916 | 0.0954 | 0.0282 | 0.1731 |
| reliability.diagram | 0.2253 | 0.0748 | -0.0710 | 0.2188 | 0.1032 | 0.1000 | 0.0356 | 0.1660 |
| gradient.noise.scale | 0.1735 | 0.2236 | 0.0302 | -0.0086 | 0.1454 | 0.1332 | 0.0448 | -0.0097 |
| pac.bayes.magflat | -0.0696 | 0.1594 | -0.1888 | 0.2762 | -0.0670 | 0.1376 | -0.0275 | 0.0948 |
| ece | 0.2068 | 0.1780 | -0.1810 | 0.3038 | 0.0320 | 0.1435 | -0.0292 | 0.1927 |
| ace | 0.2070 | 0.1792 | -0.1763 | 0.3054 | 0.0308 | 0.1451 | -0.0283 | 0.1931 |
| waic.bias.term | 0.1196 | 0.1389 | -0.0208 | 0.0777 | -0.0439 | 0.1680 | -0.0981 | 0.2103 |
| fisher.rao.norm | 0.2487 | 0.1792 | -0.1541 | 0.2888 | 0.0786 | 0.1707 | -0.0263 | 0.2049 |
| gradient.noise.var | 0.3143 | 0.2954 | -0.1818 | 0.2911 | 0.1365 | 0.2491 | -0.0078 | 0.1964 |
| gradient.noise.final.var | 0.3145 | 0.2972 | -0.1818 | 0.2927 | 0.1364 | 0.2494 | -0.0071 | 0.1954 |
| flatness.proxy | 0.3159 | 0.2968 | -0.1826 | 0.2888 | 0.1364 | 0.2496 | -0.0046 | 0.1951 |
| input.gradient.norm | 0.3274 | 0.2829 | -0.1755 | 0.3110 | 0.1446 | 0.2508 | -0.0014 | 0.1526 |
| gradient.norm | 0.3044 | 0.3129 | -0.1897 | 0.2999 | 0.1276 | 0.2513 | 0.0018 | 0.1964 |
| inverse.margin.p10 | 0.4279 | 0.3338 | -0.1880 | 0.3151 | 0.2885 | 0.2814 | 0.0272 | 0.2088 |
| tic.bias.term | 0.3673 | 0.3676 | -0.1889 | 0.2999 | 0.1919 | 0.3153 | 0.0057 | 0.2260 |
| sharpness.magnitude.init | 0.3567 | 0.1362 | 0.1253 | -0.0499 | 0.4116 | 0.3692 | 0.1546 | 0.0649 |
| hessian.trace | 0.3600 | 0.4504 | -0.1850 | 0.3023 | 0.2665 | 0.3941 | 0.0463 | 0.2152 |
| sharpness.magnitude | 0.3844 | 0.1824 | 0.1538 | -0.1085 | 0.4412 | 0.3981 | 0.1651 | 0.0337 |
| hessian.top.eigenvalue | 0.3558 | 0.4365 | -0.1834 | 0.2729 | 0.2814 | 0.4126 | 0.0356 | 0.2082 |
| sharpness.magflat | 0.4470 | 0.2764 | 0.1277 | -0.0349 | 0.5071 | 0.4586 | 0.1716 | 0.1118 |
| sharpness | 0.3899 | 0.4597 | -0.1325 | 0.2675 | 0.4265 | 0.5465 | 0.1725 | 0.1307 |
| adaptive.sharpness | 0.4603 | 0.4828 | -0.0317 | 0.1333 | 0.5249 | 0.5712 | 0.2212 | 0.0594 |

## C.2 CIFAR-10 Local Sign-Error Analyses

Figures 6–10 extend the main-text NiN analysis to three architectures and five clean or shifted gap targets. The clean-to-shifted targets expose additional local ranking failures under the stricter degradation objective.

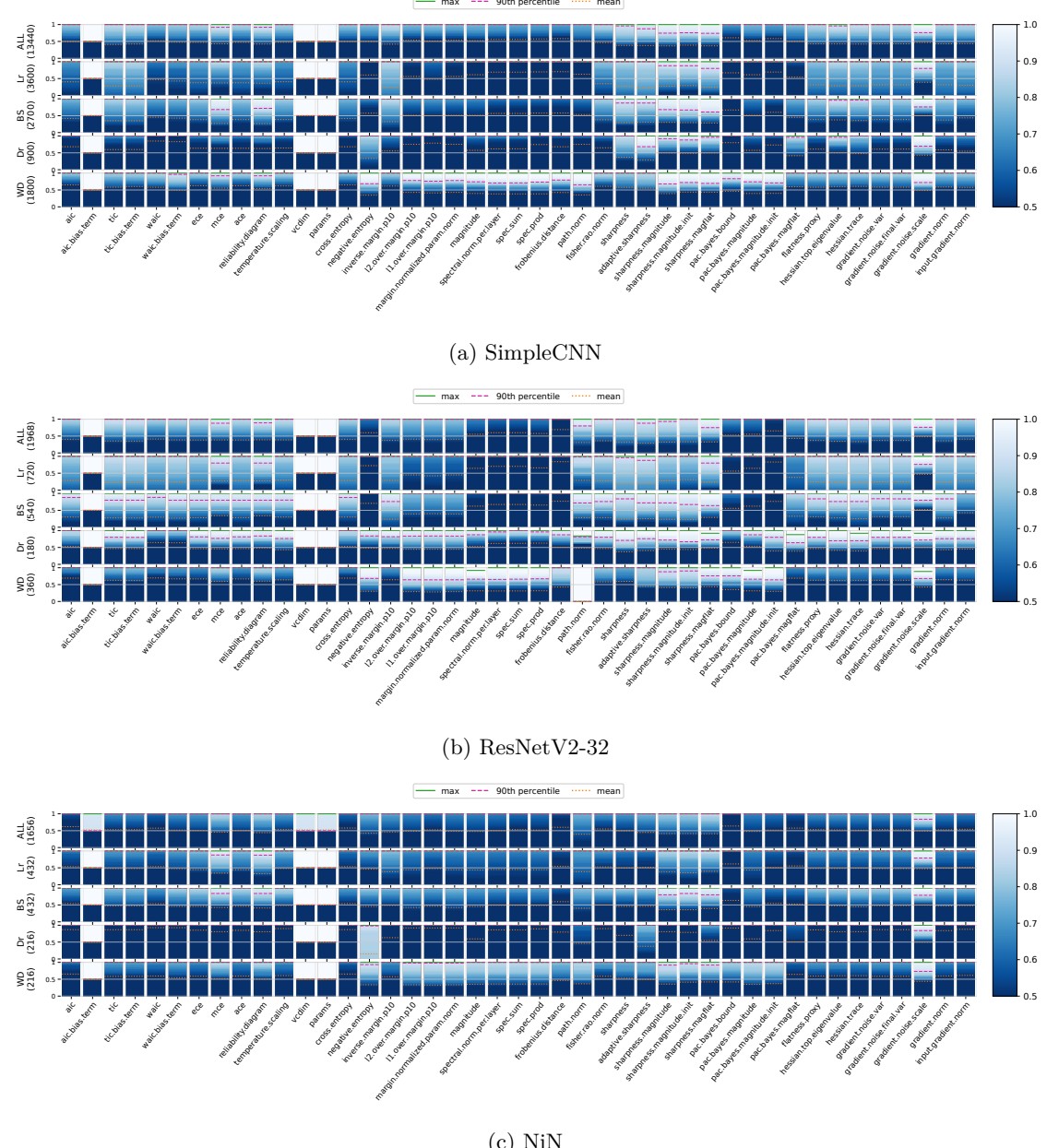

(a) SimpleCNN

(b) ResNetV2-32

(c) NiN

Figure 6: Sign-error distributions for the IID CIFAR-10 generalization gap. The target is `GenGap_CIFAR10`.

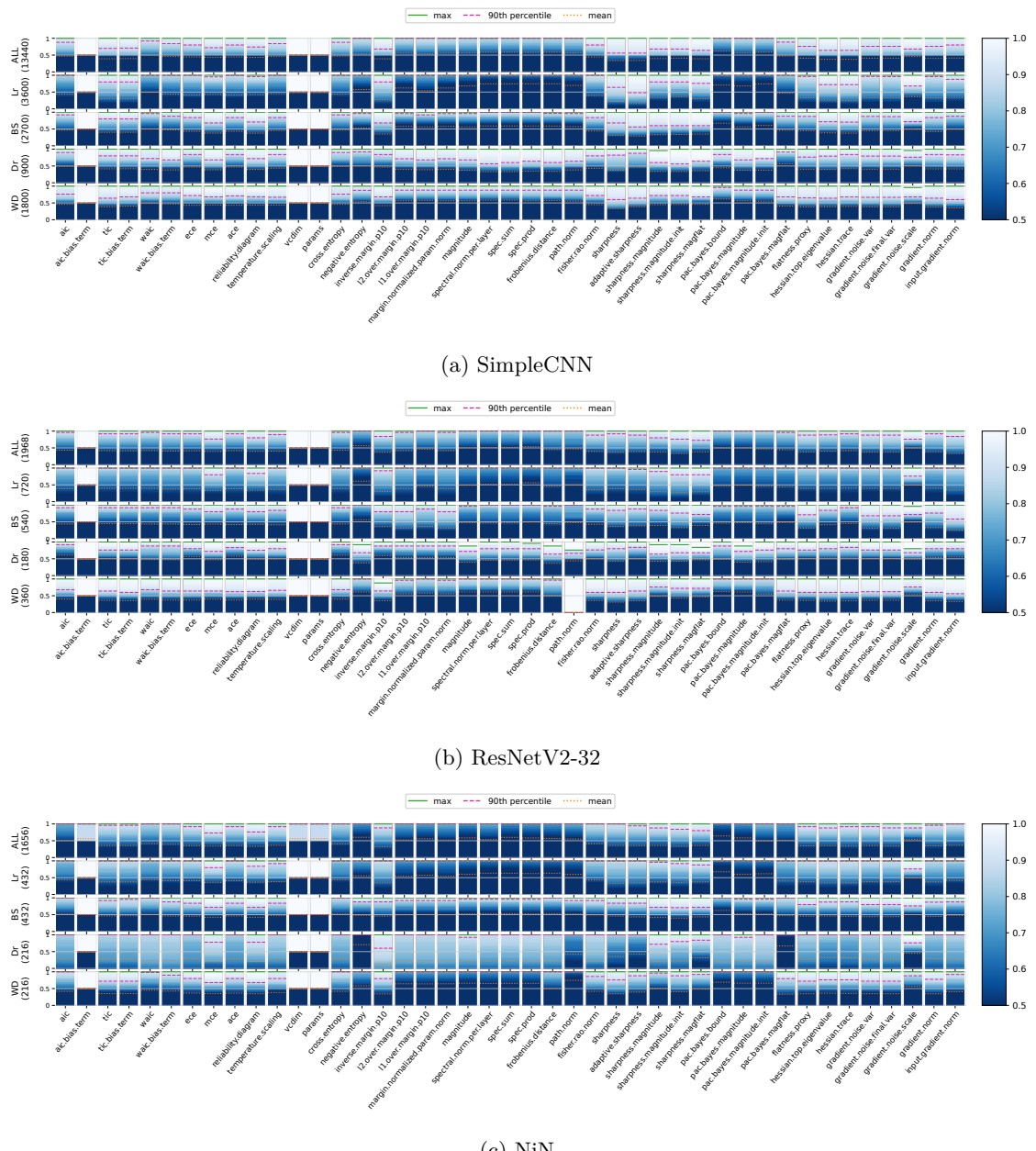

(a) SimpleCNN

(b) ResNetV2-32

(c) NiN

Figure 7: Sign-error distributions for CIFAR-10-C train-to-shifted-test gap. The target is `OODGenGap_C`, computed from CIFAR-10 training accuracy to CIFAR-10-C test accuracy.

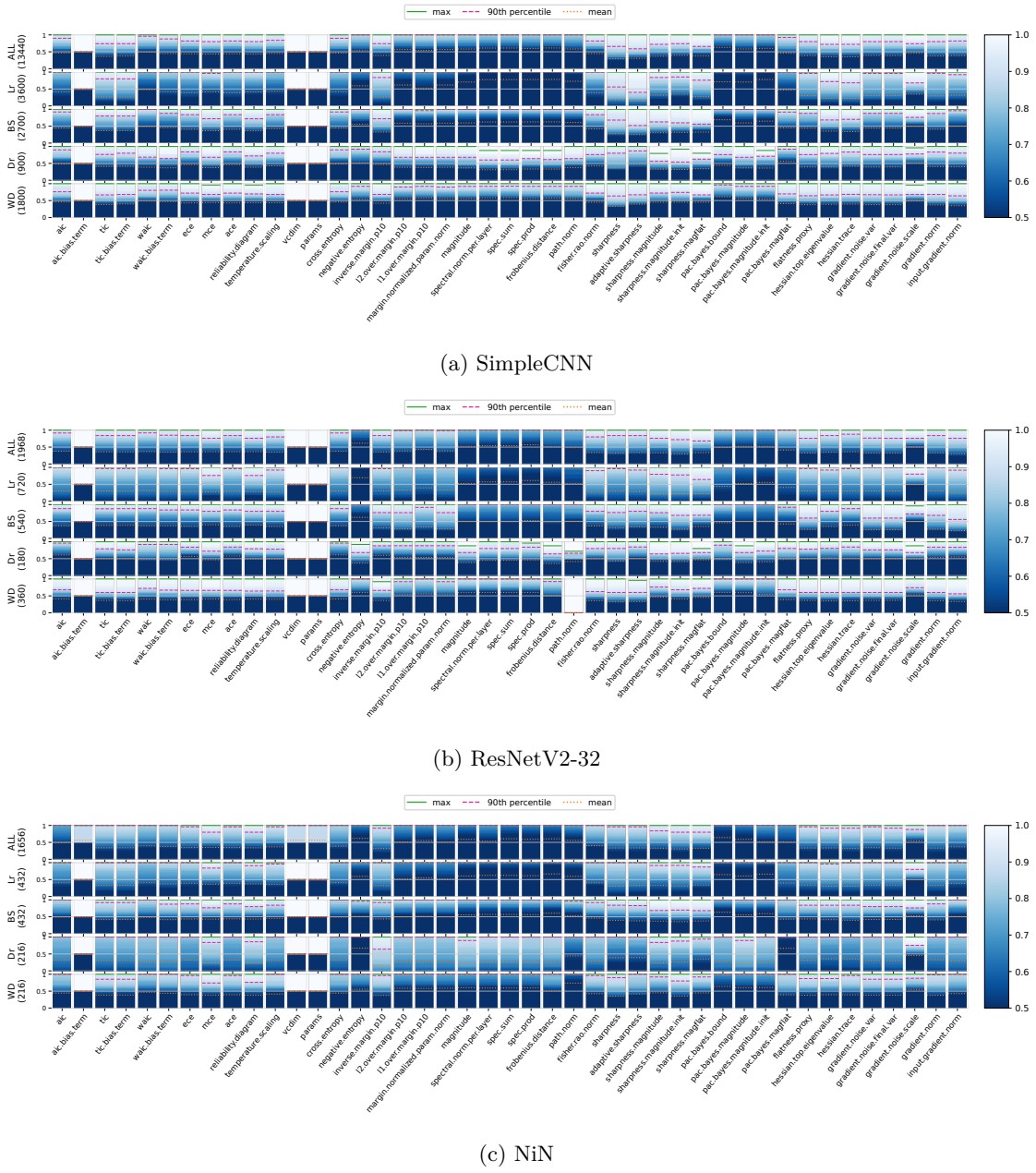

(a) SimpleCNN

(b) ResNetV2-32

(c) NiN

Figure 8: Sign-error distributions for CIFAR-10-P train-to-shifted-test gap. The target is `OODGenGap_P`, computed from CIFAR-10 training accuracy to CIFAR-10-P test accuracy. The NiN panel is the CIFAR-10-P counterpart to Figure 2.

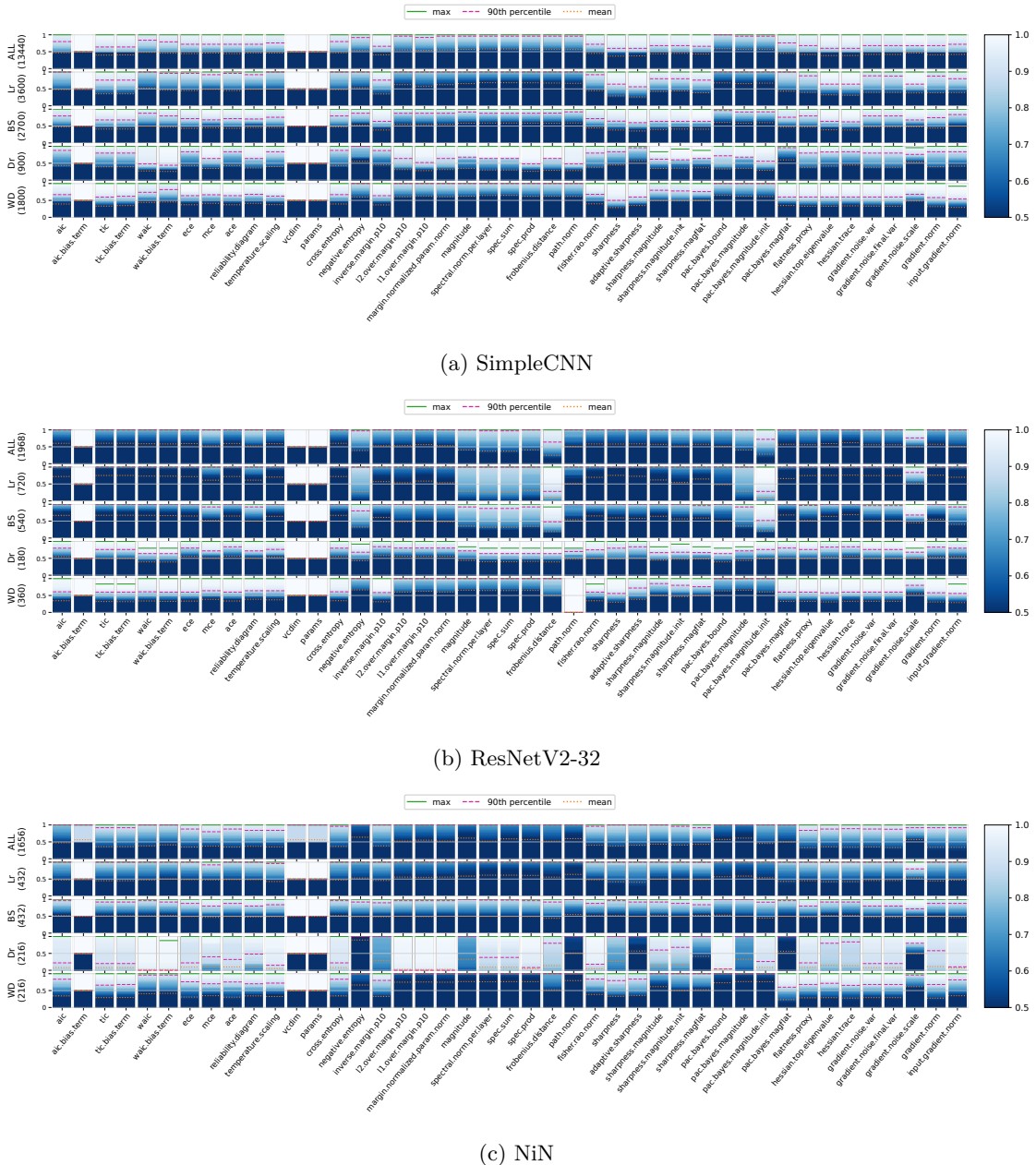

(a) SimpleCNN

(b) ResNetV2-32

(c) NiN

Figure 9: Sign-error distributions for CIFAR-10-C test degradation. The target is `OODTestGap_C`, computed from clean CIFAR-10 test accuracy to CIFAR-10-C test accuracy.

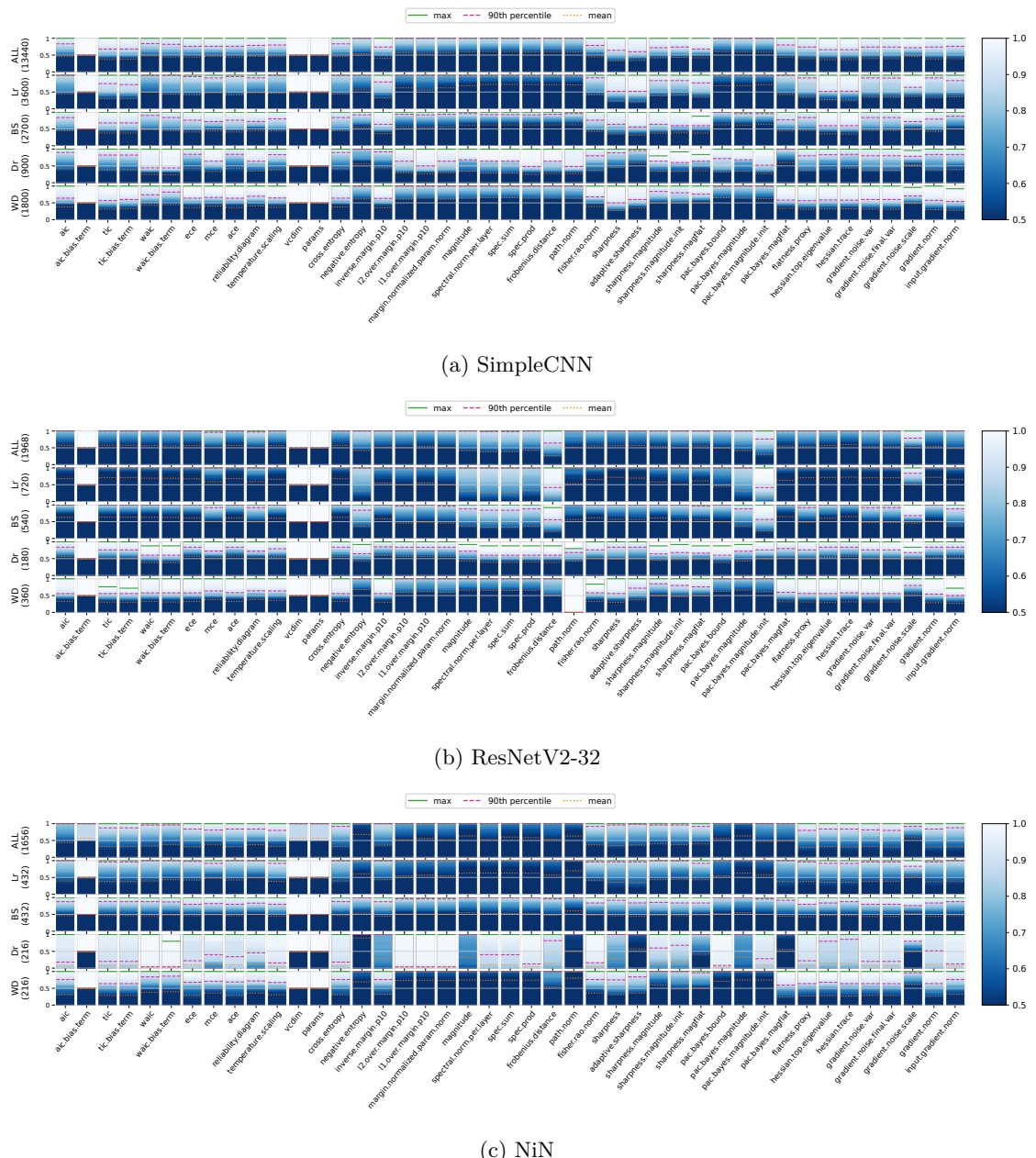

(a) SimpleCNN

(b) ResNetV2-32

(c) NiN

Figure 10: Sign-error distributions for CIFAR-10-P test degradation. The target is `OODTestGap_P`, computed from clean CIFAR-10 test accuracy to CIFAR-10-P test accuracy.

### C.3 Decision-Level Model Selection Analyses

#### C.3.1 Protocol details

This section gives the implementation details for pre-target selection of low held-out `OODGenGap_C/P` summarized in Section 3.6.

**Candidate sets and filtering.** A candidate set $C_{a,s,t}$ is indexed by architecture $a$, sweep identity $s$, and shifted target $t \in \{\text{CIFAR-10-C}, \text{CIFAR-10-P}\}$. It contains finished runs that share $(a, s, t)$, have finite selector and target values, and satisfy `cross.entropy` $\leq 0.01$. This is an analysis restriction rather than a definition of convergence; because cross-entropy is also a selector, all rankings are conditional on the restricted range. The analysis contains 26 candidate sets, 13 for each target.

**Selectors and leakage check.** Generalization-measure selectors use only pre-target measure values, with direction fixed a priori. Source/clean baselines use training accuracy/loss and clean validation accuracy/loss. Accuracy-based selectors and negative.entropy are maximized; all other selectors are minimized. Clean-test accuracy/loss are excluded from this baseline and reported separately. Random samples uniformly from the candidate set. Oracle OOD is the run with minimum `final/OODGenGap_C` or `final/OODGenGap_P` and is included only as a reference.

**Source metric provenance.** The source training split is the 80% CIFAR-10 training subset produced by the run seed; the clean validation split is the remaining 20%. `cross.entropy` is read from the `cross_entropy` measure output and is also the eligibility filter. `train.loss` and `train.acc` use their final training-loop aliases. `clean.val.acc` and `clean.val.loss` use the canonical `cifar10/val_acc` and `cifar10/val_loss` columns recovered from the logged validation summaries. In Appendix C.4, `source accuracy` is read from `final/cifar10/train_acc_eval`; `source error` is $1 -$ source accuracy after percent-to-fraction conversion.

**Metrics, family-level aggregation, and confidence intervals.** Regret and normalized regret follow Section 3.6; normalized regret is zero when the oracle-to-worst range is zero. A top-$k$ hit means that the selected run lies among the $\lceil k|C|/100 \rceil$ smallest held-out target values for $k \in \{10, 20\}$. Family results average available individual-selector outcomes within each candidate set and then across the 26 sets. The six families contain 5, 10, 12, 5, 3, and 5 selectors, respectively, and the Source/clean group contains 4; `path.norm` has support on 22 sets rather than 26. These summaries therefore depend on family membership and support and do not define an ensemble or unseen-member guarantee. Candidate-set bootstrap intervals are conditional on the observed benchmark and do not model dependence from shared architectures, sweeps, or runs.

#### C.3.2 Family-level results and clean-test references

Table 24 reports clean-test accuracy/loss as separate references; they remain excluded from the main Source/clean baseline.

Table 23: Decision-level selection for minimum OOD generalization gap on CIFAR-10-C/P. The table reports family-level averages over pre-target selectors. For each family and candidate set, metrics from the available individual selectors are averaged within the family, and the resulting candidate-set values are then averaged over 26 candidate sets. Source/clean baselines exclude clean-test quantities; clean-test references are reported in Appendix C.3.1. Lower normalized regret is better; higher top-$k$ hit rates are better.

| Family / reference | Mean normalized regret ↓ | 95% CI | Top-10% hit ↑ | Top-20% hit ↑ | # candidate sets |
|---|---|---|---|---|---|
| Oracle | 0.000 | [0.000, 0.000] | 1.000 | 1.000 | 26 |
| Calibration & Confidence | 0.223 | [0.158, 0.295] | 0.323 | 0.469 | 26 |
| Source/clean baselines | 0.283 | [0.229, 0.336] | 0.260 | 0.442 | 26 |
| Optimization-based | 0.221 | [0.173, 0.270] | 0.346 | 0.446 | 26 |
| Sharpness-based | 0.291 | [0.245, 0.340] | 0.288 | 0.404 | 26 |
| Information Criteria | 0.242 | [0.192, 0.294] | 0.282 | 0.423 | 26 |
| Baseline & Output-based | 0.334 | [0.275, 0.395] | 0.215 | 0.354 | 26 |
| Norm & Margin-based | 0.504 | [0.435, 0.574] | 0.105 | 0.174 | 26 |
| Random | 0.394 | [0.361, 0.426] | 0.107 | 0.203 | 26 |

Note: These are descriptive blind-family-commitment summaries for the included selectors, not selectors, ensembles, or robustness estimates for unseen family members. The measure-family member counts are 5 Baseline & Output-based, 10 Norm & Margin-based, 12 Sharpness-based, 5 Optimization-based, 3 Information Criteria, and 5 Calibration & Confidence; Source/clean baselines contain 4 selectors. Unequal membership affects family averages and observed spreads. Most displayed individual selectors have support on 26 candidate sets, but `path.norm` has support on 22; the aggregate table does not retain the per-set denominator or reason for missingness. Intervals are candidate-set bootstrap summaries conditional on the observed benchmark and do not model shared architecture/sweep/run dependence.

Table 24: **Source/clean baseline variants on CIFAR-10-C/P.** The main decision-selection table excludes clean-test quantities from Source/clean baselines. Clean-test accuracy/loss are shown only as information-rich source-domain references because they use the clean test split, although they still do not use CIFAR-10-C/P target labels or OOD metrics.

| Selector group | Included selectors | Mean normalized regret ↓ | 95% CI | Top-10% hit ↑ | Top-20% hit ↑ | # candidate sets |
|---|---|---|---|---|---|---|
| Source/clean baselines | train_acc, train_loss, clean_val_acc, clean_val_loss | 0.283 | [0.229, 0.337] | 0.260 | 0.442 | 26 |
| Clean-test optimistic reference | clean_test_acc, clean_test_loss | 0.326 | [0.234, 0.419] | 0.250 | 0.308 | 26 |

Table 25: Direct source-statistics ablation for OODGenGap-based decision selection on common candidate-set support.

| Comparison | $\Delta$ normalized regret | 95% paired CI | Candidate sets |
|---|---|---|---|
| Cross-entropy − Calibration family | -0.028 | [-0.059, -0.001] | 26 |
| Negative entropy − Calibration family | -0.028 | [-0.059, -0.001] | 26 |
| Cross-entropy − ECE | -0.011 | [-0.023, -0.003] | 26 |
| Negative entropy − ECE | -0.011 | [-0.023, -0.003] | 26 |

Note: Differences are source statistic minus calibration comparator on their paired common observed support; negative values favor the source statistic. The requirement `cross.entropy` $\leq 0.01$ is an analysis restriction with no available provenance and restricts the evaluated range of the cross-entropy selector. Intervals are candidate-set bootstrap summaries conditional on this observed benchmark and do not model shared architecture/sweep/run dependence.

Table 26: Cross-entropy threshold sensitivity for the direct source-statistics ablation.

| Cross-entropy threshold | Eligible memberships | Cross-entropy − Calibration family | Cross-entropy − ECE |
|---|---|---|---|
| 0.005 | 18,182 | -0.032 [-0.065, -0.002] | -0.011 [-0.023, -0.003] |
| 0.010 | 19,362 | -0.028 [-0.059, -0.001] | -0.011 [-0.023, -0.003] |
| 0.020 | 20,068 | -0.028 [-0.057, -0.002] | -0.011 [-0.023, -0.003] |

Entries are mean paired differences in normalized regret with 95% candidate-set bootstrap intervals. All thresholds retain the same 26 candidate sets. Cross-entropy and negative entropy select the same run in all 26 sets at every threshold, so their contrasts are identical. Eligible run memberships count runs within candidate sets and count membership in the CIFAR-10-C and CIFAR-10-P sets separately.

### C.3.3 Direct Source-Statistics Ablation on Common Support

This ablation compares source cross-entropy and negative entropy with the Calibration & Confidence family average and ECE on common candidate-set support; negative differences favor the source statistic.

Both source statistics have point-estimate differences of -0.028 relative to the Calibration family average and -0.011 relative to ECE; the paired intervals are shown in Table 25.

Table 26 repeats the ablation at cross-entropy thresholds of 0.005 and 0.02. The 26 candidate sets are unchanged, while eligible run memberships range from 18,182 to 20,068. The cross-entropy contrasts remain negative with similar magnitudes and paired intervals at all three thresholds. Cross-entropy and negative entropy select the same run in all 26 candidate sets at each threshold, so their contrasts are identical. The ablation conclusion is therefore robust across the tested thresholds, while remaining conditional on the restricted candidate pools.

### C.3.4 Individual selectors and architecture checks

Table 27 gives the full individual ranking; `path.norm` uses 22 candidate sets rather than 26 and therefore has unequal support.

Table 27: Decision-level selection for minimum OOD generalization gap on CIFAR-10-C/P: individual selectors.

| Selector | Mean normalized regret ↓ | 95%CI | Top-20% hit ↑ | #candidate |
|---|---|---|---|---|
| oracle.ood | 0.000 | [0.000, 0.000] | 1.000 | 26 |
| sharpness.magnitude.init | 0.132 | [0.084, 0.188] | 0.577 | 26 |
| input.gradient.norm | 0.135 | [0.088, 0.189] | 0.654 | 26 |
| sharpness | 0.144 | [0.097, 0.196] | 0.615 | 26 |
| sharpness.magnitude | 0.148 | [0.094, 0.213] | 0.615 | 26 |
| gradient.noise.final.var | 0.160 | [0.111, 0.216] | 0.423 | 26 |
| gradient.noise.var | 0.160 | [0.110, 0.215] | 0.423 | 26 |
| sharpness.magflat | 0.165 | [0.118, 0.221] | 0.500 | 26 |
| hessian.trace | 0.170 | [0.116, 0.232] | 0.615 | 26 |
| tic.bias.term | 0.175 | [0.107, 0.250] | 0.462 | 26 |
| flatness.proxy | 0.176 | [0.108, 0.255] | 0.500 | 26 |
| gradient.norm | 0.183 | [0.115, 0.261] | 0.462 | 26 |
| fisher.rao.norm | 0.185 | [0.121, 0.255] | 0.462 | 26 |
| hessian.top.eigenvalue | 0.192 | [0.139, 0.250] | 0.423 | 26 |
| cross.entropy | 0.195 | [0.122, 0.275] | 0.500 | 26 |
| negative.entropy | 0.195 | [0.124, 0.275] | 0.500 | 26 |
| inverse.margin.p10 | 0.195 | [0.131, 0.264] | 0.577 | 26 |
| waic.bias.term | 0.200 | [0.136, 0.276] | 0.462 | 26 |
| ece | 0.206 | [0.138, 0.283] | 0.423 | 26 |
| ace | 0.212 | [0.141, 0.290] | 0.500 | 26 |
| reliability.diagram | 0.212 | [0.134, 0.301] | 0.500 | 26 |
| mce | 0.219 | [0.145, 0.305] | 0.423 | 26 |
| adaptive.sharpness | 0.243 | [0.163, 0.325] | 0.423 | 26 |
| temperature.scaling | 0.265 | [0.175, 0.365] | 0.500 | 26 |
| pac.bayes.magflat | 0.330 | [0.227, 0.434] | 0.423 | 26 |
| aic.bias.term | 0.351 | [0.253, 0.452] | 0.346 | 26 |
| params | 0.351 | [0.253, 0.451] | 0.346 | 26 |
| vcdim | 0.351 | [0.254, 0.452] | 0.346 | 26 |
| path.norm | 0.435 | [0.323, 0.552] | 0.136 | 22 |
| gradient.noise.scale | 0.465 | [0.356, 0.571] | 0.269 | 26 |
| spectral.norm.per.layer | 0.509 | [0.395, 0.625] | 0.192 | 26 |
| l1.over.margin.p10 | 0.544 | [0.431, 0.654] | 0.115 | 26 |
| l2.over.margin.p10 | 0.550 | [0.442, 0.657] | 0.077 | 26 |
| margin.normalized.param.norm | 0.550 | [0.442, 0.661] | 0.077 | 26 |
| pac.bayes.bound | 0.556 | [0.450, 0.663] | 0.077 | 26 |
| magnitude | 0.581 | [0.468, 0.696] | 0.077 | 26 |
| pac.bayes.magnitude | 0.581 | [0.466, 0.696] | 0.077 | 26 |
| spec.prod | 0.624 | [0.516, 0.723] | 0.077 | 26 |
| spec.sum | 0.658 | [0.555, 0.755] | 0.038 | 26 |
| pac.bayes.magnitude.init | 0.659 | [0.573, 0.748] | 0.000 | 26 |
| frobenius.distance | 0.753 | [0.671, 0.831] | 0.000 | 26 |
| clean.val.acc | 0.178 | [0.111, 0.255] | 0.577 | 26 |
| train.loss | 0.180 | [0.120, 0.250] | 0.615 | 26 |
| train.acc | 0.341 | [0.261, 0.425] | 0.346 | 26 |
| clean.val.loss | 0.435 | [0.302, 0.573] | 0.231 | 26 |
| random | 0.394 | [0.360, 0.426] | 0.203 | 26 |

Note: `path.norm` is evaluated on 22 candidate sets; every other displayed individual selector is evaluated on 26. Comparisons involving `path.norm` therefore have unequal support.

Tables 28–30 show that the lowest-regret family differs across the three architectures.

Table 28: Decision-level selection for minimum CIFAR-10-C/P OOD generalization gap for SimpleCNN.

| Selector family | Mean normalized regret ↓ | 95% CI | Top-10% hit ↑ | Top-20% hit ↑ | # candidate sets | # runs |
|---|---|---|---|---|---|---|
| Oracle | 0.000 | [0.000, 0.000] | 1.000 | 1.000 | 12 | 13700 |
| Calibration & Confidence | 0.217 | [0.112, 0.336] | 0.383 | 0.517 | 12 | 13700 |
| Source/clean baselines | 0.221 | [0.142, 0.307] | 0.333 | 0.562 | 12 | 13700 |
| Optimization-based | 0.180 | [0.103, 0.268] | 0.467 | 0.583 | 12 | 13700 |
| Sharpness-based | 0.254 | [0.174, 0.347] | 0.382 | 0.521 | 12 | 13700 |
| Information Criteria | 0.173 | [0.097, 0.266] | 0.417 | 0.639 | 12 | 13700 |
| Baseline & Output-based | 0.229 | [0.152, 0.324] | 0.300 | 0.533 | 12 | 13700 |
| Norm & Margin-based | 0.442 | [0.323, 0.572] | 0.158 | 0.225 | 12 | 13700 |
| Random | 0.389 | [0.323, 0.450] | 0.111 | 0.210 | 12 | 13700 |

Note: The table reports family-level averages over pre-target selectors for a fixed architecture. For each candidate set, selector metrics are averaged within a family, and the resulting candidate-set values are averaged over candidate sets. Bootstrap confidence intervals resample candidate sets. Lower normalized regret is better. Higher top-k hit rates are better.

Table 29: Decision-level selection for minimum CIFAR-10-C/P OOD generalization gap for ResNetV2-32.

| Selector family | Mean normalized regret ↓ | 95% CI | Top-10% hit ↑ | Top-20% hit ↑ | # candidate sets | # runs |
|---|---|---|---|---|---|---|
| Oracle | 0.000 | [0.000, 0.000] | 1.000 | 1.000 | 6 | 2696 |
| Calibration & Confidence | 0.350 | [0.224, 0.483] | 0.067 | 0.067 | 6 | 2696 |
| Source/clean baselines | 0.325 | [0.266, 0.383] | 0.167 | 0.250 | 6 | 2696 |
| Optimization-based | 0.307 | [0.225, 0.389] | 0.133 | 0.133 | 6 | 2696 |
| Sharpness-based | 0.338 | [0.269, 0.393] | 0.111 | 0.167 | 6 | 2696 |
| Information Criteria | 0.332 | [0.262, 0.402] | 0.056 | 0.111 | 6 | 2696 |
| Baseline & Output-based | 0.394 | [0.318, 0.461] | 0.000 | 0.000 | 6 | 2696 |
| Norm & Margin-based | 0.536 | [0.459, 0.588] | 0.000 | 0.000 | 6 | 2696 |
| Random | 0.364 | [0.330, 0.402] | 0.101 | 0.204 | 6 | 2696 |

Note: The table reports family-level averages over pre-target selectors for a fixed architecture. For each candidate set, selector metrics are averaged within a family, and the resulting candidate-set values are averaged over candidate sets. Bootstrap confidence intervals resample candidate sets. Lower normalized regret is better. Higher top-k hit rates are better.

Table 30: Decision-level selection for minimum CIFAR-10-C/P OOD generalization gap for NiN.

| Selector family | Mean normalized regret ↓ | 95% CI | Top-10% hit ↑ | Top-20% hit ↑ | # candidate sets | # runs |
|---|---|---|---|---|---|---|
| Oracle | 0.000 | [0.000, 0.000] | 1.000 | 1.000 | 8 | 2966 |
| Calibration & Confidence | 0.136 | [0.097, 0.179] | 0.425 | 0.700 | 8 | 2966 |
| Source/clean baselines | 0.346 | [0.243, 0.425] | 0.219 | 0.406 | 8 | 2966 |
| Optimization-based | 0.217 | [0.175, 0.257] | 0.325 | 0.475 | 8 | 2966 |
| Sharpness-based | 0.312 | [0.262, 0.361] | 0.281 | 0.406 | 8 | 2966 |
| Information Criteria | 0.277 | [0.234, 0.323] | 0.250 | 0.333 | 8 | 2966 |
| Baseline & Output-based | 0.448 | [0.407, 0.494] | 0.250 | 0.350 | 8 | 2966 |
| Norm & Margin-based | 0.573 | [0.503, 0.655] | 0.103 | 0.228 | 8 | 2966 |
| Random | 0.425 | [0.406, 0.446] | 0.105 | 0.193 | 8 | 2966 |

Note: The table reports family-level averages over pre-target selectors for a fixed architecture. For each candidate set, selector metrics are averaged within a family, and the resulting candidate-set values are averaged over candidate sets. Bootstrap confidence intervals resample candidate sets. Lower normalized regret is better. Higher top-k hit rates are better.

Table 31 reports observed max–min spread among included selectors; unequal, design-dependent membership precludes an unseen-member interpretation.

Table 32 gives the three architectures equal weight.

Table 31: Observed maximum-minus-minimum normalized-regret spread among included individual selectors for OODGenGap-based decision selection.

| Family | min | max | spread |
|---|---|---|---|
| Calibration & Confidence | 0.206 (ece) | 0.265 (temperature.scaling) | 0.059 |
| Optimization-based | 0.135 (input.gradient.norm) | 0.465 (gradient.noise.scale) | 0.330 |
| Sharpness-based | 0.132 (sharpness.magnitude.init) | 0.659 (pac.bayes.magnitude.init) | 0.527 |
| Information Criteria | 0.175 (tic.bias.term) | 0.351 (aic.bias.term) | 0.176 |

Note: Min and max are computed among included selectors in Table 27. The displayed families contain 5 Calibration & Confidence, 5 Optimization-based, 12 Sharpness-based, and 3 Information Criteria selectors; all displayed members have support on 26 candidate sets. Because membership is unequal and design dependent, the observed spread is neither an uncertainty estimate nor a stability guarantee for an unseen family member. Lower normalized regret is better.

Table 32: Pooled family-level OODGenGap selection estimates and equal-architecture macro averages.

| Family | Pooled | Macro-average (3 architectures) |
|---|---|---|
| Calibration & Confidence | 0.223 | 0.234 |
| Source/clean baselines | 0.283 | 0.297 |
| Optimization-based | 0.221 | 0.235 |
| Sharpness-based | 0.291 | 0.301 |
| Information Criteria | 0.242 | 0.261 |
| Baseline & Output-based | 0.334 | 0.357 |
| Norm & Margin-based | 0.504 | 0.517 |
| Random | 0.394 | 0.393 |

Note: The macro-average gives SimpleCNN, ResNetV2-32, and NiN equal weight. Lower normalized regret is better.

### C.3.5 Severity-controlled decision results

This analysis compares CIFAR-10-C candidate sets separately at severities 1–5, fixing target and severity before selection. Tables 33–35 report individual and family regret plus adjacent-severity ranking stability.

Table 33: CIFAR-10-C severity-controlled decision results for all individual selectors. Entries are mean normalized regret within same-severity candidate sets; lower is better. CIFAR-10-P is excluded.

| Selector | S1 (n=5) | S2 (n=4) | S3 (n=10) | S4 (n=4) | S5 (n=4) |
|---|---|---|---|---|---|
| oracle.ood | 0.000 | 0.000 | 0.000 | 0.000 | 0.000 |
| sharpness | 0.138 | 0.049 | 0.138 | 0.102 | 0.103 |
| hessian.trace | 0.205 | 0.078 | 0.174 | 0.116 | 0.139 |
| hessian.top.eigenvalue | 0.212 | 0.090 | 0.178 | 0.132 | 0.155 |
| tic.bias.term | 0.140 | 0.094 | 0.210 | 0.193 | 0.213 |
| fisher.rao.norm | 0.103 | 0.302 | 0.196 | 0.129 | 0.130 |
| input.gradient.norm | 0.131 | 0.319 | 0.168 | 0.128 | 0.116 |
| temperature.scaling | 0.152 | 0.300 | 0.223 | 0.114 | 0.132 |
| flatness.proxy | 0.131 | 0.319 | 0.235 | 0.128 | 0.116 |
| waic.bias.term | 0.333 | 0.103 | 0.237 | 0.206 | 0.219 |
| pac.bayes.magflat | 0.177 | 0.072 | 0.292 | 0.149 | 0.415 |
| ace | 0.170 | 0.344 | 0.257 | 0.162 | 0.177 |
| sharpness.magnitude.init | 0.138 | 0.340 | 0.201 | 0.199 | 0.234 |
| ece | 0.153 | 0.348 | 0.240 | 0.184 | 0.194 |
| gradient.norm | 0.136 | 0.353 | 0.226 | 0.206 | 0.219 |
| gradient.noise.final.var | 0.131 | 0.368 | 0.223 | 0.208 | 0.215 |
| gradient.noise.var | 0.131 | 0.368 | 0.223 | 0.208 | 0.215 |
| cross.entropy | 0.136 | 0.353 | 0.247 | 0.206 | 0.219 |
| negative.entropy | 0.136 | 0.353 | 0.247 | 0.206 | 0.219 |
| adaptive.sharpness | 0.218 | 0.178 | 0.260 | 0.279 | 0.285 |
| sharpness.magflat | 0.295 | 0.127 | 0.248 | 0.269 | 0.315 |
| inverse.margin.p10 | 0.266 | 0.384 | 0.216 | 0.196 | 0.213 |
| reliability.diagram | 0.354 | 0.339 | 0.235 | 0.171 | 0.187 |
| sharpness.magnitude | 0.111 | 0.339 | 0.195 | 0.435 | 0.208 |
| aic.bias.term | 0.176 | 0.322 | 0.328 | 0.104 | 0.365 |
| params | 0.176 | 0.322 | 0.328 | 0.104 | 0.365 |
| vcdim | 0.176 | 0.322 | 0.328 | 0.104 | 0.365 |
| mce | 0.349 | 0.348 | 0.240 | 0.184 | 0.194 |
| gradient.noise.scale | 0.275 | 0.429 | 0.424 | 0.244 | 0.253 |
| pac.bayes.bound | 0.516 | 0.560 | 0.614 | 0.632 | 0.406 |
| magnitude | 0.516 | 0.560 | 0.643 | 0.632 | 0.406 |
| pac.bayes.magnitude | 0.516 | 0.560 | 0.643 | 0.632 | 0.406 |
| l1.over.margin.p10 | 0.447 | 0.574 | 0.655 | 0.680 | 0.475 |
| path.norm | 0.507 | 0.574 | 0.550 | 0.653 | 0.673 |
| spectral.norm.per.layer | 0.638 | 0.681 | 0.546 | 0.705 | 0.464 |
| l2.over.margin.p10 | 0.532 | 0.620 | 0.647 | 0.739 | 0.515 |
| margin.normalized.param.norm | 0.532 | 0.620 | 0.647 | 0.739 | 0.515 |
| spec.prod | 0.640 | 0.687 | 0.697 | 0.713 | 0.722 |
| spec.sum | 0.695 | 0.759 | 0.748 | 0.788 | 0.544 |
| pac.bayes.magnitude.init | 0.873 | 0.793 | 0.609 | 0.803 | 0.823 |
| frobenius.distance | 0.923 | 0.864 | 0.743 | 0.845 | 0.851 |
| train.acc | 0.399 | 0.109 | 0.332 | 0.149 | 0.171 |
| train.loss | 0.322 | 0.352 | 0.184 | 0.167 | 0.166 |
| clean.val.acc | 0.129 | 0.171 | 0.346 | 0.331 | 0.365 |
| clean.val.loss | 0.165 | 0.211 | 0.635 | 0.601 | 0.375 |

Table 34: CIFAR-10-C severity-controlled decision results for family averages. Entries are mean normalized regret within same-severity candidate sets; lower is better. CIFAR-10-P is excluded.

| Family | S1 (n=5) | S2 (n=4) | S3 (n=10) | S4 (n=4) | S5 (n=4) |
|---|---|---|---|---|---|
| Oracle | 0.000 | 0.000 | 0.000 | 0.000 | 0.000 |
| Calibration & Confidence | 0.235 | 0.336 | 0.239 | 0.163 | 0.177 |
| Source/clean baselines | 0.254 | 0.211 | 0.374 | 0.312 | 0.269 |
| Optimization-based | 0.160 | 0.367 | 0.253 | 0.199 | 0.204 |
| Sharpness-based | 0.294 | 0.292 | 0.316 | 0.323 | 0.300 |
| Information Criteria | 0.216 | 0.173 | 0.259 | 0.167 | 0.266 |
| Baseline & Output-based | 0.228 | 0.382 | 0.358 | 0.250 | 0.315 |
| Norm & Margin-based | 0.528 | 0.606 | 0.566 | 0.619 | 0.511 |
| Random | 0.335 | 0.396 | 0.414 | 0.526 | 0.508 |

Table 35: Adjacent-severity stability on CIFAR-10-C same-severity candidate sets. Spearman correlations rank non-oracle selectors or families by mean normalized regret at adjacent severities; top-$k$ overlap uses $k = 5$ for individual selectors and $k = 3$ for family averages.

| Analysis | Adjacent severities | Spearman $\rho$ | Top-$k$ overlap | Shared top-$k$ entries |
|---|---|---|---|---|
| Individual selectors | S1–S2 | 0.577 | 0/5 | – |
| Family averages | S1–S2 | 0.476 | 1/3 | Information Criteria |
| Individual selectors | S2–S3 | 0.639 | 3/5 | hessian.top.eigenvalue, hessian.trace, sharpness |
| Family averages | S2–S3 | 0.524 | 1/3 | Information Criteria |
| Individual selectors | S3–S4 | 0.719 | 1/5 | sharpness |
| Family averages | S3–S4 | 0.905 | 3/3 | Calibration & Confidence, Information Criteria, Optimization-based |
| Individual selectors | S4–S5 | 0.779 | 2/5 | sharpness, temperature.scaling |
| Family averages | S4–S5 | 0.905 | 3/3 | Calibration & Confidence, Information Criteria, Optimization-based |

| Selector | S1 (n=5) | S2 (n=4) | S3 (n=10) | S4 (n=4) | S5 (n=4) |
|---|---|---|---|---|---|
| random | 0.335 | 0.396 | 0.414 | 0.526 | 0.508 |

Individual and family orderings vary by severity; the small, unequal candidate-set counts (5, 4, 10, 4, and 4) make these results exploratory.

### C.3.6 Separate CIFAR-10-C and CIFAR-10-P decision results

CIFAR-10-C and CIFAR-10-P have different sampling structures, so Tables 36 and 37 report them separately. CIFAR-10-P remains a perturbation-sequence benchmark without sequence-block uncertainty; aggregation details appear in Appendix B.1.1.

Table 36: Decision-level selection for minimum OOD generalization gap on CIFAR-10-C: individual selectors.

| Selector | Mean normalized regret ↓ | 95%CI | Top-20% hit ↑ | #candidate |
|---|---|---|---|---|
| oracle.ood | 0.000 | [0.000, 0.000] | 1.000 | 13 |
| input.gradient.norm | 0.149 | [0.070, 0.244] | 0.692 | 13 |
| sharpness.magnitude.init | 0.160 | [0.078, 0.260] | 0.538 | 13 |
| sharpness | 0.165 | [0.093, 0.246] | 0.615 | 13 |
| sharpness.magflat | 0.181 | [0.106, 0.270] | 0.462 | 13 |
| sharpness.magnitude | 0.182 | [0.090, 0.288] | 0.538 | 13 |
| gradient.noise.final.var | 0.196 | [0.113, 0.288] | 0.385 | 13 |
| gradient.noise.var | 0.196 | [0.113, 0.289] | 0.385 | 13 |
| hessian.trace | 0.203 | [0.109, 0.306] | 0.615 | 13 |
| hessian.top.eigenvalue | 0.205 | [0.120, 0.297] | 0.538 | 13 |
| tic.bias.term | 0.205 | [0.105, 0.326] | 0.462 | 13 |
| flatness.proxy | 0.211 | [0.106, 0.333] | 0.462 | 13 |
| gradient.norm | 0.213 | [0.110, 0.335] | 0.462 | 13 |
| fisher.rao.norm | 0.214 | [0.118, 0.321] | 0.462 | 13 |
| inverse.margin.p10 | 0.222 | [0.121, 0.333] | 0.615 | 13 |
| cross.entropy | 0.232 | [0.117, 0.358] | 0.538 | 13 |
| negative.entropy | 0.232 | [0.117, 0.361] | 0.538 | 13 |
| ece | 0.244 | [0.133, 0.367] | 0.462 | 13 |
| waic.bias.term | 0.248 | [0.137, 0.376] | 0.462 | 13 |
| reliability.diagram | 0.250 | [0.126, 0.393] | 0.538 | 13 |
| ace | 0.254 | [0.137, 0.376] | 0.538 | 13 |
| mce | 0.258 | [0.137, 0.394] | 0.462 | 13 |
| adaptive.sharpness | 0.263 | [0.144, 0.386] | 0.385 | 13 |
| temperature.scaling | 0.297 | [0.166, 0.436] | 0.462 | 13 |
| pac.bayes.magflat | 0.348 | [0.194, 0.506] | 0.462 | 13 |
| aic.bias.term | 0.376 | [0.228, 0.523] | 0.308 | 13 |
| params | 0.376 | [0.227, 0.526] | 0.308 | 13 |
| vcdim | 0.376 | [0.228, 0.525] | 0.308 | 13 |
| path.norm | 0.404 | [0.241, 0.582] | 0.182 | 11 |
| spectral.norm.per.layer | 0.465 | [0.305, 0.633] | 0.231 | 13 |
| gradient.noise.scale | 0.479 | [0.319, 0.635] | 0.385 | 13 |
| l1.over.margin.p10 | 0.508 | [0.331, 0.684] | 0.231 | 13 |
| pac.bayes.bound | 0.515 | [0.349, 0.679] | 0.154 | 13 |
| l2.over.margin.p10 | 0.523 | [0.352, 0.696] | 0.154 | 13 |
| margin.normalized.param.norm | 0.523 | [0.354, 0.693] | 0.154 | 13 |
| magnitude | 0.538 | [0.362, 0.714] | 0.154 | 13 |
| pac.bayes.magnitude | 0.538 | [0.358, 0.713] | 0.154 | 13 |
| spec.prod | 0.579 | [0.410, 0.742] | 0.077 | 13 |
| spec.sum | 0.604 | [0.449, 0.756] | 0.077 | 13 |
| pac.bayes.magnitude.init | 0.604 | [0.469, 0.742] | 0.000 | 13 |
| frobenius.distance | 0.696 | [0.571, 0.820] | 0.000 | 13 |
| clean.val.acc | 0.170 | [0.066, 0.298] | 0.615 | 13 |
| train.loss | 0.205 | [0.103, 0.328] | 0.615 | 13 |
| train.acc | 0.383 | [0.259, 0.508] | 0.308 | 13 |
| clean.val.loss | 0.432 | [0.231, 0.639] | 0.308 | 13 |
| random | 0.425 | [0.367, 0.475] | 0.201 | 13 |

Table 37: Decision-level selection for minimum OOD generalization gap on CIFAR-10-P: individual selectors.

| Selector | Mean normalized regret ↓ | 95%CI | Top-20% hit ↑ | #candidate |
|---|---|---|---|---|
| oracle.ood | 0.000 | [0.000, 0.000] | 1.000 | 13 |
| sharpness.magnitude.init | 0.104 | [0.059, 0.151] | 0.615 | 13 |
| sharpness.magnitude | 0.115 | [0.065, 0.177] | 0.692 | 13 |
| input.gradient.norm | 0.121 | [0.073, 0.171] | 0.615 | 13 |

| Selector | Mean normalized regret ↓ | 95%CI | Top-20% hit ↑ | #candidate |
|---|---|---|---|---|
| sharpness | 0.122 | [0.071, 0.188] | 0.615 | 13 |
| gradient.noise.final.var | 0.125 | [0.075, 0.180] | 0.462 | 13 |
| gradient.noise.var | 0.125 | [0.073, 0.179] | 0.462 | 13 |
| hessian.trace | 0.138 | [0.087, 0.196] | 0.615 | 13 |
| flatness.proxy | 0.140 | [0.068, 0.239] | 0.538 | 13 |
| tic.bias.term | 0.144 | [0.069, 0.248] | 0.462 | 13 |
| sharpness.magflat | 0.150 | [0.093, 0.217] | 0.538 | 13 |
| gradient.norm | 0.152 | [0.075, 0.254] | 0.462 | 13 |
| waic.bias.term | 0.153 | [0.091, 0.221] | 0.462 | 13 |
| fisher.rao.norm | 0.156 | [0.078, 0.254] | 0.462 | 13 |
| cross.entropy | 0.157 | [0.083, 0.251] | 0.462 | 13 |
| negative.entropy | 0.157 | [0.083, 0.255] | 0.462 | 13 |
| ece | 0.168 | [0.097, 0.263] | 0.385 | 13 |
| inverse.margin.p10 | 0.168 | [0.094, 0.249] | 0.538 | 13 |
| ace | 0.170 | [0.098, 0.251] | 0.462 | 13 |
| reliability.diagram | 0.174 | [0.092, 0.276] | 0.462 | 13 |
| hessian.top.eigenvalue | 0.180 | [0.121, 0.253] | 0.308 | 13 |
| mce | 0.181 | [0.101, 0.280] | 0.385 | 13 |
| adaptive.sharpness | 0.223 | [0.119, 0.341] | 0.462 | 13 |
| temperature.scaling | 0.234 | [0.117, 0.373] | 0.538 | 13 |
| pac.bayes.magflat | 0.312 | [0.180, 0.456] | 0.385 | 13 |
| aic.bias.term | 0.326 | [0.203, 0.461] | 0.385 | 13 |
| params | 0.326 | [0.203, 0.459] | 0.385 | 13 |
| vcdim | 0.326 | [0.200, 0.458] | 0.385 | 13 |
| gradient.noise.scale | 0.451 | [0.301, 0.607] | 0.154 | 13 |
| path.norm | 0.466 | [0.323, 0.620] | 0.091 | 11 |
| spectral.norm.per.layer | 0.554 | [0.390, 0.713] | 0.154 | 13 |
| l2.over.margin.p10 | 0.577 | [0.444, 0.715] | 0.000 | 13 |
| margin.normalized.param.norm | 0.577 | [0.444, 0.714] | 0.000 | 13 |
| l1.over.margin.p10 | 0.579 | [0.443, 0.710] | 0.000 | 13 |
| pac.bayes.bound | 0.597 | [0.467, 0.728] | 0.000 | 13 |
| magnitude | 0.624 | [0.484, 0.769] | 0.000 | 13 |
| pac.bayes.magnitude | 0.624 | [0.483, 0.768] | 0.000 | 13 |
| spec.prod | 0.668 | [0.541, 0.788] | 0.077 | 13 |
| spec.sum | 0.713 | [0.585, 0.830] | 0.000 | 13 |
| pac.bayes.magnitude.init | 0.714 | [0.617, 0.816] | 0.000 | 13 |
| frobenius.distance | 0.810 | [0.722, 0.897] | 0.000 | 13 |
| train.loss | 0.155 | [0.098, 0.219] | 0.615 | 13 |
| clean.val.acc | 0.186 | [0.111, 0.271] | 0.538 | 13 |
| train.acc | 0.298 | [0.203, 0.400] | 0.385 | 13 |
| clean.val.loss | 0.438 | [0.269, 0.621] | 0.154 | 13 |
| random | 0.363 | [0.334, 0.390] | 0.206 | 13 |

The lowest-regret selectors are similar but not identical across targets; ECE is below Random on both but is not the lowest-regret selector.

### C.3.7 Cross-shift proxy-supervised meta-selection

This proxy-supervised analysis uses labeled outcomes from one shift to rank measures before evaluation on the other, without using held-out target outcomes. It tests whether regret rankings transfer approximately between CIFAR-10-C and CIFAR-10-P; architecture, training, and shift construction can break this assumption.

**Protocol.** Using the restriction in Appendix C.3.1, we deduplicate finished runs, fix both shifts at severity 3, and analyze OODGenGap and OODTestGap separately. Requiring at least five runs with finite paired C/P and clean-gap values yields 10 common-support candidate sets: four SimpleCNN, three ResNetV2-32, and three NiN.

The common fold-wise eligibility intersection contains 36 measures per target. Each leave-one-set-out fold ranks measures on the other nine sets, applies the selected measure to the held-out set using its prespecified direction, and averages tied outcomes without target-based tie breaking. Regret uses the normalization in Appendix C.3.1.

Within the nine training sets, mean measure regret is averaged within families to select a family and then its lowest-regret measure. Clean-selected, target-aware, fixed cross-entropy, fixed ECE, and uniform-random references are reported. Intervals rerun the nested procedure over 10,000 candidate-set bootstrap samples and remain descriptive because shared architecture, sweep, or run dependence is not modeled.

Rank transfer compares training-fold proxy regret with held-out target regret at family and measure levels. Fold-wise Kendall $\tau$-a is averaged into $\Psi$; architecture breakdowns report family-rank Kendall $\tau$-b.

Table 38: Cross-shift decision-level transfer for OODGenGap after restricting the analysis to finished runs with source cross-entropy at most 0.01. C→P uses CIFAR-10-C as the proxy shift and CIFAR-10-P as the held-out target shift; P→C reverses these roles. Lower mean normalized regret is better, and higher hit rates are better. Confidence intervals are percentile intervals from 10,000 candidate-set bootstrap resamples.

| Direction | Procedure | Mean normalized regret | 95% CI | Top-10% hit rate | Top-20% hit rate | Folds |
|---|---|---|---|---|---|---|
| C→P | Proxy-selected | 0.196 | [0.070, 0.296] | 0.300 | 0.400 | 10 |
| C→P | Clean-selected | 0.159 | [0.093, 0.318] | 0.300 | 0.500 | 10 |
| C→P | Fixed cross-entropy | 0.197 | [0.102, 0.320] | 0.200 | 0.400 | 10 |
| C→P | Fixed ECE | 0.205 | [0.113, 0.325] | 0.180 | 0.398 | 10 |
| C→P | Target-aware | 0.211 | [0.068, 0.293] | 0.300 | 0.400 | 10 |
| C→P | Random model | 0.377 | [0.353, 0.401] | 0.111 | 0.201 | 10 |
| P→C | Proxy-selected | 0.274 | [0.110, 0.361] | 0.200 | 0.400 | 10 |
| P→C | Clean-selected | 0.207 | [0.144, 0.366] | 0.400 | 0.600 | 10 |
| P→C | Fixed cross-entropy | 0.247 | [0.147, 0.368] | 0.300 | 0.500 | 10 |
| P→C | Fixed ECE | 0.245 | [0.149, 0.364] | 0.276 | 0.498 | 10 |
| P→C | Target-aware | 0.249 | [0.111, 0.355] | 0.200 | 0.400 | 10 |
| P→C | Random model | 0.421 | [0.395, 0.446] | 0.112 | 0.201 | 10 |

Table 39: Transfer of family rankings measured by granulated Kendall $\Psi$ for OODGenGap after restricting the analysis to finished runs with source cross-entropy at most 0.01. Family and measure $\Psi$ compare proxy-shift training regrets with target-shift regret on the held-out candidate set. Higher $\Psi$ and match rates are better; lower family excess regret is better.

| Direction | Mean family $\Psi$ | 95% CI | Top-1 family match rate | Top-2 family match rate | Mean family excess regret | Mean measure $\Psi$ | Folds |
|---|---|---|---|---|---|---|---|
| C→P | 0.467 | [0.220, 0.793] | 0.400 | 0.500 | 0.021 | 0.404 | 10 |
| P→C | 0.467 | [0.233, 0.787] | 0.300 | 0.500 | 0.038 | 0.372 | 10 |

Table 40: Architecture-specific cross-shift results for OODGenGap after restricting the analysis to finished runs with source cross-entropy at most 0.01. The selection rule is learned globally across the training candidate sets and evaluated here by the architecture of the held-out set. Lower regret is better; higher family-rank Kendall $\tau$ is better.

| Direction | Architecture | Proxy-selected regret | Clean-selected regret | Cross-entropy regret | Family-rank Kendall $\tau$ | Sets |
|---|---|---|---|---|---|---|
| C→P | NiN | 0.081 | 0.076 | 0.076 | 0.540 | 3 |
| C→P | ResNetV2-32 | 0.315 | 0.315 | 0.315 | 0.289 | 3 |
| C→P | SimpleCNN | 0.192 | 0.106 | 0.199 | 0.562 | 4 |
| P→C | NiN | 0.102 | 0.074 | 0.074 | 0.632 | 3 |
| P→C | ResNetV2-32 | 0.485 | 0.404 | 0.404 | 0.333 | 3 |
| P→C | SimpleCNN | 0.245 | 0.160 | 0.258 | 0.458 | 4 |

**OOD generalization-gap results.** For OODGenGap, proxy selection improves on Random but not consistently on clean or fixed source-domain references. Family-rank transfer is positive, but best-family recovery is limited and the decision result is architecture dependent.

**OOD test-gap results.** OODTestGap instead measures degradation from clean to shifted test accuracy.

Table 41: Cross-shift decision-level transfer for OODTestGap after restricting the analysis to finished runs with source cross-entropy at most 0.01. C→P uses CIFAR-10-C as the proxy shift and CIFAR-10-P as the held-out target shift; P→C reverses these roles. Lower mean normalized regret is better, and higher hit rates are better. Confidence intervals are percentile intervals from 10,000 candidate-set bootstrap resamples.

| Direction | Procedure | Mean normalized regret | 95% CI | Top-10% hit rate | Top-20% hit rate | Folds |
|---|---|---|---|---|---|---|
| C→P | Proxy-selected | 0.260 | [0.121, 0.466] | 0.300 | 0.600 | 10 |
| C→P | Clean-selected | 0.239 | [0.154, 0.426] | 0.400 | 0.600 | 10 |
| C→P | Fixed cross-entropy | 0.286 | [0.154, 0.427] | 0.300 | 0.500 | 10 |
| C→P | Fixed ECE | 0.280 | [0.147, 0.425] | 0.436 | 0.478 | 10 |
| C→P | Target-aware | 0.295 | [0.120, 0.457] | 0.100 | 0.300 | 10 |
| C→P | Random model | 0.405 | [0.370, 0.441] | 0.111 | 0.201 | 10 |
| P→C | Proxy-selected | 0.352 | [0.173, 0.495] | 0.100 | 0.300 | 10 |
| P→C | Clean-selected | 0.295 | [0.220, 0.472] | 0.300 | 0.500 | 10 |
| P→C | Fixed cross-entropy | 0.341 | [0.218, 0.473] | 0.200 | 0.400 | 10 |
| P→C | Fixed ECE | 0.326 | [0.203, 0.461] | 0.336 | 0.469 | 10 |
| P→C | Target-aware | 0.317 | [0.172, 0.504] | 0.200 | 0.300 | 10 |
| P→C | Random model | 0.449 | [0.410, 0.490] | 0.111 | 0.201 | 10 |

Table 42: Transfer of family rankings measured by granulated Kendall $\Psi$ for OODTestGap after restricting the analysis to finished runs with source cross-entropy at most 0.01. Family and measure $\Psi$ compare proxy-shift training regrets with target-shift regret on the held-out candidate set. Higher $\Psi$ and match rates are better; lower family excess regret is better.

| Direction | Mean family $\Psi$ | 95% CI | Top-1 family match rate | Top-2 family match rate | Mean family excess regret | Mean measure $\Psi$ | Folds |
|---|---|---|---|---|---|---|---|
| C→P | 0.060 | [-0.100, 0.613] | 0.000 | 0.000 | 0.136 | 0.263 | 10 |
| P→C | 0.073 | [-0.093, 0.593] | 0.100 | 0.300 | 0.108 | 0.278 | 10 |

Table 43: Architecture-specific cross-shift results for OODTestGap after restricting the analysis to finished runs with source cross-entropy at most 0.01. The selection rule is learned globally across the training candidate sets and evaluated here by the architecture of the held-out set. Lower regret is better; higher family-rank Kendall $\tau$ is better.

| Direction | Architecture | Proxy-selected regret | Clean-selected regret | Cross-entropy regret | Family-rank Kendall $\tau$ | Sets |
|---|---|---|---|---|---|---|
| C→P | NiN | 0.043 | 0.049 | 0.049 | 0.273 | 3 |
| C→P | ResNetV2-32 | 0.461 | 0.538 | 0.538 | -0.422 | 3 |
| C→P | SimpleCNN | 0.273 | 0.158 | 0.273 | 0.273 | 4 |
| P→C | NiN | 0.172 | 0.112 | 0.112 | 0.362 | 3 |
| P→C | ResNetV2-32 | 0.567 | 0.567 | 0.567 | -0.422 | 3 |
| P→C | SimpleCNN | 0.325 | 0.230 | 0.345 | 0.236 | 4 |

OODTestGap transfer is weaker: family-rank intervals include zero, and ResNetV2-32 transfer is negative in both directions. Proxy selection again improves on Random but not consistently on clean or fixed references.

Varying CIFAR-10-C proxy severity produces non-monotone regret and rank transfer over 5, 3, 10, 3, and 3 eligible sets; this sensitivity check is exploratory and partly reflects changing support.

Because both targets are Gaussian-noise constructions, this is not literal leave-one-corruption-type-out validation. The six tables support only a conditional cross-construction stress test whose transfer must be checked for the intended architecture and regime.

Table 44: Primary residual association for calibration measures. All calibration measures and controls are computed from source CIFAR-10 data. CIFAR-10-C/P quantities are used only as held-out outcomes.

| Target | Calibration measure | Raw $\Psi$ | Residual $\Psi$ | $\Delta\Psi$ | 95% CI | Sweeps | Subspaces | Runs |
|---|---|---|---|---|---|---|---|---|
| CIFAR-10-C | ECE | 0.101 | 0.106 | 0.005 | [0.058, 0.180] | 15 | 1178 | 12887 |
| CIFAR-10-C | MCE | 0.139 | 0.098 | -0.042 | [0.065, 0.159] | 15 | 1178 | 12887 |
| CIFAR-10-C | ACE | 0.099 | 0.112 | 0.014 | [0.058, 0.190] | 15 | 1178 | 12887 |
| CIFAR-10-C | Reliability diagram | 0.139 | 0.111 | -0.028 | [0.076, 0.179] | 15 | 1178 | 12887 |
| CIFAR-10-C | Temperature scaling | 0.104 | 0.053 | -0.051 | [0.009, 0.111] | 15 | 1178 | 12887 |
| CIFAR-10-P | ECE | 0.177 | 0.152 | -0.025 | [0.113, 0.188] | 15 | 1178 | 12887 |
| CIFAR-10-P | MCE | 0.212 | 0.106 | -0.106 | [0.080, 0.135] | 15 | 1178 | 12887 |
| CIFAR-10-P | ACE | 0.178 | 0.159 | -0.019 | [0.103, 0.200] | 15 | 1178 | 12887 |
| CIFAR-10-P | Reliability diagram | 0.211 | 0.118 | -0.093 | [0.091, 0.151] | 15 | 1178 | 12887 |
| CIFAR-10-P | Temperature scaling | 0.185 | 0.053 | -0.132 | [-0.021, 0.131] | 15 | 1178 | 12887 |

Residual $\Psi$ controls for ranked source cross-entropy and source error, where source error is $1 -$ source accuracy. Intervals use sweep-level bootstrap resampling of sweep statistics.

## C.4 Residual Calibration Analysis with Source Loss and Accuracy Controls

### C.4.1 Association residualization

We run an ablation to measure calibration associations after controlling for source-domain loss and accuracy. The analysis uses the same finished CIFAR-10-suite runs as the decision-level selection study. We remove exact duplicate hyperparameter-seed records and analyze SimpleCNN, ResNetV2-32, and NiN. All calibration measures and source controls are computed from source CIFAR-10 data. Source loss is source cross-entropy, and source error is defined as $1 -$ source accuracy. CIFAR-10-C/P values are used only as held-out shifted outcomes or held-out decision-evaluation targets.

The calibration variables are `ece`, `mce`, `ace`, `reliability_diagram`, and `temperature_scaling`. The primary tests focus on `ece` and `mce`; the other calibration measures are reported as secondary checks. The main controls are source cross-entropy and source error. The extended control specification adds `negative_entropy`.

For the association analysis, all variables are converted to percentile ranks within each architecture-sweep group. Let $g$ index an architecture-sweep group, and let $i$ index a run in that group. Define $C_{ig} = \mathrm{rank}_g(c_{ig})$, $Y_{ig} = \mathrm{rank}_g(y_{ig})$, $L_{ig} = \mathrm{rank}_g(\mathrm{CE}_{ig})$, and $E_{ig} = \mathrm{rank}_g(1 - \mathrm{acc}_{\mathrm{source},ig})$. The main residualization fits the following two regressions:

$$C_{ig} = \alpha_g^c + \beta_L^c L_{ig} + \beta_E^c E_{ig} + \varepsilon_{ig}^c \tag{57}$$

$$Y_{ig} = \alpha_g^y + \beta_L^y L_{ig} + \beta_E^y E_{ig} + \varepsilon_{ig}^y \tag{58}$$

Here, the group intercepts $\alpha_g^c$ and $\alpha_g^y$ remove average differences across architecture-sweep groups, while $L_{ig}$ and $E_{ig}$ remove the linear component associated with source loss and source error. The extended specification adds $H_{ig} = \mathrm{rank}_g(h_{ig})$, where $h_{ig}$ is `negative_entropy`, to both regressions. We then compute the same granulated Kendall score $\Psi$ used in the main correlation analysis between $r_c = \widehat{\varepsilon}_{ig}^c$ and $r_y = \widehat{\varepsilon}_{ig}^y$. The shifted outcomes are `OODGenGap_C` and `OODGenGap_P`. Confidence intervals use sweep-level bootstrap resampling of sweep contributions, with the same subspace definitions as the main CIFAR-10 correlation analysis.

Table 44 shows positive residual association for ECE and MCE on both shifted targets after controlling source cross-entropy and source error. The controls reduce several other calibration associations, especially on CIFAR-10-P; these are correlation-level results rather than evidence of incremental decision utility.

Table 45 shows that residual ECE is partly NiN-dependent and decreases further when negative entropy is added as a control. Residual association nevertheless remains in the pooled analysis.

### C.4.2 Decision-level residualization

Table 45: Architecture and control sensitivity for residual calibration association.

| Target | Scope | Measure | Primary $\Psi$ | 95% CI | Extended $\Psi$ | 95% CI | Sweeps | Subspaces | Runs |
|---|---|---|---|---|---|---|---|---|---|
| CIFAR-10-C | All architectures | ECE | 0.106 | [0.058, 0.180] | 0.077 | [0.044, 0.123] | 15 | 1178 | 12887 |
| CIFAR-10-C | All architectures | MCE | 0.098 | [0.065, 0.159] | 0.104 | [0.071, 0.173] | 15 | 1178 | 12887 |
| CIFAR-10-C | Excluding NiN | ECE | 0.056 | [0.036, 0.079] | 0.055 | [0.020, 0.100] | 11 | 888 | 10726 |
| CIFAR-10-C | Excluding NiN | MCE | 0.069 | [0.038, 0.132] | 0.076 | [0.043, 0.150] | 11 | 888 | 10726 |
| CIFAR-10-C | SimpleCNN | ECE | 0.058 | [0.031, 0.096] | 0.038 | [0.005, 0.082] | 6 | 582 | 8938 |
| CIFAR-10-C | SimpleCNN | MCE | 0.038 | [0.016, 0.091] | 0.041 | [0.023, 0.100] | 6 | 582 | 8938 |
| CIFAR-10-C | ResNetV2-32 | ECE | 0.052 | [0.027, 0.075] | 0.089 | [0.043, 0.168] | 5 | 306 | 1788 |
| CIFAR-10-C | ResNetV2-32 | MCE | 0.129 | [0.107, 0.188] | 0.144 | [0.114, 0.195] | 5 | 306 | 1788 |
| CIFAR-10-C | NiN | ECE | 0.260 | [0.219, 0.287] | 0.144 | [0.126, 0.170] | 4 | 290 | 2161 |
| CIFAR-10-C | NiN | MCE | 0.184 | [0.161, 0.213] | 0.190 | [0.143, 0.250] | 4 | 290 | 2161 |
| CIFAR-10-P | All architectures | ECE | 0.152 | [0.113, 0.188] | 0.120 | [0.073, 0.153] | 15 | 1178 | 12887 |
| CIFAR-10-P | All architectures | MCE | 0.106 | [0.080, 0.135] | 0.106 | [0.090, 0.154] | 15 | 1178 | 12887 |
| CIFAR-10-P | Excluding NiN | ECE | 0.127 | [0.083, 0.150] | 0.102 | [0.047, 0.132] | 11 | 888 | 10726 |
| CIFAR-10-P | Excluding NiN | MCE | 0.100 | [0.065, 0.136] | 0.100 | [0.084, 0.157] | 11 | 888 | 10726 |
| CIFAR-10-P | SimpleCNN | ECE | 0.144 | [0.081, 0.168] | 0.093 | [0.014, 0.109] | 6 | 582 | 8938 |
| CIFAR-10-P | SimpleCNN | MCE | 0.092 | [0.030, 0.121] | 0.075 | [0.065, 0.118] | 6 | 582 | 8938 |
| CIFAR-10-P | ResNetV2-32 | ECE | 0.097 | [0.069, 0.114] | 0.117 | [0.049, 0.200] | 5 | 306 | 1788 |
| CIFAR-10-P | ResNetV2-32 | MCE | 0.117 | [0.084, 0.178] | 0.146 | [0.126, 0.208] | 5 | 306 | 1788 |
| CIFAR-10-P | NiN | ECE | 0.227 | [0.199, 0.253] | 0.175 | [0.134, 0.212] | 4 | 290 | 2161 |
| CIFAR-10-P | NiN | MCE | 0.125 | [0.080, 0.149] | 0.126 | [0.077, 0.212] | 4 | 290 | 2161 |

Primary controls are source cross-entropy and source error. Extended controls add negative entropy. Positive values indicate that the residual calibration rank remains positively associated with the residual shifted-gap rank.

For the decision-level ablation, calibration rank is residualized against ranked source cross-entropy and source error within each architecture–sweep group; lower residuals are preferred:

$$\tilde{c}_i = \mathrm{rank}(c_i) - \widehat{\mathbb{E}}[\mathrm{rank}(c_i) \mid \mathrm{rank}(\mathrm{CE}_i), \mathrm{rank}(\text{source error}_i)]. \tag{59}$$

Negative values indicate better calibration rank than predicted by the controls. Raw and residualized selectors use the same candidate sets and regret definition as Appendix C.3.1, without shifted-target information at selection time.

Tables 46 and 47 show that residualized ECE and MCE are weaker than their raw counterparts and source cross-entropy; contrasts against Random remain uncertain.

### C.4.3 Interpretation

This rank-linear ablation is not causal: it can remove useful shared signal as well as confounding variation. It supports only the conclusion that some residual association remains, especially with NiN included, while residualized ECE and MCE do not provide incremental decision utility under the selection protocol.

Table 46: Raw and residualized decision-level selectors.

| Target | Selector | Mean normalized regret | 95% CI | Top-20% hit | Candidate sets |
|---|---|---|---|---|---|
| CIFAR-10-C/P | cross.entropy | 0.195 | [0.125, 0.276] | 0.500 | 26 |
| CIFAR-10-C/P | source accuracy | 0.341 | [0.262, 0.422] | 0.346 | 26 |
| CIFAR-10-C/P | negative.entropy | 0.195 | [0.123, 0.274] | 0.500 | 26 |
| CIFAR-10-C/P | raw ECE | 0.206 | [0.137, 0.284] | 0.423 | 26 |
| CIFAR-10-C/P | residualized ECE | 0.307 | [0.201, 0.426] | 0.385 | 26 |
| CIFAR-10-C/P | raw MCE | 0.219 | [0.143, 0.307] | 0.423 | 26 |
| CIFAR-10-C/P | residualized MCE | 0.382 | [0.287, 0.491] | 0.192 | 26 |
| CIFAR-10-C/P | Random | 0.394 | [0.361, 0.426] | 0.203 | 26 |
| CIFAR-10-C/P | Oracle OOD | 0.000 | [0.000, 0.000] | 1.000 | 26 |
| CIFAR-10-C | cross.entropy | 0.232 | [0.119, 0.362] | 0.538 | 13 |
| CIFAR-10-C | source accuracy | 0.383 | [0.259, 0.507] | 0.308 | 13 |
| CIFAR-10-C | negative.entropy | 0.232 | [0.119, 0.361] | 0.538 | 13 |
| CIFAR-10-C | raw ECE | 0.244 | [0.133, 0.370] | 0.462 | 13 |
| CIFAR-10-C | residualized ECE | 0.339 | [0.185, 0.513] | 0.462 | 13 |
| CIFAR-10-C | raw MCE | 0.258 | [0.136, 0.396] | 0.462 | 13 |
| CIFAR-10-C | residualized MCE | 0.363 | [0.230, 0.511] | 0.231 | 13 |
| CIFAR-10-C | Random | 0.425 | [0.366, 0.473] | 0.201 | 13 |
| CIFAR-10-C | Oracle OOD | 0.000 | [0.000, 0.000] | 1.000 | 13 |
| CIFAR-10-P | cross.entropy | 0.157 | [0.083, 0.255] | 0.462 | 13 |
| CIFAR-10-P | source accuracy | 0.298 | [0.204, 0.405] | 0.385 | 13 |
| CIFAR-10-P | negative.entropy | 0.157 | [0.081, 0.256] | 0.462 | 13 |
| CIFAR-10-P | raw ECE | 0.168 | [0.099, 0.262] | 0.385 | 13 |
| CIFAR-10-P | residualized ECE | 0.275 | [0.141, 0.444] | 0.308 | 13 |
| CIFAR-10-P | raw MCE | 0.181 | [0.102, 0.280] | 0.385 | 13 |
| CIFAR-10-P | residualized MCE | 0.401 | [0.267, 0.555] | 0.154 | 13 |
| CIFAR-10-P | Random | 0.363 | [0.335, 0.390] | 0.206 | 13 |
| CIFAR-10-P | Oracle OOD | 0.000 | [0.000, 0.000] | 1.000 | 13 |

Lower normalized regret is better. Eligible runs have `cross.entropy` $\leq$ 0.01. Source accuracy is read from `final/cifar10/train_acc_eval`; source error is $1 -$ source accuracy. Residualized ECE and MCE use only source CIFAR-10 cross-entropy and source error before selection. Oracle OOD minimizes `OODGenGap_C/P`; target values are used only after selection. Interpretation is conditional on the analyzed candidate sets and the stated cross-entropy restriction; see Sections 3.6 and C.3.1 for provenance, range-restriction, and dependence caveats.

Table 47: Paired decision-level contrasts for residualized calibration selectors.

| Target | Contrast | $\Delta$ regret | 95% paired CI | Paired candidate sets |
|---|---|---|---|---|
| CIFAR-10-C/P | residualized ECE - raw ECE | 0.101 | [0.036, 0.180] | 26 |
| CIFAR-10-C/P | residualized ECE - cross.entropy | 0.112 | [0.050, 0.188] | 26 |
| CIFAR-10-C/P | residualized ECE - Random | -0.087 | [-0.210, 0.053] | 26 |
| CIFAR-10-C/P | residualized MCE - raw MCE | 0.163 | [0.024, 0.300] | 26 |
| CIFAR-10-C/P | residualized MCE - cross.entropy | 0.188 | [0.059, 0.318] | 26 |
| CIFAR-10-C/P | residualized MCE - Random | -0.012 | [-0.107, 0.089] | 26 |
| CIFAR-10-C | residualized ECE - raw ECE | 0.095 | [0.011, 0.189] | 13 |
| CIFAR-10-C | residualized ECE - cross.entropy | 0.107 | [0.025, 0.197] | 13 |
| CIFAR-10-C | residualized ECE - Random | -0.086 | [-0.267, 0.136] | 13 |
| CIFAR-10-C | residualized MCE - raw MCE | 0.105 | [-0.088, 0.276] | 13 |
| CIFAR-10-C | residualized MCE - cross.entropy | 0.131 | [-0.044, 0.290] | 13 |
| CIFAR-10-C | residualized MCE - Random | -0.062 | [-0.176, 0.064] | 13 |
| CIFAR-10-P | residualized ECE - raw ECE | 0.107 | [0.022, 0.233] | 13 |
| CIFAR-10-P | residualized ECE - cross.entropy | 0.118 | [0.031, 0.248] | 13 |
| CIFAR-10-P | residualized ECE - Random | -0.088 | [-0.241, 0.111] | 13 |
| CIFAR-10-P | residualized MCE - raw MCE | 0.221 | [0.028, 0.422] | 13 |
| CIFAR-10-P | residualized MCE - cross.entropy | 0.244 | [0.058, 0.440] | 13 |
| CIFAR-10-P | residualized MCE - Random | 0.038 | [-0.105, 0.190] | 13 |

The contrast is first selector minus second selector. Positive $\Delta$ regret means the first selector has higher regret and is worse. Negative $\Delta$ regret means the first selector has lower regret and is better. Interpretation is conditional on the analyzed candidate sets and the stated cross-entropy restriction; see Sections 3.6 and C.3.1 for provenance, range-restriction, and dependence caveats.

