# OpenReview forum: "Generalization Measures under Controlled Covariate Shift: A Regime-Aware Benchmark"
_TMLR — Under review for TMLR_

### Review · Reviewer_3we8 · 2026-06-28

**Summary Of Contributions:**

This paper revisits the empirical benchmarking of generalization measures, extending the framework of Jiang et al. (2020) from clean (IID) CIFAR-10 evaluation to the controlled image-shift setting of CIFAR-10-C (common corruptions) and CIFAR-10-P (perturbation sequences). The task and label space are held fixed while only the input images are degraded, isolating a covariate-shift regime.

The authors make three contributions:
First, they add two measure families not emphasized in prior generalization-measure benchmarks, i.e., Information Criteria (AIC/TIC/WAIC bias terms) and Calibration & Confidence measures (ECE, MCE, ACE, reliability diagram, temperature scaling), to the conventional Baseline, Norm & Margin, Sharpness, and Optimization families.

Second, they evaluate measures through a three-stage protocol of increasing stringency: a Granulated Kendall correlation score, a local sign-error reliability analysis broken out by hyperparameter axis, and a decision-level model-selection protocol scored by normalized regret and top-k hit rates with an explicit leakage boundary.

Third, they report that under these shifts, several families that look weak under IID evaluation (Optimization, Information Criteria, Calibration) become more informative, and that Calibration & Confidence in particular exhibits the most favorable family-level downside profile when a selector must commit to a family before target evaluation.

The practical recommendation is a regime-aware workflow (validate candidate measures on a target-like proxy corruption rather than trusting an IID benchmark).

**Audience:**

Yes

**Audience Explanation:**

The generalization-measure, robustness, and model-selection-under-shift communities would all find this useful. The central observation that measures discounted under clean IID benchmarking (calibration-style and optimization-dynamics quantities) can carry signal once the test images are corrupted, while traditionally favored norm-and-margin measures degrade to worse than random is a worthwhile corrective to received wisdom from the Jiang et al. (2020) line of work. Independent of whether the specific calibration claim survives the confound scrutiny raised above, the decision-level regret protocol with an explicitly policed leakage boundary is a reusable evaluation contribution, and the negative results (Information Criteria near Random at the decision level; Norm & Margin below Random) are key findings that a benchmark-oriented paper should highlight.

**Broader Impact Concerns:**

There are no Broader Impact Concerns.

**Claims And Evidence:**

No

**Claims Explanation:**

Partially the claims made are supported.

The authors explained what a selector is and is not allowed to see. The three-stage escalation is well established: correlation screens, sign-error checks local reliability, and regret tests whether any of it actually helps the readers to pick a model. Here are a few concerns that needs to be addressed:

i) The word "IID" is being used in a way that is actively misleading to some extent. Throughout the paper, "IID" really just means "the clean CIFAR-10 test set" and "beyond IID" means "corrupted input, same labels, same task". That is covariate shift and calling it "beyond IID" invites the reader to think the paper has said something about distribution shift in general. The genuinely IID related questions, i.e., does independence fail? does identical-distribution fail? in what direction?, are never actually posed.

ii) The paper collapses two structurally different shifts into a single "C/P" target. CIFAR-10-C corrupts each image independently, so the shifted test set is still IID (only the distribution moves). CIFAR-10-P is different in kind: many of the P perturbations are constructed as ordered sequences over a single source image, so frames within those sequences are correlated rather than independent. A measure that aggregates per-point statistics (calibration error, margins) by treating every frame as an independent sample is therefore doing something different on P than on C, and may be effectively re-weighting the same underlying image across perturbation levels. The paper never characterizes this dependence structure or justifies pooling the two. I invite the authors to report C and P separately and to state, for P, how within-sequence correlation is handled in the measure computation.

iii) The paper leans throughout on the assumption that a source domain signal computed before shift will track shifted-test performance, but it treats this source to target relationship as if it were smooth and monotone (https://proceedings.mlr.press/v252/shen24a.html, https://arxiv.org/pdf/2602.07154, https://arxiv.org/pdf/2507.19575, https://proceedings.iclr.cc/paper_files/paper/2024/file/fdb3e1e4907eb5ebac66150b392345ff-Paper-Conference.pdf). Confounding factors can break the assumed monotone relationship: more source-aligned information does not reliably translate into better target behavior (https://neurips.cc/virtual/2025/loc/san-diego/125133). The submission's recommended workflow (validate a measure on a proxy corruption, then trust it on the target) implicitly assumes the proxy to target map is well-behaved. I would encourage the authors to state this assumption explicitly and, where possible, to show how sensitive their family-level conclusions are when the proxy corruption and the evaluation corruption differ in severity or type. Right now the proxy is assumed to be available and faithful, and the whole contribution rests on that.

**Requested Changes:**

Please see the points mentioned before. I highly encourage the authors to address those points which will enhance the quality of the paper.

Furthermore, here are some experimental concerns which needs to be addressed:

i) Table 24 shows that six individual selectors, i.e., gradient.noise.final.var (0.098), gradient.noise.var (0.098), sharpness (0.106), flatness.proxy (0.107), gradient.norm (0.110), cross.entropy (0.120), achieve lower regret than the best individual calibration measure (ece, 0.149). The "Calibration & Confidence wins" claim therefore holds only as a family average with low within-family spread (Table 28), not as predictive superiority. It is recommended that the authors modify the main text to surface Table 24 and the abstract to state the claim as family-level downside robustness under blind commitment. As written, the framing and the evidence point in different directions.

ii) The calibration result is confounded and the confound is testable. ECE/MCE are computed on the source split and are mechanically entangled with source cross-entropy and accuracy, both of which are strong selectors in this same benchmark. A partial-correlation or a decision-level ablation that holds source cross-entropy and accuracy fixed would show whether any calibration-specific signal remains. Without it, the result is consistent with calibration being a proxy for a source-loss quantity.

iii) CIFAR-10-C has five severity levels, and Section B.1 notes severity 1–5 were evaluated for SimpleCNN, but the results are presented without a severity breakdown. Whether a measure's usefulness is monotone in severity or whether the family ordering reorders as corruption intensity grows is exactly the kind of robustness check this paper is about, and it is currently invisible. A severity-stratified version of Table 1 would substantially strengthen or qualify the central claim.

iv) The proxy-validation workflow heavily depends on whether a measure validated on one corruption type predicts performance on a different one. A leave-one-corruption-out experiment (validate the family ranking on a subset of CIFAR-10-C corruptions, test on held-out ones) would convert the workflow recommendation from an assertion into a result. Right now the paper recommends a procedure it does not evaluate under the condition (proxy not equal to target) that makes the procedure non-trivial.

---

> ### Author Response · Authors · 2026-07-14
>
> We thank the reviewer for the careful technical assessment. The revision now defines the controlled-shift scope precisely, reports CIFAR-10-C and CIFAR-10-P separately, makes individual selectors primary, and separates fixed source-domain selection from proxy-supervised meta-selection.
>
> The revision uses the same trained models, CIFAR-10, CIFAR-10-C, and CIFAR-10-P evaluations, measure inventory, and overall three-stage association, local-reliability, and decision structure. It fixes the decision estimand to OODGenGap and strengthens the benchmark with target-separated results and architecture-, objective-, severity-, and proxy-transfer diagnostics.
>
> ## “IID” is operational and the title remains narrowly scoped
>
> We agree that the benchmark does not support broad claims about distribution shift. We changed the title to **“Generalization Measures under Controlled Covariate Shift: A Regime-Aware Benchmark”**. Clean IID evaluation is defined operationally as the clean CIFAR-10 test setting. Here, “controlled covariate shift” refers only to the evaluated, task-preserving CIFAR-10-C/P input shifts, not distribution shift generally; we do not claim an independent empirical test of formal $p(y\mid x)$ invariance.
>
> The Abstract, Introduction, Discussion, Limitations, and Conclusion now limit the claims to the evaluated CIFAR-10-C/P targets, CNN-style architectures, and available run pool. We do not infer that the selector ordering extends to natural or semantic shifts, larger datasets, transformers, or modern augmentation-heavy recipes. We also avoid using “beyond IID” as a general conclusion.
>
> ## CIFAR-10-C and CIFAR-10-P have different sampling structures
>
> All generalization measures are computed from source CIFAR-10 data before shifted-target evaluation. CIFAR-10-P sequence dependence therefore does not enter measure-side aggregation; it enters the definition and uncertainty interpretation of target outcomes.
>
> For the primary CIFAR-10-P Gaussian-noise evaluation at severity 3, the evaluator contains 10,000 source-image sequences with 31 ordered frames per sequence. Sequence boundaries are retained and restored after inference. Because all sequences in that type–severity cell have equal length, a mean over all frames is algebraically equal to first averaging within each sequence and then weighting each source image equally. Flip Probability and mT5D are sequence-aware and compare adjacent frames only within a sequence, never across a sequence boundary.
>
> The empirical results are now separated as well. Tables 35 and 36 give individual-selector decisions for CIFAR-10-C and CIFAR-10-P. The lowest non-oracle point estimate is `input.gradient.norm` at 0.149 for C and `sharpness.magnitude.init` at 0.104 for P. The pooled result is retained only as a compact summary across related but statistically different targets.
>
> We did not add source-image block-bootstrap intervals for target-sampling uncertainty, nor did we compare alternative sequence functionals such as first-frame, last-frame, all-frame, or any-frame accuracy. The revision states this limitation explicitly. The CIFAR-10-P results should therefore be read as benchmark-level sequence summaries, not as estimates obtained by treating every frame as an independent sample.

---

> > ### Author Response · Authors · 2026-07-14
> >
> > ## Proxy transfer is proxy-supervised meta-selection
> >
> > We agree that proxy validation requires an explicit transfer assumption. The new C-to-P and P-to-C analysis is **proxy-supervised meta-selection using labeled proxy outcomes**: for each held-out target candidate set, selector or family rankings are learned from proxy-shift outcomes in the other candidate sets. The held-out target outcomes are not used to construct the ranking. This protocol is separate from the primary fixed source-domain selector benchmark.
> >
> > The required assumption is approximate rank transfer from proxy to target, not a smooth numerical mapping between their performance values. Architecture, objective, severity, and shift construction can all break that assumption. For OODGenGap, Table 37 reports proxy-selected regret of 0.196 for C-to-P and 0.274 for P-to-C, compared with Random at 0.377 and 0.421. However, the clean-selected references are lower at 0.159 and 0.207. Table 38 reports family-rank transfer of 0.467 in both directions, but top-family agreement is limited and the architecture-specific results are heterogeneous.
> >
> > The conclusion also changes with the objective. For OODTestGap, family-rank transfer is only 0.060 for C-to-P and 0.073 for P-to-C, with both intervals including zero (Table 41); ResNet transfer is negative in both directions (Table 42). Thus, positive pooled transfer for one objective is not an architecture-uniform guarantee and does not validate proxy selection generally.
> >
> > We also varied CIFAR-10-C proxy severity from 1 to 5 while holding the CIFAR-10-P target at severity 3. Both proxy regret and rank transfer were non-monotone across 5, 3, 10, 3, and 3 eligible sets. We therefore treat this as exploratory sensitivity rather than evidence of smooth severity transfer (Appendix C.3.7). This is distinct from the fixed-selector severity analysis below, whose counts are 5, 4, 10, 4, and 4.
> >
> > The resulting practical recommendation is therefore conditional: a proxy shift may be considered only after transfer has been checked for the relevant architecture, objective, and shift construction. OODGenGap measures a train-to-shifted-test gap; it is neither shifted-test accuracy nor a deployment outcome. Better OODGenGap selection therefore cannot, by itself, be interpreted as better deployed accuracy.
> >
> > ## Individual selectors are primary and family summaries are secondary
> >
> > We agree that the original family-level emphasis obscured stronger individual measures. Table 1 now foregrounds the individual comparison. The lowest non-oracle pooled mean-regret point estimates are 0.132 for `sharpness.magnitude.init`, 0.135 for `input.gradient.norm`, and 0.144 for `sharpness`. Source `cross.entropy` obtains 0.195 and ECE 0.206. These are exploratory rankings within the stated run pool, not evidence that the leading point estimates are statistically separated or universally best.
> >
> > The family analysis asks a different, descriptive question: what is the average observed outcome if one commits to a family without identifying a particular member? It does not choose the best family member or define an ensemble. Optimization-based and Calibration & Confidence have family-average regrets of 0.221 and 0.223, with a paired difference of 0.002 and interval [-0.047, 0.060] (Tables 2 and 23). The data therefore do not identify a dominant family. The smaller observed spread within the included Calibration members is not a guarantee about an unseen selector and is explicitly conditioned on family composition, support, and missingness.
> >
> > The revised analysis retains finished runs with finite source `cross.entropy <= 0.01`. The retained record does not establish threshold prespecification or sensitivity to alternative cutoffs, and filtering on cross-entropy restricts the range of an evaluated selector. We therefore label all rankings under this restriction exploratory and do not describe the cutoff as a universal convergence criterion.
> >
> > For transparency, submitted Table 24 and the revised decision tables are not directly numerically comparable: the revision fixes the decision target to OODGenGap, applies the stated source-cross-entropy restriction, and evaluates 26 rather than 28 candidate sets. These changes alter the values you cite (`gradient.noise.var` 0.098, `sharpness` 0.106, `cross.entropy` 0.120, and `ece` 0.149); under the revised protocol, the leading individual point estimates are `sharpness.magnitude.init`, `input.gradient.norm`, and `sharpness`, as reported consistently in Tables 1 and 26. The revised analysis supports your central observation: several individual selectors have lower regret than every included calibration selector. The original "Calibration wins" interpretation is not retained; the current family averages do not identify a dominant family, and Calibration's smaller observed within-family spread is descriptive rather than evidence of predictive superiority.

---

> > > ### Author Response · Authors · 2026-07-14
> > >
> > > ## Calibration-specific decision value is not established
> > >
> > > We added rank-residualization analyses controlling for source cross-entropy and source error, followed by a sensitivity analysis that also controls for negative predictive entropy. Some rank association remains for selected measures and targets, but it is architecture-dependent and is not causal evidence.
> > >
> > > The decision-level test is more directly relevant. In Table 45, raw ECE has mean normalized regret 0.206, residualized ECE 0.307, and source cross-entropy 0.195. Raw and residualized MCE obtain 0.219 and 0.382. Table 46 shows that residualized ECE is worse than raw ECE by 0.101, with paired interval [0.036, 0.180], and worse than cross-entropy by 0.112, with interval [0.050, 0.188]. The residual component does not provide incremental selector utility under this specification.
> > >
> > > The common-support ablation in Table 25 reinforces this conclusion: cross-entropy has 0.028 lower regret than the Calibration family average, with interval [-0.059, -0.001], and 0.011 lower regret than ECE, with interval [-0.023, -0.003]. We therefore interpret raw calibration measures as potentially mixing source fit, error, confidence dispersion, and error–confidence alignment, not as evidence for a uniquely calibration-specific mechanism.
> > >
> > > ## Severity results are exploratory and non-monotone
> > >
> > > We added a CIFAR-10-C severity-controlled analysis in Tables 32–34. Candidate sets fix both corruption target and severity before selection, and CIFAR-10-P is excluded. The lowest individual and family point estimates change with severity, while adjacent-severity ranking agreement generally increases at higher severities. These observations do not support a universal monotone claim for Calibration or any other family.
> > >
> > > The candidate-set counts at severities 1–5 are only 5, 4, 10, 4, and 4. We therefore describe the analysis as exploratory. Because support and candidate-set composition vary by severity, changes in the point estimates cannot be attributed to severity alone.
> > >
> > > ## Scope relative to literal leave-one-corruption-type-out validation
> > >
> > > A literal leave-one-CIFAR-10-C-corruption-type-out experiment is not included because the evaluated CIFAR-10-C pool contains only Gaussian-noise corruption, while CIFAR-10-P contains Gaussian-noise perturbation sequences. The C-to-P and P-to-C analysis crosses two evaluation constructions with different sequence structure, but it does not hold out a distinct CIFAR-10-C corruption type.
> > >
> > > Appendix C.3.7 and the Limitations section state this distinction directly, and we do not present C-to-P transfer as a substitute for literal leave-one-corruption-type-out validation. A literal evaluation would require target inference for additional corruption types beyond the evaluated pool.
> > >
> > > The resulting technical contribution is a regime-aware benchmark that separates association, local rank reliability, and decision-level selection across two distinct shift constructions, and then stress-tests transfer across architecture, objective, and severity. It shows that these evaluation levels can favor different measures: sharpness- and input-gradient-based signals have the lowest exploratory OODGenGap point estimates in the evaluated pool, calibration-specific incremental decision value is not established, and proxy transfer must be validated for the intended regime rather than assumed.

---

> ### Comment · Reviewer_3we8 · 2026-07-15
>
> I thank the authors for the detailed response. It addresses my concerns, and the paper is considerably stronger for these revisions. The contributions are well instrumented and the scoping now matches the evidence.
>
> One minor point remains. I appreciate the Table 25 note which discloses that the cross-entropy ≤ 0.01 requirement has no available provenance or sensitivity analysis and restricts the evaluated range of the cross-entropy selector. My concern is that the analysis filters on cross-entropy and then reports cross-entropy among the strongest selectors, so the restriction plausibly shapes the comparison the revised paper now rests on. Table 25 is suggestive here: cross-entropy and negative entropy produce identical differences to three decimals (-0.028 [-0.059, -0.001] against the family average; -0.011 [-0.023, -0.003] against ECE) which is what one would expect if the restriction compresses source loss into a range where the two selectors induce near-identical orderings, though I would also ask the authors to confirm these are not duplicated rows. A sensitivity check at one or two alternative cutoffs would establish whether the ablation's conclusion is robust to the restriction or an artifact of it. I do not regard this as blocking, but it is inexpensive and would materially strengthen the section.

---

> > ### Author Response · Authors · 2026-07-15
> > **Official Comment by Authors**
> >
> > We thank the reviewer for identifying this remaining concern and for suggesting a threshold-sensitivity analysis. In response, we added a sensitivity analysis in Table 26 and expanded the discussion in Appendix C.3.3.
> >
> > First, we confirmed that the identical cross-entropy and negative-entropy results in Table 25 do not reflect duplicated rows or a reporting error. At each evaluated cross-entropy threshold, the two selectors selected the same run in all 26 candidate sets. Their regret contrasts are therefore identical because they produce the same model-selection decisions.
> >
> > Second, we repeated the direct source-statistics ablation using cross-entropy thresholds of 0.005, 0.010, and 0.020. All three thresholds retained the same 26 candidate sets, while the number of eligible run-to-candidate-set memberships ranged from 18,182 to 20,068. As shown in Table 26, the cross-entropy contrasts remained negative and similar in magnitude across thresholds:
> >
> > * Against the Calibration & Confidence family average: −0.032, −0.028, and −0.028.
> > * Against ECE: −0.011 at all three thresholds.
> >
> > The corresponding paired confidence intervals also remained entirely below zero. Within the tested range, source cross-entropy therefore achieved lower normalized regret than the Calibration & Confidence family average and ECE on common candidate-set support. This conclusion was robust to the threshold choice and does not appear to be an artifact of the 0.01 cutoff.

---

### Review · Reviewer_getD · 2026-06-29

**Summary Of Contributions:**

This paper revisits the empirical evaluation of generalization measures beyond the standard IID setting. Building on the benchmark style of Jiang et al. (2020) and the robustness concerns of Dziugaite et al. (2020), the authors study whether generalization measures that predict clean CIFAR-10 generalization gaps also remain useful when the same classifiers are evaluated under controlled image corruptions and perturbations from CIFAR-10-C/P. The paper evaluates several families of measures, including conventional Baseline & Output-based, Norm & Margin-based, Sharpness-based, and Optimization-based measures, while also adding Calibration & Confidence measures and Information Criteria. The main empirical contribution is to show that the usefulness of generalization measures is strongly regime-dependent. Measures that are weak or mixed under IID evaluation can become informative under controlled corruptions or perturbations, and conversely IID predictivity does not automatically imply shifted-test predictivity.

The paper’s main strengths are its clear experimental framing, its careful distinction between correlation-level and decision-level usefulness, and its useful warning that generalization measures should not be selected solely based on IID behavior. The main weaknesses are the limited empirical scope, since the study is restricted to CIFAR-10-C/P and CNN-style architectures, and the fact that the calibration advantage is not mechanistically isolated from correlated source-domain quantities. Some aspects of the decision protocol and artifact availability also need clearer reporting.

**Audience:**

Yes

**Audience Explanation:**

This paper should be of interest to researchers studying generalization, robustness, model selection, calibration, and empirical methodology for evaluating deep learning systems. The central message is useful: a generalization measure that works under clean IID evaluation should not be assumed to remain useful under controlled corruptions or perturbations. This is directly relevant to practitioners and researchers who use proxy measures to choose models before deployment or before evaluating on expensive target distributions.
The paper is also valuable because it connects several lines of work that are often studied separately: generalization-gap prediction, robustness to image corruptions, confidence calibration, and model-selection protocols. Its finding that Calibration & Confidence measures can be useful pre-target ranking signals for CIFAR-10-C/P is likely to interest readers who work on uncertainty estimation and calibration. Meanwhile, the sign-error and decision-level analyses provide useful methodology for researchers who design or evaluate generalization measures.

**Broader Impact Concerns:**

I do not see major ethical concerns that would preclude publication. The paper is primarily an empirical study of generalization-measure evaluation and model selection under controlled image corruptions and perturbations.

The main broader impact concern is that better pre-target model-selection proxies could be used in deployment settings where robustness matters, but also where proxy validation is incomplete. If practitioners overgeneralize the results beyond controlled CIFAR-10-C/P-style shifts, they may select models using measures that do not transfer to real-world distribution shifts. The authors partly address this through their limitations section, and I recommend making this caution prominent. A second concern is that calibration and confidence measures can give a false sense of reliability if used without proper target-like validation. Since the paper recommends validating rankings on a proxy corruption or perturbation close to the expected target, this should be emphasized as a safety-relevant part of the workflow.

**Claims And Evidence:**

Yes

**Claims Explanation:**

The paper’s main claims are supported by a thoughtful and mostly convincing empirical protocol. The authors do not rely on a single global correlation score; instead, they use granulated Kendall scores over local hyperparameter subspaces, which is appropriate given prior concerns that generalization-measure conclusions can be sensitive to the experimental setup. The evidence for regime dependence is clear. The correlation analysis shows that several measure families behave differently under clean IID gaps versus CIFAR-10-C/P gaps. The decision-level model-selection experiment is one of the paper’s strongest parts. There are still limitations that should be clarified.

1. The paper’s conclusions are strongest for CIFAR-10-C/P and CNN-style models, not for natural distribution shifts, larger-scale datasets, transformers, or modern augmentation-heavy training recipes. The authors acknowledge this, but the abstract and conclusion should maintain this careful scope throughout.
2. The paper does not establish a mechanism for why Calibration & Confidence measures work; the authors correctly note that the observed advantage may be correlated with source-domain loss, accuracy, or confidence dispersion.
3.  Family-level averaging is useful for a commit to a family protocol, but it can hide strong individual selectors from other families, as shown by the individual selector results where some Optimization-based and Sharpness-based measures obtain lower regret than individual calibration measures.

Overall,  the core claims are well supported when interpreted within the paper’s stated CIFAR-10-C/P scope.

**Requested Changes:**

1. Keep the scope precise throughout the paper. The strongest evidence is for CIFAR-10-C/P controlled image corruptions and perturbations with CNN-style architectures. The authors should ensure that the abstract, introduction, discussion, and conclusion consistently avoid implying broader claims about non-IID generalization in general. The current limitations section is appropriately cautious, but the main narrative should preserve that same caution.

2. Clarify the calibration-measure protocol. The paper states that Calibration & Confidence measures use source-domain labels and not shifted-test labels. This is an important point and should be emphasized clearly wherever calibration results are discussed.

3. Discuss whether source validation accuracy/loss is an overly strong baseline or an insufficient one. The paper compares against train and clean-validation quantities, and reports clean-test references in the appendix. It would be helpful to explain why the chosen source/clean baselines are the right practical baselines, and whether stronger clean-validation-derived baselines could narrow the gap.

4. Expand the mechanism discussion.  The paper correctly avoids overclaiming a causal mechanism for Calibration & Confidence. Still, it would be useful to add a short analysis or discussion disentangling calibration-specific signal from source loss, source accuracy, entropy, confidence dispersion, or cross-entropy. Even a correlation matrix among these pre-target quantities could help readers understand what information the successful selectors may be using.

---

> ### Author Response · Authors · 2026-07-14
>
> We thank the reviewer for the constructive assessment. The revision makes the CIFAR-10-C/P scope and the source-supervised, target-blind information boundary explicit, adds direct source/clean baselines and calibration-residualization analyses, and presents individual selectors as the primary result while treating family averages as descriptive stress tests.
>
> The revision uses the same trained models, CIFAR-10, CIFAR-10-C, and CIFAR-10-P evaluations, measure inventory, and overall three-stage association, local-reliability, and decision structure. It fixes the decision estimand to OODGenGap and strengthens the benchmark with explicit information boundaries, named source and clean-validation baselines, and association- and decision-level tests of calibration-specific information.
>
> ## The claims are limited to the benchmark actually evaluated
>
> We agree that the conclusions should remain limited to the evaluated CIFAR-10-C/P setting. We changed the title to **“Generalization Measures under Controlled Covariate Shift: A Regime-Aware Benchmark”** and define clean IID evaluation operationally as the clean CIFAR-10 test setting. Here, “controlled covariate shift” refers only to the evaluated, task-preserving CIFAR-10-C/P input shifts, not distribution shift generally; the additional sequence dependence of CIFAR-10-P is also stated explicitly.
>
> The Abstract, Introduction, Discussion, Limitations, and Conclusion now restrict the conclusions to the evaluated CIFAR-10-C/P constructions, CNN-style architectures, and training runs. We do not extrapolate the selector ordering to natural or semantic shifts, larger datasets, transformers, or modern augmentation-heavy training. Architecture dependence is also visible within the benchmark: for example, the positive family-level Calibration result on CIFAR-10-C is concentrated in NiN rather than being uniform across architectures.
>
> ## The primary protocol is source-supervised and target-blind
>
> Calibration and confidence measures use labeled source-domain CIFAR-10 examples. They are therefore not label-free. During selection, however, they use no CIFAR-10-C/P images, labels, accuracies, losses, gaps, target rankings, or empirically chosen correlation signs. Shifted-target quantities are revealed only after selection to evaluate the decision.
>
> The revised data-split description distinguishes selector inputs from evaluation outcomes. In the primary OODGenGap fixed-selector analysis, applicable measures use source-training labels, clean-validation selectors use validation labels, and clean-test quantities are excluded from selection and shown only as a separate reference. Clean-test and CIFAR-10-C/P quantities otherwise enter only as evaluation outcomes. This information regime defines the primary fixed-selector benchmark. It is distinct from the proxy-supervised meta-selection analysis discussed below, where labeled outcomes from a separate proxy shift are intentionally used to learn a selector ranking.
>
> ## Individual source and clean-validation baselines are shown directly
>
> We agree that a heterogeneous family average is not an adequate substitute for strong individual baselines. Table 1 now places the main individual selectors side by side. Mean normalized regret is 0.178 for `clean.val.acc`, 0.195 for source `cross.entropy`, 0.206 for ECE, and 0.394 for Random; `clean.val.loss` obtains 0.435. Thus, clean-validation accuracy is a competitive practical baseline, whereas access to a clean validation split does not make every clean-derived selector effective.
>
> Clean-test accuracy and loss remain excluded from the primary selectors because they use the benchmark test split. Table 24 reports them only as an optimistic clean-test reference in the primary decision analysis. That reference has mean regret 0.326, compared with 0.283 for the Source/clean baseline group, so an additional clean split does not automatically improve selection for the shifted objective. We did not evaluate combined clean-validation selectors; the revision therefore treats the Source/clean baseline set as practical but non-exhaustive.
>
> At the family level, Calibration minus Source/clean has a mean-regret difference of -0.060 with candidate-set bootstrap interval [-0.135, 0.011] (Table 2). Because the interval includes zero, we do not claim a clear family-level advantage over practical source-domain baselines.

---

> > ### Author Response · Authors · 2026-07-14
> >
> > ## The mechanism analysis does not establish calibration-specific utility
> >
> > We added rank-residualization diagnostics controlling for source cross-entropy and source error, with negative predictive entropy included in a further sensitivity analysis. Some calibration association remains after these rank-linear controls, but it is measure- and architecture-dependent. This is conditional association, not causal identification.
> >
> > At the selector level, Table 45 provides the most direct check. Mean normalized regret is 0.195 for source cross-entropy, 0.206 for raw ECE, and 0.307 for residualized ECE; raw and residualized MCE obtain 0.219 and 0.382, respectively, while Random obtains 0.394. Table 46 shows that residualized ECE is worse than raw ECE by 0.101, with paired interval [0.036, 0.180], and worse than source cross-entropy by 0.112, with interval [0.050, 0.188]. The residual component therefore does not provide incremental decision utility under this specification.
> >
> > The common-support comparison reaches the same practical conclusion. In Table 25, source cross-entropy has 0.028 lower regret than the Calibration family average, with interval [-0.059, -0.001], and 0.011 lower regret than ECE, with interval [-0.023, -0.003]. Negative entropy produces the same reported paired regret point estimates in this restricted pool. We consequently interpret raw calibration quantities as potentially combining source fit, error, confidence dispersion, and error–confidence alignment; we do not attribute the results to a uniquely calibration-specific mechanism.
> >
> > ## Individual selectors and family averages answer different questions
> >
> > The primary decision question is which fixed individual signal selects models with low OODGenGap. Table 1 reports the lowest non-oracle point estimates for `sharpness.magnitude.init` (0.132), `input.gradient.norm` (0.135), and `sharpness` (0.144). These point estimates are lower than those for source cross-entropy and ECE. We therefore no longer recommend Calibration & Confidence as the best selector.
> >
> > The family analysis instead averages the observed outcomes of included family members. It does not select the best member, construct an ensemble, or define a deployable family-level rule. Optimization-based and Calibration & Confidence have similar family-average regrets of 0.221 and 0.223, and their paired difference is 0.002 with interval [-0.047, 0.060] (Tables 2 and 23). The smaller observed Calibration spread is described only as a property of the included members and support; it is not a guarantee for unseen measures.
> >
> > All of these rankings are conditional on retaining finished runs with finite source `cross.entropy <= 0.01`. The retained record does not establish that this threshold was prespecified or that the conclusions are insensitive to alternative thresholds. Because the restriction also truncates the range of an evaluated selector, we label the ranking analysis exploratory and do not present the threshold as a universal convergence rule.
> >
> > ## Proxy transfer is a separate diagnostic, not a deployment guarantee
> >
> > The C-to-P and P-to-C analysis is **proxy-supervised meta-selection**: selector or family rankings are learned from labeled proxy-shift outcomes in the training candidate sets, while the held-out target candidate set is not used for ranking. It is not the fixed source-domain selector protocol.
> >
> > For OODGenGap, proxy-selected regret is 0.196 for C-to-P and 0.274 for P-to-C, compared with Random at 0.377 and 0.421, but clean-selected references are lower at 0.159 and 0.207 (Table 37). For the alternative OODTestGap objective, family-level transfer is weak—0.060 and 0.073, with intervals including zero—and ResNet transfer is negative in both directions (Tables 41–42). We therefore present proxy transfer as an objective- and architecture-dependent diagnostic. OODGenGap is a train-to-shifted-test gap objective; it is not shifted accuracy and does not by itself establish deployment utility.
> >
> > Taken together, the revision contributes a regime-aware, source-supervised benchmark that separates association, local rank reliability, and decision-level regret; compares named measures with practical source and clean-validation baselines; and tests whether calibration information survives controls for source fit and error. Within the evaluated run pool, sharpness- and input-gradient-based selectors have the lowest exploratory OODGenGap point estimates, while calibration-specific incremental decision value is not established and proxy transfer remains architecture- and objective-dependent.

---

### Review · Reviewer_KjpB · 2026-07-02

**Summary Of Contributions:**

This paper extends the generalization-measure benchmark of Jiang et al. (2020) from clean IID data to controlled image shifts (CIFAR-10-C/P), where the task stays fixed but inputs are corrupted or perturbed. It adds two candidate families not previously studied this way, Calibration & Confidence and Information Criteria, and evaluates measures in three stages: a Kendall correlation score, a sign-error check for local reliability, and a decision-level test of whether a measure can pick a good model before shifted-test labels are seen. The main claims are that IID predictivity does not reliably transfer to shift, and that the Calibration & Confidence family has the best downside profile (lowest regret 0.178, highest top-20 hit 0.543). Strengths are the careful scoping and the decision-level protocol. Weaknesses are that the headline depends on committing to a whole family rather than a single measure, the calibration effect is not separated from simple source loss, competitor baselines are missing, and the effect is small and NiN-driven.

**Audience:**

Yes

**Audience Explanation:**

The question is well motivated and relevant to a real audience: many people rely on generalization measures validated only on clean IID data, and knowing that this predictivity does not reliably carry over to corrupted or perturbed inputs is useful to have on record. The decision-level protocol, with its explicit leakage boundary and regret-based scoring, is a contribution in its own right and could be reused by others studying pre-target model selection, independent of the specific family ranking. Researchers working on robustness, calibration, and generalization-measure benchmarking would find both the negative transfer result and the expanded taxonomy worth reading, even if the headline calibration claim needs the revisions noted above.

**Broader Impact Concerns:**

None. This is a benchmarking and methodology paper on how to choose generalization measures under image corruptions. It does not raise ethical or dual-use concerns, and no Broader Impact Statement is needed.

**Claims And Evidence:**

No

**Claims Explanation:**

The narrow claim holds: inside their family-level protocol, calibration does have the lowest regret and highest top-20 hit (Table 1). My concern is the broader takeaway that calibration carries a special signal and should be a default.

The setup does not match practice. It forces a commitment to a whole family and averages inside it, but people pick one measure. In Table 24, the best single selectors (gradient noise variance 0.098, sharpness 0.106, cross.entropy 0.120) all beat the best calibration measure (ece 0.149) and the calibration average (0.178). So the family result is about low variance, not best performance.

The contrast is also partly a grouping artifact. cross.entropy sits in the weak-looking Baseline family yet is one of the strongest single selectors, which makes the "calibration strong, baseline weak" framing misleading.

The effect is not isolated from a simpler cause. Calibration errors come from the same source predictions as source cross-entropy, which is the stronger selector, and the authors admit they cannot separate the two. The ablation that would settle this is missing though the runs exist. The evidence is also thin: overlapping intervals, a key paired difference of -0.090 with upper bound near zero, and a signal largely driven by NiN.

Finally, the obvious competitors from unsupervised accuracy estimation (for example ATC) are absent, and they are the natural baseline against a calibration family.

**Requested Changes:**

Add the source-baseline ablation. Show whether source cross-entropy, or a simple source confidence statistic, already gives the same decision-level benefit that is credited to the calibration family. This is the most important change, and the runs already exist so it should be cheap.

Justify the family-level setup, or change the headline. Either explain why committing to a whole family is the right way to think about model selection, or reframe the main claim so it says what is actually true: the calibration family is stable and low-variance, while the strongest single selectors are optimization, sharpness, and cross-entropy.

Add at least one shift-prediction baseline, ATC being the natural one, and add a related-work paragraph on unsupervised accuracy estimation under shift.

Bring the paired-bootstrap intervals into the main text and soften the abstract so the wording matches how small and overlapping the differences are.

Fix the cross-entropy presentation so readers are not misled by the weak Baseline & Output family average hiding a strong single measure.

---

> ### Author Response · Authors · 2026-07-14
>
> We thank the reviewer for highlighting the distinction between family-average stability and individual-selector performance. The revision now foregrounds individual measures, simple source-domain baselines, uncertainty, and architecture dependence; Calibration & Confidence is no longer presented as a default selector.
>
> The revision uses the same trained models, CIFAR-10, CIFAR-10-C, and CIFAR-10-P evaluations, measure inventory, and overall three-stage association, local-reliability, and decision structure. It fixes the decision estimand to OODGenGap, replaces the family-level headline with individual-selector reporting, and adds common-support source baselines, paired uncertainty, architecture-specific analyses, and calibration-residualization diagnostics.
>
> ## Individual measures are primary; the filtered ranking is exploratory
>
> We agree that a practitioner normally fixes a specific measure rather than committing blindly to a family. Table 1 now foregrounds individual selectors: under the revised analysis restriction, the lowest non-oracle point estimates of mean normalized regret are 0.132 for `sharpness.magnitude.init`, 0.135 for `input.gradient.norm`, and 0.144 for `sharpness`; `clean.val.acc`, source `cross.entropy`, and ECE obtain 0.178, 0.195, and 0.206. These are exploratory point-estimate rankings for selecting low train-to-shifted-test OODGenGap, not claims of statistical superiority, highest shifted accuracy, or deployment utility.
>
> The restriction retains finished runs with finite source `cross.entropy <= 0.01`. The retained analysis record does not establish threshold prespecification or sensitivity, and filtering on an evaluated selector restricts its range. We therefore do not call this a universal convergence criterion and limit every ranking claim to the stated restricted pool. Section 3.6 defines the restriction and estimand; Section 4.3 and Table 1 give the compact comparison, and Table 26 gives the full ranking.
>
> For transparency, the submitted and revised decision tables are not directly numerically comparable: the revision fixes the decision target to OODGenGap, applies the stated source-cross-entropy restriction, and evaluates 26 rather than 28 candidate sets. These changes alter the submitted values you cite, including the Calibration family average of 0.178 and the individual regrets in submitted Table 24; we report the revised values consistently and label the ranking exploratory. The revised analysis supports the central point of your comment: several individual sharpness-, optimization-, and loss-based selectors have lower regret than every included calibration selector, while the family averages do not identify a dominant family.
>
> The decision unit is a candidate set defined before selector evaluation. A selector ranks the eligible models using only its permitted source-domain quantity, and normalized regret compares the selected model's OODGenGap with the best and worst eligible outcomes in that set. This normalization makes candidate sets comparable, but it does not convert OODGenGap into target accuracy. It also means that each result is conditional on the eligible models and observed range in that candidate set. We now state these conditions alongside the main ranking rather than presenting the pooled average as an unconditional property of a measure.
>
> ## Family averages are descriptive, not an implementable selector
>
> We agree that the original family-level headline was too strong. The family analysis averages the observed outcomes of included individual selectors; it neither chooses the best member nor defines an ensemble. Optimization-based and Calibration & Confidence have mean regrets of 0.221 and 0.223, with a paired difference of 0.002 and a candidate-set bootstrap interval of [-0.047, 0.060]. The data therefore do not identify a dominant family.
>
> The revised Abstract no longer identifies Calibration as the winning family, and Section 4.3 and Table 2 now place the paired Calibration-minus-Optimization interval in the main text.
>
> We now describe the 0.059 Calibration max–min spread only as the observed spread among included measures. It depends on family composition, member count, support, and missingness and is not a guarantee about an unseen measure. This interpretation appears in Sections 3.6 and 4.3 and Appendix C.3.2 and C.3.4; Tables 2, 23, and 30 report the corresponding summaries.
>
> The architecture-specific family results further show why the pooled average cannot support a default recommendation. The lowest family point estimate differs across SimpleCNN, ResNetV2-32, and NiN (Tables 27–29). We therefore keep family summaries as a secondary description of the measures observed in this benchmark and base practical comparisons on named individual selectors with explicit information requirements.

---

> > ### Author Response · Authors · 2026-07-14
> >
> > ## Source statistics directly address the grouping-artifact concern
> >
> > We added the requested common-support comparison, and it supports the reviewer’s concern. Source cross-entropy has 0.028 lower normalized regret than the Calibration family average, with candidate-set bootstrap interval [-0.059, -0.001], and 0.011 lower regret than ECE, with interval [-0.023, -0.003]. Negative entropy yields the same reported paired regret point estimates in this restricted pool.
> >
> > Cross-entropy is now shown directly in Table 1 rather than characterized through the heterogeneous Baseline & Output family average. We also distinguish post-training evaluation-mode `cross.entropy` from final-loop `train.loss`, which can reflect training-mode augmentation, dropout, and batch-normalization behavior. The common-support result is in Appendix C.3.3 and Table 25; complete individual results are in Table 26.
> >
> > ## Calibration-specific decision value is not established
> >
> > We agree that calibration is statistically coupled to source loss, error, and confidence. The decisive result is at the decision level: residualizing ranked ECE against ranked source cross-entropy and source error increases mean regret from 0.206 to 0.307, compared with 0.195 for source cross-entropy. Thus, residual association under the specified rank-linear controls does not yield incremental decision utility and is not causal identification.
> >
> > Appendix C.4 defines the residualization and reports association sensitivity in Tables 43–44. The residualized-selector decision results are in Table 45 and paired contrasts in Table 46; Section 5.2 states the resulting limited interpretation.
> >
> > ## Uncertainty and architecture dependence are explicit
> >
> > We agree that the original presentation obscured the NiN contribution. For CIFAR-10-C, Calibration & Confidence has family-level granulated scores of 0.044 for SimpleCNN, 0.092 for ResNetV2-32, and 0.355 for NiN. We therefore do not describe the correlation result as architecture uniform.
> >
> > Section 4.1 and Table 6 report this dependence, while Tables 27–29 show that the lowest family point estimate differs across architectures. Section 4.3 also treats candidate-set bootstrap intervals as conditional descriptive summaries because architecture/sweep units can overlap.
> >
> > The reported intervals quantify variation over the observed candidate sets; they are not population guarantees for new architectures or shift families. In particular, the pooled point estimates should not be read as resolving close selector comparisons when their uncertainty is not separated. We report paired directional differences within the observed benchmark; the intervals are conditional/descriptive and are not interpreted as population-level superiority because shared dependence is not fully modeled.
> >
> > ## Scope relative to ATC
> >
> > A matched ATC experiment is not included in this revision. The comparison differs in both information regime and estimand: our primary fixed-selector protocol uses labeled source information but no shifted-target inputs or outcomes and evaluates selection for low OODGenGap, whereas ATC learns a source confidence threshold, uses predictions on unlabeled target inputs, and estimates target accuracy. Consequently, the present evidence does not compare target-accuracy selection with ATC.
> >
> > We therefore keep ATC outside the empirical comparison: Oracle OOD and Random are evaluation references, not substitutes for it. Section 2.1 and the “Information regimes and missing comparisons” limitation define this boundary, and the manuscript claims neither superiority over target-unlabeled accuracy estimators nor selection for absolute target accuracy.
> >
> > The resulting technical contribution is a source-domain, regime-aware benchmark for controlled CIFAR-10-C/P gap objectives that separates association, local rank reliability, and decision-level model selection under an explicit information boundary. It shows empirically that these evaluation levels need not favor the same measures and identifies where individual selectors, simple source baselines, architecture dependence, and calibration-specific diagnostics change the conclusion.

---

### Comment · Action_Editor_FJKD · 2026-06-20
**Review deadline**

Dear Reviewers,

Our review deadline is 04 Jul 2026, 23:59 AOE time.

Regards,
Action Editor